# Revisiting $1$-peer exponential graph for enhancing decentralized learning efficiency

**Kenta Niwa**
NTT Communication Science Laboratories
<kenta.niwa@ntt.com>

**Yuki Takezawa**
Kyoto University
Okinawa Institute of Science and Technology
<yuki-takezawa@ml.ist.i.kyoto-u.ac.jp>

**Guoqiang Zhang**
University of Exeter
<G.Z.Zhang@exeter.ac.uk>

**W. Bastiaan Kleijn**
Victoria University of Wellington
<bastiaan.kleijn@vuw.ac.nz>

## Abstract

For communication-efficient decentralized learning, it is essential to employ dynamic graphs designed to improve the expected spectral gap by reducing deviations from global averaging. The 1-peer exponential graph demonstrates its finite-time convergence property–achieved by maximizing the expected spectral gap–but only when the number of nodes $n$ is a power of two. However, its efficiency across any $n$ and the commutativity of mixing matrices remain unexplored. We delve into the principles underlying the 1-peer exponential graph to explain its efficiency across any $n$ and leverage them to develop new dynamic graphs. We propose two new dynamic graphs: the $k$-peer exponential graph and the null-cascade graph. Notably, the null-cascade graph achieves finite-time convergence for any $n$ while ensuring commutativity. Our experiments confirm the effectiveness of these new graphs, particularly the null-cascade graph, in most test settings. https://github.com/garden1984/NullCascadeGraph

## 1 Introduction

In the era of large-scale datasets, complex models (e.g., Deep Neural Networks: DNNs), and powerful computing resources capable of high throughput and simultaneous multicasting, decentralized learning of models over arbitrary network topologies (graphs)–for tasks such as natural language processing and image recognition–has been essential for advanced collective intelligence. The convergence rates of decentralized learning algorithms, including DSGD [20, 26], are primarily influenced by the graphs used for partial averaging of models within $n \in \mathbb{N}$ nodes [13]. Thus, optimizing the communication graphs to improve the expected spectral gap (by reducing deviations from global averaging), given a maximum degree $k \in \mathbb{N}$ (i.e., the maximum number of connections or multicast transmissions per node), is vital for communication-efficient decentralized learning.

Graphs used in decentralized learning are broadly categorized into static ($\tau=1$) and dynamic ($\tau \geq 2$), with the dynamic graphs undergoing changes involving a periodic interval $\tau \in \mathbb{N}$. In contrast, static graphs, such as the ring, grid, torus [24], and exponential graph [1, 43], remain unchanged over time. Dynamic graphs rely on $\tau$ mixing matrices, specifying varying connection paths and weights. A noteworthy dynamic graph is the 1-peer exponential graph [43], characterized by 1-peer communication and known to achieve desirable finite-time convergence–i.e., the expected spectral gap of the product of $\tau$ mixing matrices is maximized when $n$ is a power of two, as in Table 1.

The introduction of $k$-peer communication with $k \geq 2$ offers the potential for improved decentralized learning, especially in settings where simultaneous multicasting is supported. This scenario has been

Table 1: Comparison of dynamic graphs. The expected spectral gap $p$ and the periodic interval $\tau$ are defined in Assumption 4 in Sec. 2. The $\tau$ is estimated when $n$ is factorized as $n = \prod_{i=1}^{\lambda}(\nu_i+1)$, where $\nu_1 \leq ... \leq \nu_\kappa \leq k < \nu_{\kappa+1} \leq ... \leq \nu_\lambda$, and each $\nu_i$ is a natural number, with $\kappa$ factors being less than or equal to $k$. Uniform degree is satisfied ($\checkmark$) when every node has the same number of connections.

| Dynamic graphs | # of nodes $n$ | Max. degree $k$ | Uniform degree | Expected spectral gap $p$ | Periodic interval $\tau$ | Commu tativity |
|---|---|---|---|---|---|---|
| 1-peer hypercube [31] | A power of 2 | 1 | $\checkmark$ | 1 (finite-time convergence) for limited $n$ | $\log_2(n)$ | $\checkmark$ |
| 1-peer exponential [43] | $\forall n \in \mathbb{N}$ | 1 | $\checkmark$ | Not analyzed for any $n$ | $\lceil \log_2(n) \rceil$ | $\checkmark$ |
| Base-$(k{+}1)$ [35] | $\forall n \in \mathbb{N}$ | $\forall k \in \mathbb{N}$ | – | 1 (finite-time convergence) | $\kappa + \sum_{i=\kappa+1}^{\lambda} 2\lceil \log_{k+1}(\nu_i) \rceil$ | – |
| $k$-peer exponential (Sec. 3) | $\forall n \in \mathbb{N}$ | $\forall k \in \mathbb{N}$ | $\checkmark$ | $1 - \max\limits_{i=1,...,n-1} \left\| \frac{1}{(k+1)^\tau} \frac{\sin(\pi i(k+1)^\tau/n)}{\sin(\pi i/n)} \right\|$ | $\lfloor \log_{k+1}(n) \rfloor$ | $\checkmark$ |
| Null-cascade (Sec. 4) | $\forall n \in \mathbb{N}$ | $\forall k \in \mathbb{N}$ s.t. $k \geq 2$ | $\checkmark$ | 1 (finite-time convergence) | $\kappa + \sum_{i=\kappa+1}^{\lambda} \nu_i/(2\lfloor k/2 \rfloor)$ | $\checkmark$ |

explored, such as the base-$(k+1)$ graph [35], which achieves finite-time convergence for any $(n, k)$ configuration. Nevertheless, several challenges remain. First, the required periodic intervals $\tau$ can become large depending on $(n, k)$, and the emergence of nonuniform degrees (i.e., unequal numbers of connections per node) can hinder convergence rates in practice. Second, the base-$(k+1)$ graph does not ensure commutativity of the $\tau$ mixing matrices (see Definition 4 in Sec. 2), necessitating strict coordination of periodic and synchronized communication cycles across $n$ nodes.

For further communication-efficient decentralized learning by leveraging $k$-peer communication with $k \geq 2$, we propose two new dynamic graphs: the $k$-peer exponential graph and the null-cascade graph. Our key contributions are as follows:

**Exploring principles underlying the 1-peer exponential graph.** We revisit the advantageous properties of the 1-peer exponential graph. Ying et al. [43] proved its finite-time convergence only when $n$ is a power of two. However, its efficiency across any $n$ and the commutativity of its mixing matrices remain unexplored, warranting further investigation. In Sec. 3, we explore the extension to the $k$-peer exponential graph, which generalizes the 1-peer exponential graph to allow $k$-peer communication. The $k$-peer exponential graph, with $k \geq 1$, is characterized by $\tau$ circulant mixing matrices of uniform weights. Their eigenvalues are analytically determined by the Discrete Fourier Transform (DFT) of expansions of Moving Average Filters (MAFs) with uniform weights. This allows us to analyze the efficiencies of the $k$-peer exponential graph for any $n$ by utilizing established Digital Signal Processing (DSP) techniques. Our findings are twofold: i) finite-time convergence can be interpreted as isolating the Direct Current (DC) component and eliminating (nullifying) non-zero frequency components through a cascade of $\tau$ mixing matrices, and ii) the $k$-peer exponential graph functions effectively by isolating the DC component while nullifying non-zero frequency components. Although this suggests desiring finite-time convergence is only achieved when $n$ is a power of $k+1$.

**Proposition of null-cascade graph.** To achieve finite-time convergence for any $n$ while preserving commutativity, our approach builds on the principles underlying the $k$-peer exponential graph. Our main idea is to incorporate not only MAFs with uniform weights but also Steerable Nulling Filters (SNFs) with non-uniform weights to nullify all non-zero frequencies, as detailed in Sec. 4. We refer to the resulting graph as the null-cascade graph, which ensures finite-time convergence for any $n$ while maintaining commutativity. Limitations in the null-cascade graph are that i) it requires $k \geq 2$ for forming SNFs and ii) it is most effective when the factorization of $n$ does not include large prime numbers, as it yields a comparatively small increase in the periodic interval $\tau$ (see Table 1). The effectiveness of the proposed graphs, particularly the null-cascade graph, was confirmed through numerical experiments in Sec. 5.

## 2 Preliminaries

The simplest decentralized learning algorithm, DSGD [16], and its convergence rate are presented in Sec. 2.1. As a baseline graph, 1-peer exponential graph is introduced in Sec. 2.2.

### 2.1 DSGD and its convergence rate

Consider a network consisting of $n$ local nodes collaborating to solve an optimization problem:

$$\min_{\mathbf{x} \in \mathbb{R}^d} \left\{ f(\mathbf{x}) = \frac{1}{n} \sum_{i=1}^{n} f_i(\mathbf{x}) \right\}, \tag{1}$$

where $f_i : \mathbb{R}^d \to \mathbb{R}$ denotes the local loss function and differentiable. A fundamental algorithm for solving this problem is DSGD. It iteratively repeats two steps: (i) local updates of $n$ local models $\mathbf{X} = [\mathbf{x}_1, \ldots, \mathbf{x}_n] \in \mathbb{R}^{d \times n}$ and (ii) partial mixing of local parameters using mixing matrices. Following [13, 43, 35], we consider dynamic communication among local nodes by periodically repeating a sequence of $\tau$ mixing matrices $\{\boldsymbol{W}^{(0)}, \ldots, \boldsymbol{W}^{(\tau-1)}\} \in \mathbb{R}_{\geq 0}^{(n \times n) \times \tau}$. Each mixing matrix is doubly stochastic, satisfying $\boldsymbol{W}^{(\ell)} \mathbf{1}_n = \boldsymbol{W}^{(\ell)\top} \mathbf{1}_n = \mathbf{1}_n$. The update rules are given by:

$$\text{(i) } \mathbf{X}^{(r+\frac{1}{2})} = \mathbf{X}^{(r)} - \eta \nabla F(\mathbf{X}^{(r)}; \xi^{(r)}), \qquad \text{(ii) } \mathbf{X}^{(r+1)} = \mathbf{X}^{(r+\frac{1}{2})} \boldsymbol{W}^{(\mathrm{mod}(r,\tau))}, \qquad (2)$$

where $r \in \{0, \ldots, R-1\}$ denotes the index of communication round, $\eta$ is the learning rate, $\nabla F(\mathbf{X}; \xi) \in \mathbb{R}^{d \times n}$ represents stochastic gradients using local sampling data samples $\xi$, and mixing matrices are initially given. Representative dynamic graphs include the 1-peer exponential graph [43], 1-peer hypercube graph [31], 1-peer EquiDyn graph [32], and base-$(k+1)$ graph [35]. Conversely, static graphs, such as ring, grid, torus [24], and exponential graph [1] are available by setting $\tau = 1$.

Associated with (2), a convergence analysis is provided in [13]. Under the Assumptions 1-4 listed below, a convergence rate of DSGD is given:

**Assumption 1.** *There exists $L(>0)$ such that $\|\nabla f_i(\mathbf{a}) - \nabla f_i(\mathbf{b})\|_2 \leq L\|\mathbf{a} - \mathbf{b}\|_2$, $\forall \mathbf{a}, \mathbf{b} \in \mathbb{R}^d$.*

**Assumption 2.** *There exists $\sigma$ such that $\mathbb{E}_\xi \|\nabla f_i(\mathbf{x}) - \nabla F_i(\mathbf{x}; \xi)\|_2^2 \leq \sigma^2$, $\forall \mathbf{x} \in \mathbb{R}^d$.*

**Assumption 3.** *There exists $\zeta$ such that $\frac{1}{n} \sum_{i=1}^n \|\nabla f_i(\mathbf{x}) - \nabla f(\mathbf{x})\|_2^2 \leq \zeta^2$, $\forall \mathbf{x} \in \mathbb{R}^d$.*

**Assumption 4** (Expected spectral gap). *There exists $p \in (0, 1]$ and $\tau (\geq 1)$ such that: $\mathbb{E}_W \|\mathbf{X}\mathbf{W} - \overline{\mathbf{X}}\|_F^2 \leq (1-p)\|\mathbf{X} - \overline{\mathbf{X}}\|_F^2$, $\forall \mathbf{X} \in \mathbb{R}^{d \times n}$, where $\mathbf{W} = \boldsymbol{W}^{(0)} \cdots \boldsymbol{W}^{(\tau-1)}$ and $\overline{\mathbf{X}} = \frac{1}{n}\mathbf{X}\mathbf{1}_n\mathbf{1}_n^\top$.*

**Theorem 1** (Complexity estimates of DSGD [13]). *Suppose that Assumptions 1-4 hold. For any target accuracy $\epsilon (> 0)$, there exists a learning rate (potentially depending on $\epsilon$) such that the accuracy $\frac{1}{R+1} \sum_{r=0}^R \mathbb{E}\|\nabla f(\overline{\mathbf{x}}^{(r)})\|_2^2 \leq \epsilon$ can be reached after*

$$R = \mathcal{O}\left(\frac{\sigma^2}{n\epsilon^2} + \frac{\zeta\tau + \sigma\sqrt{p\tau}}{p\epsilon^{3/2}} + \frac{\tau}{p\epsilon}\right) \cdot Lf_0$$

*communication rounds, where $f_0 = f(\overline{\mathbf{x}}^{(0)}) - f^*$ and $f^*$ denotes the minimum of $f$.*

Theorem 1 highlights that maximizing the expected spectral gap ($p \to 1$), and shortening the periodic interval $\tau$ are crucial for communication-efficient decentralized learning. To this end, various graphs have been explored, one of which is introduced in the next subsection.

## 2.2 1-peer exponential graph

As a baseline dynamic graph, the 1-peer exponential graph is introduced. It can be characterized by a sequence of circulant mixing matrices:

**Definition 1** (Graph consisting of circulant mixing matrices). *Consider a graph defined by $\tau$ circulant mixing matrices, denoted by $\{\boldsymbol{W}^{(0)}, \ldots, \boldsymbol{W}^{(\tau-1)}\} \in \mathbb{R}_{\geq 0}^{(n \times n) \times \tau}$ for each $\ell = 0, 1, \ldots, \tau-1$, as*

$$\boldsymbol{W}^{(\ell)} = \mathrm{circ}(h^{(\ell)}(0), h^{(\ell)}(1), \ldots, h^{(\ell)}(n-1)) = \begin{bmatrix} h^{(\ell)}(0) & h^{(\ell)}(n-1) & \cdots & h^{(\ell)}(1) \\ h^{(\ell)}(1) & h^{(\ell)}(0) & \cdots & h^{(\ell)}(2) \\ \vdots & \ddots & \ddots & \vdots \\ h^{(\ell)}(n-1) & h^{(\ell)}(n-2) & \cdots & h^{(\ell)}(0) \end{bmatrix}, \quad (3)$$

*where $\{h^{(\ell)}(0), \ldots, h^{(\ell)}(n-1)\}$ satisfies $\sum_{i=0}^{n-1} h^{(\ell)}(i) = 1$ to be doubly stochastic.*

Unless the graph consists of circulant mixing matrices, each node has the same number of connections (uniform degree). The 1-peer exponential graph decomposes the connections of the exponential graph into $\tau = \lceil \log_2(n) \rceil$ mixing matrices, enabling 1-peer communication as follows:

**Definition 2** (1-peer exponential graph [43]). *The 1-peer exponential graph is a dynamic graph composed of $\tau = \lceil \log_2(n) \rceil$ mixing matrices ($\ell \in \{0, 1, \ldots, \tau-1\}$) in the form (3) with:*

$$h^{(\ell)}(i) = \begin{cases} \frac{1}{2} & i = 0, 2^\ell \\ 0 & \text{otherwise} \end{cases}.$$

**Definition 3** (Finite-time convergence). *A dynamic graph composed of $\tau$ mixing matrices $\{\boldsymbol{W}^{(0)}, \ldots, \boldsymbol{W}^{(\tau-1)}\}$ is considered a $\tau$-finite time convergent graph if it satisfies: $\boldsymbol{W}^{(0)} \cdots \boldsymbol{W}^{(\tau-1)} = \frac{1}{n}\mathbf{1}_n\mathbf{1}_n^\top$. This indicates that the expected spectral gap in Assumption 4 becomes one $(p=1)$.*

In [43], it is proven that the 1-peer exponential graph achieves finite-time convergence for limited $n$ (a power of 2). An additional key property is the commutativity of $\tau$ mixing matrices. Specifically, their product–regardless of the permutation of the order–consistently satisfies the following:

**Definition 4** (Commutativity of mixing matrices). *For any permutation $\rho$ of indices $\{0, \ldots, \tau-1\}$, the product $\boldsymbol{W}^{(\rho(0))} \cdots \boldsymbol{W}^{(\rho(\tau-1))}$ is equal to $\boldsymbol{W}^{(0)} \cdots \boldsymbol{W}^{(\tau-1)}$.*

As shown in Theorem 1, maximizing the expected spectral gap $(p \to 1)$ and maintaining a small periodic interval are effective for communication-efficient decentralized learning. However, the 1-peer exponential graph only achieves finite-time convergence for a limited $n$ such that $n$ is a power of two. Moreover, its efficiency across any $n$ and the commutativity of $\tau$ mixing matrices remain unexplored, warranting further investigation.

# 3 Revising 1-peer exponential graph to explore its underlying principles

The underlying principles of the 1-peer exponential graph are explored. The previous study [43] established its finite-time convergence property for specific $n$. However, its efficiency across any $n$ and the commutativity of $\tau$ mixing matrices remain unexplored. Further investigation may lead to more communication-efficient decentralized learning methods. Section 3.1 presents the $k$-peer exponential graph, a generalized form of the 1-peer exponential graph supporting $k$-peer communications. In Sec. 3.2, we explore the conditions to satisfy finite-time convergence and the underlying principles of the $k$-peer exponential graph to explain its efficiency and commutativity.

## 3.1 $k$-peer exponential graph

The fact that the 1-peer exponential graph achieves finite-time convergence for specific $n$ can be extended to allow $k$-peer communication for specific $n$ (a power of $k+1$). To illustrate this, we first introduce the $k$-peer exponential graph, defined as follows:

**Definition 5** ($k$-peer exponential graph). *Given $(n, k)$ and $\tau = \lfloor \log_{k+1}(n) \rfloor$[1], the $k$-peer exponential graph consists of $\tau$ circulant mixing matrices ($\ell \in \{0, 1, \ldots, \tau-1\}$), defined in the form of* (3) *with:*

$$h^{(\ell)}(i) = \begin{cases} \frac{1}{k+1} & i = 0, (k+1)^\ell, 2(k+1)^\ell, \ldots, k(k+1)^\ell \\ 0 & \text{otherwise} \end{cases}. \tag{4}$$

An example is illustrated in Fig. 1(a); however, this does not achieve finite-time convergence since $(n, k) = (15, 2)$. A theoretical analysis of the $k$-peer exponential graph, including its expected spectral gap and commutativity, is provided in the following subsection and Appendix A.

## 3.2 Principles underlying $k$-peer exponential graph

This subsection presents two of our key findings: i) a necessary condition for a graph of $\tau$ circulant mixing matrices to achieve finite-time convergence, and ii) a description of the efficiency of the $k$-peer exponential graph, a generalization of the 1-peer exponential graph, across any $n$.

**i) Condition to be a finite-time convergent graph.** To establish a condition for a graph composed of $\tau$ circulant mixing matrices to achieve finite-time convergence, we first analyze the target property of mixing matrix product; namely, the complete graph to connect all nodes: $\mathbf{W}_{\text{comp}} = \frac{1}{n}\mathbf{1}_n\mathbf{1}_n^\top$. Since $\mathbf{W}_{\text{comp}}$ is also a circulant mixing matrix, its eigenvalue decomposition is facilitated by the DFT [7] as $\mathbf{W}_{\text{comp}} = \boldsymbol{D}^\top \text{diag}([1, 0, \ldots, 0])\boldsymbol{D}$, where $\boldsymbol{D} \in \mathbb{C}^{n \times n}$ denotes the DFT matrix. This reveals that averaging $n$ local parameters is equivalent to isolating the DC component (0-th frequency

---

[1]In the 1-peer exponential graph in Sec. 2.2, the ceiling function was used as $\tau = \lceil \log_2(n) \rceil$; however, we use the flooring function to derive expected spectral gap for any $n$ in Sec. 3.2.

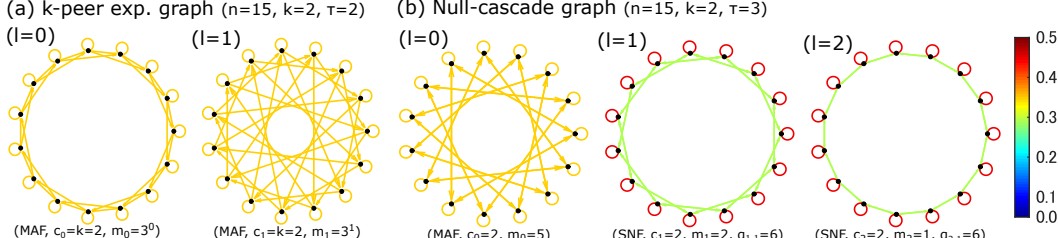

(a) k-peer exp. graph (n=15, k=2, τ=2)   (b) Null-cascade graph (n=15, k=2, τ=3)

(l=0)            (l=1)            (l=0)            (l=1)            (l=2)

(MAF, $c_0$=k=2, $m_0$=$3^0$)   (MAF, $c_1$=k=2, $m_1$=$3^1$)   (MAF, $c_0$=2, $m_0$=5)   (SNF, $c_1$=2, $m_1$=2, $q_{1,1}$=6)   (SNF, $c_2$=2, $m_2$=1, $q_{2,1}$=6)

Figure 1: Proposed graphs: (a) $k$-peer exponential graph with $(n, k) = (15, 2)$, which does not achieve finite-time convergence. (b) Null-cascade graph with $(n, k) = (15, 2)$, which achieves finite-time convergence. Each graph consists of $n = 15$ nodes, depicted as black dots arranged in a circle and interconnected by lines. The colors of these lines indicate the mixing weights between the nodes in $\boldsymbol{W}^{(\ell)}$. In (a), uniform weights ($W_{ij}^{(\ell)} = 1/3$) are employed (depicted in yellow), whereas in (b), $n = 15$ nodes when $\ell = 1, 2$ are interconnected with lines colored differently (red and green), indicating non-uniform weights specified by SNFs.

component) while nullifying all non-zero frequencies ($i_\omega \in \{1, \dots, n-1\}$). Similarly, we analyze the product of $\tau$ circulant mixing matrices through eigenvalue decomposition:

$$\boldsymbol{W}^{(0)} \cdots \boldsymbol{W}^{(\tau-1)} = (\boldsymbol{D}^\top \boldsymbol{H}^{(0)} \boldsymbol{D}) \cdots (\boldsymbol{D}^\top \boldsymbol{H}^{(\tau-1)} \boldsymbol{D}) = \boldsymbol{D}^\top \boldsymbol{H}^{(0)} \cdots \boldsymbol{H}^{(\tau-1)} \boldsymbol{D}, \qquad (5)$$

where $\boldsymbol{H}^{(\ell)} = \text{diag}([H^{(\ell)}(0), \dots, H^{(\ell)}(n-1)])$ denotes the eigenvalue matrix, of which the diagonal elements are in fact the frequency responses of $\{h^{(\ell)}(0), \dots, h^{(\ell)}(n-1)\}$: $H^{(\ell)}(i_\omega) = \sum_{i=0}^{n-1} h^{(\ell)}(i) e^{-2\pi \text{j} i_\omega i / n}, (i_\omega \in \{0, 1, \dots, n-1\})$, where the DC component is preserved as $H^{(\tau-1)}(0) \cdots H^{(0)}(0) = 1$ due to the use of the doubly stochastic sequence $\sum_{i=0}^{n-1} h^{(\ell)}(i) = 1$. Additionally, since the results of (5) remain unchanged even when the order of $\tau$ circulant matrices is permuted, dynamic graphs defined in Definition 1 exhibit commutativity for any $n$.

From (5), the product of $\tau$ circulant mixing matrices can be interpreted as a cascade of $\tau$ (spatial) filters that maintain the DC component at 1. Thus, a condition for finite-time convergence is given by the following equation, which ensures that all non-zero frequency components are nullified through the cascade of $\tau$ circulant mixing matrices: $H^{(\tau-1)}(i_\omega) \cdots H^{(0)}(i_\omega) = 0, (i_\omega = 1, \dots, n-1)$. If at least one of the $\tau$ circulant mixing matrices has a zero response, $H^{(\ell)}(i_\omega) = 0$ for every non-zero frequency $i_\omega$. Let $\mathcal{N}^{(\ell)}$ denote the set of null frequency indices for the $\ell$-th mixing matrix $\boldsymbol{W}^{(\ell)}$. Thus, the condition for finite-time convergence can be equivalently expressed as follows: the cascade of $\tau$ circulant mixing matrices must form nulls at all non-zero frequency indices, ensuring:

$$\bigcup_{\ell=0}^{\tau-1} \mathcal{N}^{(\ell)} = \{1, 2, \dots, n-1\}. \qquad (6)$$

**ii) Exploring principles underlying $k$-peer exponential graph.** We next demonstrate the $k$-peer exponential graph, with $k \geq 1$, effectively increases the expected spectral gap $p$ across any $n$.

We notice that (4) is an MAF with $(k+1)$ uniform weights and its expansions.[2] This observation motivates us to analyze the efficiencies of the $k$-peer exponential graph by employing DSP techniques:

**Definition 6** (Frequency response of Moving Average Filter (MAF) [29]). *For any $c_\ell \in \mathbb{N}$, frequency response of an MAF with $(c_\ell + 1)$-length is given by*

$$H(e^{\text{j}\omega}) = \frac{1}{c_\ell + 1} \frac{\sin(\omega(c_\ell + 1)/2)}{\sin(\omega/2)} e^{-\text{j}\omega c_\ell / 2},$$

*where $\omega \in [0, 2\pi)$ denotes the angular frequency.*

**Definition 7** (Expanded sequence). *Given a sequence $h(0), \dots, h(c_\ell)$ and expansion order $m_\ell \in \mathbb{N}$ satisfying $(c_\ell + 1) m_\ell \leq n$, the expanded sequence $h_{\uparrow m_l}(0), \dots, h_{\uparrow m_\ell}(n-1)$ is formed by separating each element of the original sequence by $m_\ell - 1$ zeros:*

$$h_{\uparrow m_\ell}(i) = \begin{cases} h(j) & i = m_\ell j, \ j \in \{0, \dots, c_\ell\} \\ 0 & \text{otherwise} \end{cases}. \qquad (7)$$

---

[2]In the DSP field, "expansion" refers to the process of upsampling by inserting additional zeros between the original samples, thus lengthening the sequence for sampling rate conversion.

**Lemma 1** (Frequency response of expanded sequence [29])**.** *Given an original sequence $h(i)$ and its expansion $h_{\uparrow m_\ell}(i)$ as in (7), the frequency response of expanded sequence $H_{\uparrow m_\ell}(e^{j\omega})$ is represented as shifts in the frequency domain of the original sequence $H(e^{j\omega})$: $H_{\uparrow m_\ell}(e^{j\omega}) = H(e^{j\omega m_\ell})$.*

When the original sequence is an MAF of length $(c_\ell + 1)$, i.e., $h(0) = \cdots = h(c_\ell) = 1/(c_\ell+1)$, the frequency response of the $m_\ell$-order expended MAF can be derived by combining Definition 6 and Lemma 1, as follows:

$$H_{\uparrow m_\ell}(e^{j\omega}) = \frac{1}{c_\ell + 1} \frac{\sin(\omega m_\ell(c_\ell+1)/2)}{\sin(\omega m_\ell/2)} \exp^{-j\omega m_\ell c_\ell/2} . \tag{8}$$

Recall that the $k$-peer exponential graph (Definition 5) consists of $\tau = \lfloor \log_{k+1}(n) \rfloor$ circulant mixing matrices, where each matrix $\boldsymbol{W}^{(\ell)}$ is characterized by an MAF with a fixed communication order $k$ and exponential expansion order $m_\ell = (k+1)^\ell$. Using the DSP techniques discussed previously and the fact that the angular frequency $\omega$ and frequency index $i_\omega$ are related by $\omega = 2\pi i_\omega/n$, the expected spectral gap regarding the $k$-peer exponential graph is given as follows:

**Theorem 2** (Expected spectral gap of $k$-peer exponential graph for any $(n,k)$)**.** *The expected spectral gap of $k$-peer exponential graph for any $(n,k)$ with $\tau = \lfloor \log_{k+1}(n) \rfloor$ is given by*

$$p = \begin{cases} 1 & n \text{ is a power of } k+1 \\ 1 - \max\limits_{i \in \{1,...,n-1\}} \left| \frac{1}{(k+1)^\tau} \frac{\sin(\pi i(k+1)^\tau/n)}{\sin(\pi i/n)} \right| & \text{otherwise} \end{cases} .$$

The proof is provided in Appendix A. It shows that the $k$-peer exponential graph achieves finite-time convergence when $n$ is a power of $k + 1$. Moreover, the expected spectral gap $p$ is increased to the order of $1 - \mathcal{O}(1/(k+1)^\tau)$. This result theoretically supports the efficiency of the $k$-peer exponential graph across any $n$. In particular, $p$ is efficiently maximized when $n$ is equal to or slightly exceeds a power of $(k + 1)$, as illustrated in Appendix F. Furthermore, the convergence rate of DSGD using this graph is obtained by substituting $p$ in Theorem 2 into Theorem 1, as summarized in Appendix E.

As a secondary outcome of our exploration in this subsection, we present a simplified algorithm for constructing the base-$(k+1)$ graph [35], which achieves finite-time convergence for any $(n,k)$ configuration. The details of this are provided in Alg. 5 in Appendix C. However, as noted earlier, the base-$(k+1)$ graph does not support the commutative dynamic graph we are aiming to achieve. In the next subsection, we further extend the $k$-peer exponential graph to achieve finite-time convergence for any $n$ while maintaining commutativity.

## 4 Null-cascade graph

We propose the null-cascade graph designed to achieve finite-time convergence for any $n$ while maintaining commutativity. Recall that the $k$-peer exponential graph maintains commutativity but only achieves finite-time convergence for specific $n$ (a power of $k+1$). This limitation stems from the fixed communication order $k$. Assuming that $k \geq 2$ is available, our strategy involves the following steps: (Step 1) using MAFs with dynamic communication orders $c_\ell(\leq k)$ and dynamic expansion orders $m_\ell$ to efficiently nullify as many non-zero frequency components as possible, and (Step 2) incorporating not only MAFs with uniform weights but also SNFs with non-uniform weights, which are defined later, to nullify remaining frequencies, as detailed in Alg. 1.

To implement this strategy, we first factorize $n$ as $n = \prod_{i=1}^{\lambda}(\nu_i + 1)$, where $\nu_1 \leq ... \leq \nu_\kappa \leq k < \nu_{\kappa+1} \leq ... \leq \nu_\lambda$, with each $\nu_i \in \mathbb{N}$ (line 2 in Alg. 1). For Step 1, the null frequencies of expanded MAFs with communication order $c_\ell \in \mathbb{N}$ and expansion order $m_\ell \in \mathbb{N}$ are analytically derived from (8):

**Lemma 2** (Null frequencies of expanded MAF)**.** *Suppose an MAF with communication order $c_\ell$ conforms to Definition 6. Its expansion with order $m_\ell$ forms nulls at multiples of $\frac{n}{(c_\ell+1)m_\ell}$, excluding multiples of $\frac{n}{m_\ell}$, within $[0, n)$.*

The derivation is shown in Appendix B. In (6), the frequencies to be nullified are identified as non-zero frequencies $i_\omega \in \{1, ..., n-1\}$, equivalently represented as multiples of $b = 1$ within the interval $[0, n)$ (line 1). To effectively nullify these frequencies, it is optimal to select $m_\ell$ such that $\frac{n}{(c_\ell+1)m_\ell} = b$, yielding in $m_\ell = \frac{n}{(c_\ell+1)b}$ (line 6). To ensure $m_\ell$ be a natural number, $(c_\ell + 1)$ must be

---

**Algorithm 1** Null-cascade graph

---

1: ▷ Given $n$, $k(\geq 2)$, $\ell = 0$, $b = 1$
2: ▷ Factorization of $n$: $n = \prod_{i=1}^{\lambda}(\nu_i + 1)$, where $\nu_1 \leq \cdots \leq \nu_\kappa \leq k < \nu_{\kappa+1} \leq \cdots \leq \nu_\lambda$, where $\nu_i \in \mathbb{N}$
3: ▷ (Step 1) Null forming using expanded MAFs
4: **for** $i = \kappa, \ldots, 1$ **do**
5: $\quad c_\ell = \nu_i$
6: $\quad m_\ell = \frac{n}{(c_\ell + 1)b}$
7: $\quad h^{(\ell)}(j) = \begin{cases} \frac{1}{c_\ell + 1} & (j = 0, \ldots, c_\ell) \\ 0 & (\text{otherwise}) \end{cases}$
8: $\quad \boldsymbol{W}^{(\ell)} = \mathrm{circ}([h_{\uparrow m_\ell}^{(\ell)}(0), \ldots, h_{\uparrow m_\ell}^{(\ell)}(n-1)])$
9: $\quad \ell = \ell + 1$ $\quad$ /* mixing matrix index increment */
10: $\quad b = b(\nu_i + 1)$ $\quad$ /*base of frequencies to be nullified*/
11: **end for**
12: ▷ (Step 2) Null forming using expanded SNFs
13: **for** $i = \lambda, \ldots, \kappa + 1$ **do**
14: $\quad \{\widetilde{\gamma}_j(0), \ldots, \widetilde{\gamma}_j(n-1)\}_{j \in \{1, \ldots, \tau_{\mathrm{snf}}(\nu_i)\}} = \mathrm{SNF}(n, k, \nu_i, b)$ $\quad$ /* Alg. 3 in Appendix B */
15: $\quad$ **for** $j = 1, \ldots, \tau_{\mathrm{snf}}(\nu_i)$ **do**
16: $\quad\quad \boldsymbol{W}^{(\ell)} = \mathrm{circ}([\widetilde{\gamma}_j(0), \ldots, \widetilde{\gamma}_j(n-1)])$
17: $\quad\quad \ell = \ell + 1$ $\quad$ /* mixing matrix index increment */
18: $\quad$ **end for**
19: $\quad b = b(\nu_i + 1)$ $\quad$ /*base of frequencies to be nullified*/
20: **end for**

---

a factor of $n/b$; namely, $c_\ell = \nu_i$ ($i \in \{1, ..., \kappa\}$) (line 5). In our implementation, as detailed in Alg. 1, we sequentially select from $\nu_\kappa$ down to $\nu_1$ as $c_\ell$ ($\ell \in \{0, ..., \kappa-1\}$) (line 4), because many nulls can be efficiently formed by choosing the largest possible values for both $c_\ell$ and $m_\ell$. After generating an expanded MAF, circulant mixing matrix $\boldsymbol{W}^{(\ell)}$ is computed (lines 7-8), and the frequencies still to be nullified can be updated to multiples of $b \leftarrow b(\nu_i + 1)$ (line 10) because frequencies associated with a factor $\nu_i$ are eliminated. This process can be incrementally repeated ($\ell \in \{0, ..., \kappa-1\}$) to nullify the frequencies associated with $\{\nu_\kappa, ..., \nu_1\}$ and reaches $b = \prod_{i=1}^{\kappa}(\nu_i + 1)$ in Step 1 (lines 4-11) in Alg. 1.

However, this approach is not feasible when factors $\nu_i > k$ exist, as $c_\ell \leq k$. To address the challenge of nullifying the remaining frequencies for the factors $\{\nu_{\kappa+1}, ..., \nu_\lambda\}$ that exceed $k$, we introduce additional filters, referred to as SNFs, designed to cascade and achieve the complete nullification of non-zero frequencies in Step 2.

**Definition 8** (Steerable Nulling Filter (SNF)). *When $k \geq 2$, $c_\ell$-peer communication with $c_\ell \leq k$ is available. An SNF with a length of $c_\ell + 1$ can then be employed to form nulls at specific target frequencies. We derive the SNF by solving a polynomial specifically designed to form these nulls. By setting $c_\ell$ to an even number $c_\ell = 2\lfloor k/2 \rfloor$, the polynomial is expressed with roots corresponding to frequency indices $\{q_{\ell,1}, \ldots, q_{\ell,c_\ell/2}\}$ and their conjugate frequency indices $\{q_{\ell,1}^*, \ldots, q_{\ell,c_\ell/2}^*\}$, where $q_{\ell,\psi}^* = n - q_{\ell,\psi}$. The polynomial is expressed as*

$$\prod_{\psi=1}^{c_\ell/2}(x - e^{\mathrm{j}2\pi q_{\ell,\psi}/n})(x - e^{\mathrm{j}2\pi q_{\ell,\psi}^*/n}). \tag{9}$$

*This leads to a $c_\ell$-order polynomial equation of the form: $\gamma(0)x^{c_\ell} + \cdots + \gamma(c_\ell - 1)x + \gamma(c_\ell)$. To ensure the doubly stochastic property in a $(c_\ell + 1)$-length filter sequence, the normalized filter sequence is computed as $\overline{\gamma}(i) = \gamma(i) / \sum_{i=0}^{c_\ell} \gamma(i)$ for $i \in \{0, \ldots, c_\ell\}$. This will result in a filter sequence consisting of non-uniform weights.*

In (9), we set pairwise nulls in a conjugate relationship to ensure that SNFs are real numbers. This configuration requires $k \geq 2$. As stipulated in Definition 8, the SNF creates $c_\ell = 2\lfloor k/2 \rfloor$ nulls at $q_{\ell,\psi}$ and $q_{\ell,\psi}^*$ ($\psi \in \{1, ..., c_\ell/2\}$). The expanded SNF with order $m_\ell$ leads the following null formation:

**Lemma 3** (Null frequencies of expanded SNF). *Suppose an SNF with communication order $c_\ell$ conforms to Definition 8. Its expansion with order $m_\ell$ forms null frequencies at $\frac{q_{\ell,\psi} + \phi n}{m_\ell}$ and $\frac{q_{\ell,\psi}^* + \phi n}{m_\ell}$, where $\psi \in \{1, ..., c_\ell/2\}$, and $\phi \in \{0, ..., m_\ell - 1\}$.*

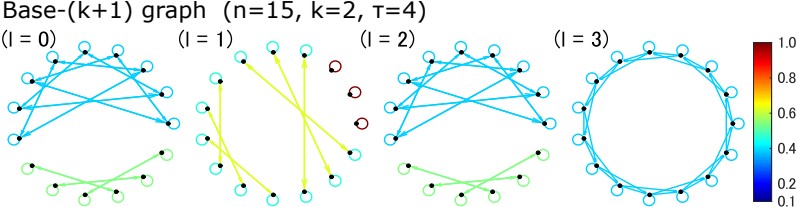

Figure 2: Base-$(k+1)$ graph with $(n, k) = (15, 2)$, where nodes are depicted as black dots arranged in a circle and interconnected by lines. The colors of these lines indicate the mixing weights between nodes in $\boldsymbol{W}^{(\ell)}$.

The derivation is provided in Appendix B. To nullify the remaining frequencies associated with $\{\nu_{\kappa+1}, ..., \nu_\lambda\}$, selection of $(m_\ell, q_{\ell,\psi})$ is required in the SNF. According to Lemma 3, $(m_\ell, q_{\ell,\psi})$ must be selected such that $q_{\ell,\psi}/m_\ell$ corresponds to the remaining frequency indices (multiples of $b$). However, since this cannot be uniquely determined, an algorithm to select $(m_\ell, q_{\ell,\psi})$ is specified in Alg. 3 and 4 in Appendix B, ensuring the positive-definite property of the SNFs. The process of sequentially generating SNFs (line 14) and corresponding circulant matrices (line 16) to nullify the frequencies associated with $\{\nu_\lambda, ..., \nu_{\kappa+1}\}$ continues until $b$ reaches $n$, as outlined in Step 2 (lines 13-20) of Alg. 1, thereby ensuring that nullification at all non-zero frequencies in (6) is accomplished.

**Theorem 3** (Expected spectral gap of null-cascade graph)**.** *The expected spectral gap of the null-cascade graph, specified by Alg. 1, is $p = 1$ (i.e., finite-time convergence) for any $n$ with $k \geq 2$.*

**Discussion.** An example of the null-cascade graph with $(n, k) = (15, 2)$ is depicted in Fig. 1(b). Since $n$ can be factorized as $n = 15 = (4+1)(2+1)$, the factors are $\nu_1 = 2 (\leq k)$ and $\nu_2 = 4 (> k)$. For frequencies associated with $\nu_1$ are nullified using an MAF, while those associated with $\nu_2$ are nullified with two SNFs, resulting in a total of $\tau = 3$ mixing matrices. In Fig. 1(b), each matrix weights vary across each matrix $\ell \in \{0, 1, 2\}$, affecting the color of lines connecting the nodes. For $\ell = 0$, an MAF with equal weights ($c_0 = 2$) is employed, coloring all connecting lines yellow. Conversely, for $\ell = 1, 2$, non-uniform weights ($c_1 = 2, c_2 = 2$) specified by SNFs are employed, resulting in lines colored differently (red and green). From Theorem 3, this achieves finite-time convergence.

Next, we present a brief comparison with the base-$(k+1)$ graph, which also achieves finite-time convergence for any $n$. Figure 2 illustrates the base-$(k+1)$ graph for $(n, k) = (15, 2)$. Comparing this to Fig. 1(b), which depicts the null-cascade graph under the same $(n, k)$ configuration, several differences emerge. Primarily, the null-cascade graph maintains a uniform degree across $n$ nodes (every node has the same number of connections), whereas the base-$(k+1)$ graph does not, leading to unbalanced parameter mixing within each subgroup and occasional parameter exchanges among subgroups ($\ell \in \{0, 1, 2\}$). This uniformity in local degree within the null-cascade graph supports stable parameter training. Moreover, the periodic interval $\tau$, which is required for finite-time convergence, differs between null-cascade graph and base-$(k+1)$ graph. For $(n, k) = (15, 2)$, $\tau$ in the null-cascade graph ($\tau = 3$) is smaller compared to that in the base-$(k+1)$ graph ($\tau = 4$). However, this is not always the case as outlined in Appendix F. Including large prime numbers $\nu_i$ in the factorization of $n$ leads to an increase in $\tau$ for the null-cascade graph, since $\tau \approx \kappa + \sum_{i=\kappa+1}^{\lambda} \nu_i/(2\lfloor k/2 \rfloor)$, as shown in Table 1. While further discussion on the possibility of redundant counting of $\tau$ can be found in Appendix F, to demonstrate that the impact of this increase in $\tau$ is not significant, the next section includes experiments where $n$ is a large prime number.

## 5 Numerical experiments

To show the effectiveness of the proposed graphs, we conducted two experimental tests. In Sec. 5.1, we examine consensus errors to asses the fundamental properties of the graphs. In Sec. 5.2, we evaluate the performance of decentralized learning of DNN models by combining DSGD with graphs.

### 5.1 Test 1: Consensus error investigation

**Comparison graphs.** We used three network configurations[3]: $(n, k) = (15, 2), (17, 2)$, and $(30, 2)$, all incorporating SNFs in mixing matrices, where $n = 17$ is a large prime number. For comparison purposes, we employed both static and dynamic graphs, considering a fair communication degree

---

[3]Other $(n, k)$-configurations are also tested in Appendix G.

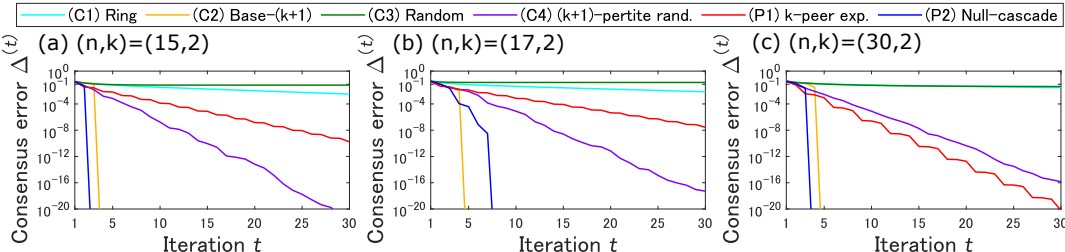

Figure 3: Comparison of consensus error $\Delta^{(t)}$ to be minimized to zero for three $(n, k)$-configurations.

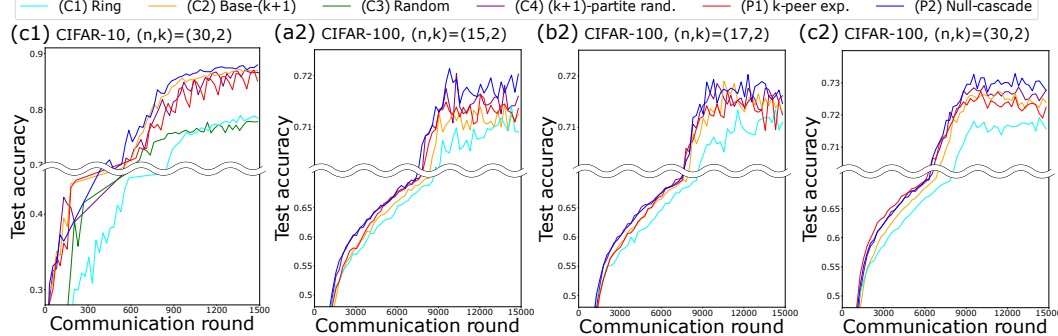

Figure 4: Convergence curves using test accuracy of global parameters for four configurations.

Table 2: Comparison of the highest test accuracy using global parameters.

| Dataset (heterogeneity level) | CIFAR-10 ($\alpha = 0.1$) | | | | | | CIFAR-100 ($\alpha = 0.1$) | | | | | |
|---|---|---|---|---|---|---|---|---|---|---|---|---|
| $(n, k)$-configuration | (a1) $(15, 2)$ | | (b1) $(17, 2)$ | | (c1) $(30, 2)$ | | (a2) $(15, 2)$ | | (b2) $(17, 2)$ | | (c2) $(30, 2)$ | |
| Tested graphs | $\tau$ | test acc. | $\tau$ | test acc. | $\tau$ | test acc. | $\tau$ | test acc. | $\tau$ | test acc. | $\tau$ | test acc. |
| (C1) Ring | 1 | 0.8210 | 1 | 0.8773 | 1 | 0.7902 | 1 | 0.7140 | 1 | 0.7134 | 1 | 0.7192 |
| (C2) Base-$(k+1)$ | 4 | 0.8469 | 5 | 0.8727 | 5 | 0.8712 | 4 | 0.7145 | 5 | 0.7190 | 5 | 0.7275 |
| (C3) Random | 1 | 0.5815 | 1 | 0.4249 | 1 | 0.7828 | 1 | 0.1781 | 1 | 0.0835 | 1 | 0.4604 |
| (C4) $(k+1)$-partite random match | – | 0.8556 | – | 0.8711 | – | 0.8691 | – | 0.7204 | – | 0.7185 | – | 0.7291 |
| (P1) $k$-peer exponential | 2 | 0.8371 | 2 | 0.8609 | 3 | 0.8709 | 2 | 0.7163 | 2 | 0.7170 | 3 | 0.7273 |
| (P2) Null-cascade | 3 | **0.8660** | 8 | **0.8817** | 4 | **0.8784** | 3 | **0.7213** | 8 | **0.7203** | 4 | **0.7329** |

($k = 2$). The static graphs consist of a (C1) ring graph, and a (C3) random graph modeled as in Erdos-Rényi random graph [5], adjusted to ensure $k$ for each configuration. The dynamic graphs include a (C2) base-$(k+1)$ graph and a (C4) $(k+1)$-partite random match graph, which is a generalization of a bipartite graph, used in [43] to allow $k$-peer communication. The proposed graphs consist of (P1) $k$-peer exponential graph and (P2) null-cascade graph. Further details are provided in Appendix G.

**Evaluation metric.**   We computed the consensus error at each iteration $t$ as $\Delta^{(t)} = \frac{1}{n} \sum_{i=1}^{n} (x_i^{(t)} - \overline{x})^2$, where $x_i^{(t)}$ denotes local parameter ($d = 1$) at the $i$-th node, and $\overline{x}^{(t)} = \frac{1}{n} \sum_{i=1}^{n} x_i^{(t)}$ is global parameter. A desirable graph leads to a small consensus error in fewer iterations. For each $i$, $x_i^{(0)}$ was drawn from Gaussian distribution $\mathcal{N}(0, 1)$, and performed 100 independent trials for each graph.

**Experimental results.**   Figure 3 presents the consensus error for each $(n, k)$-configuration. As (C2) base-$(k+1)$ graph and (P2) null-cascade graph exhibit finite-time convergence, their consensus errors drop to zero after $\tau$ iterations. As discussed in the previous section, $\tau$ for null-cascade graph tends to be small when the factorization of $n$ does not include large prime numbers. As a result, null-cascade graph performs well in Figures 3(a) and (c). In contrast, (C2) base-$(k+1)$ graph is preferable in Figure 3(b) because $n = 17$ is a large prime number, leading to a small $\tau$. Following these two graphs in performance are (C4) $(k+1)$-partite random match graph and (P1) $k$-peer exponential graph. When $n$ is equal to or slightly exceeds a power of $(k+1)$, as in (c) $(n, k) = (30, 2)$, the consensus error of the $k$-peer exponential graph becomes small. This observation aligns with Theorem 2.

## 5.2   Test 2: Decentralized learning of DNN models

**Dataset and model.**   We investigated decentralized learning performance of each graph using image classification benchmark tests using CIFAR-10 and CIFAR-100 with ResNet18 [8]. The

batch normalization layers were replaced by group normalization layers [41] to account for potential data heterogeneity in local datasets. To experimentally assess robustness against data heterogeneity, the training dataset was divided into $n$ nodes to follow a Dirichlet distribution with concentration hyperparameter $\alpha$ [38]. We set $\alpha = 0.1$, representing a scenario with strong data heterogeneity. The data distributions across the $n$ nodes are illustrated in Appendix G.

**Update rules.** The DSGD algorithm used in our experiments is outlined in Alg. 6 in Appendix G, which extends the update rules (2) to support multiple local parameter updates. The graphs used[3] are identical to those in Test 1. The learning rate $\eta$ was pre-tuned, as provided in Appendix G. As the evaluation metric, we used the test accuracy of the global parameter $\overline{\mathbf{x}}^{(r)} = \frac{1}{n} \sum_{i=1}^{n} \mathbf{x}_i^{(r)}$.

**Experimental results.** Figure 4 illustrates the convergence curves using test accuracy of the global parameter under several configurations in a strongly heterogeneous setting ($\alpha = 0.1$). The highest test accuracy during the training is summarized in Table 2. Among the tested $(n, k)$-configurations, the newly proposed (P2) null-cascade graph consistently achieved the highest test accuracy across most settings, including cases where $n$ is a large prime number. As discussed in Sec. 4, both (P2) null-cascade graph and (C2) base-$(k+1)$ graph ensure finite-time convergence. However, they differ in degree uniformity and the periodic interval $\tau$. When the gap in $\tau$ is small, the non-uniform degree in (C2) base-$(k+1)$ graph leads to unbalanced model mixing, which negatively affects training performance. In contrast, (C4) $(k+1)$-partite random match graph and (P1) $k$-peer exponential graph exhibit performance comparable to (C2) base-$(k+1)$ graph, despite showing clear differences in consensus error in Test 1. This can be attributed to the fact that the degree of (C4) remains statistically uniform, while (P1) ensures strict degree uniformity. Although (C1) ring also has uniform degree, its performance is limited by a small spectral gap. Our empirical results highlight that the proposed dynamic graphs, particularly the null-cascade graph, enhance decentralized learning under heterogeneous data settings.

## 6  Conclusion

We proposed two dynamic graphs—the $k$-peer exponential graph and the null-cascade graph—as a result of revisiting the 1-peer exponential graph. This reexamination revealed two key findings: (i) finite-time convergence can be achieved through a cascade of null formation at all non-zero frequencies using $\tau$ circulant matrices, and (ii) the expected spectral gap remains large for any $n$ in the $k$-peer exponential graph with $k \geq 1$. Leveraging these discoveries, the null-cascade graph is constructed to ensure finite-time convergence and commutativity for any $n$; however, this requires $k \geq 2$. Numerical experiments on image classification tasks demonstrated that the null-cascade graph achieved the highest test accuracy compared to conventional graphs across most test settings.

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

# A   Technical details for the $k$-peer exponential graph in Sec. 3

Technical proofs for Sec. 3 are summarized in Sec. A.1 and A.2. After introducing preliminaries regarding the calculation of null frequencies using circulant mixing matrices in Sec. A.3, several examples of the $k$-peer exponential graph are illustrated in Sec. A.4.

## A.1   Proof of Lemma 1

We investigated the frequency response of expanded sequences $h_{\uparrow m}(i)$. It is equivalently reformulated as $h_{\uparrow m}(i) = \sum_{s=-\infty}^{\infty} h(s)\delta(i-ms)$ using original sequence $h(i)$. Hence, applying DFT results in

$$
\begin{aligned}
H_{\uparrow m}(e^{\mathrm{j}\omega}) &= \sum_{i=-\infty}^{\infty} h_{\uparrow m}(i)e^{-\mathrm{j}\omega i} \\
&= \sum_{i=-\infty}^{\infty}\left[\sum_{s=-\infty}^{\infty} h(s)\delta(i-ms)\right]e^{-\mathrm{j}\omega i} \\
&= \sum_{s=-\infty}^{\infty} h(s)e^{-\mathrm{j}\omega m s} \\
&= H\left(e^{\mathrm{j}\omega m}\right).
\end{aligned}
$$

This indicates that the frequency response of the expanded sequence can be represented by shifting the frequency response of the original sequence $H(e^{\mathrm{j}\omega})$.

## A.2   Proof of Theorem 2

We computed the expected consensus rate of the $k$-peer exponential graph. As described in Sec. 3, the $k$-peer exponential graph (4) consists of $\tau = \lceil \log_{k+1}(n) \rceil$ circulant mixing matrices $\boldsymbol{W}^{(\ell)}$ for $\ell \in \{0, 1, \ldots, \tau - 1\}$. It can be decomposed by using DFT matrix $\boldsymbol{D}$ as

$$
\boldsymbol{W}^{(\ell)} = \boldsymbol{D}^{\top} \boldsymbol{H}^{(\ell)}_{\uparrow m_\ell} \boldsymbol{D},
$$

where the eigenvalue matrix is $\boldsymbol{H}^{(\ell)}_{\uparrow m_\ell} = \mathrm{diag}([H^{(\ell)}_{\uparrow m_\ell}(0), \ldots, H^{(\ell)}_{\uparrow m_\ell}(n-1)])$, with

$$
H^{(\ell)}_{\uparrow m_\ell}(i_\omega) = \sum_{i=0}^{n-1} h^{(\ell)}_{\uparrow m_\ell}(i)e^{-2\pi \mathrm{j} i_\omega i/n}, \qquad i_\omega \in \{0, 1, \ldots, n-1\}.
$$

In the $k$-peer exponential graph, we employed a filter sequence of length $(k+1)$, where each sequence is expanded to the order $m_\ell = (k+1)^\ell$. This results in the sequence $\{h^{(\ell)}_{\uparrow m_\ell}(0), \ldots, h^{(\ell)}_{\uparrow m_\ell}(n-1)\}$ for each $\ell \in \{0, 1, \ldots, \tau - 1\}$. As described in (8), the frequency response (amplitude) of the expanded $(k+1)$-length MAF with order $m_\ell$ is given by

$$
H^{(\ell)}_{\uparrow m_\ell}(e^{\mathrm{j}\omega}) = \frac{1}{k+1}\frac{\sin(\omega m_\ell(k+1)/2)}{\sin(\omega m_\ell/2)}\exp^{-\mathrm{j}\omega m_\ell k/2}, \qquad (\ell \in \{0, 1, \ldots, \tau-1\}).
$$

As noted in Sec. 3, the angular frequency $\omega$ and frequency indices $i_\omega \in \{0, \ldots, n-1\}$ are related by $\omega = 2\pi i_\omega/n$. By substituting it into the above equation, we can compute the frequency response at frequency indices $i_\omega \in \{0, \ldots, n-1\}$ as

$$
H^{(\ell)}_{\uparrow m_\ell}(i_\omega) = \frac{1}{k+1}\frac{\sin(\pi i_\omega m_\ell(k+1)/n)}{\sin(\pi i_\omega m_\ell/n)}\exp^{-\mathrm{j}\pi i_\omega m_\ell k/n}, \quad (\ell \in \{0, 1, \ldots, \tau-1\}, i_\omega \in \{0, \ldots, n-1\}),
$$

where the DC component holds $\left|H^{(\ell)}_{\uparrow m_\ell}(0)\right| = 1$, which is the maximum eigenvalue of $\boldsymbol{W}^{(\ell)}$, which is denoted by $\Lambda_1(\boldsymbol{W}^{(\ell)})$. The expected consensus rate corresponds to the second-largest eigenvalue of $\boldsymbol{W}^{(\ell)}$, which is denoted by $\Lambda_2(\boldsymbol{W}^{(\ell)})$, and can be computed by selecting the maximum argument

of nonzero frequency responses (amplitude), formulated as:

$$\Lambda_2(\boldsymbol{W}^{(\ell)}) = \max_{i_\omega \in \{1,\dots,n-1\}} \left| H_{\uparrow m_\ell}^{(\ell)}(i_\omega) \right|$$

$$= \max_{i_\omega \in \{1,\dots,n-1\}} \left| \frac{1}{k+1} \frac{\sin(\pi i_\omega m_\ell(k+1)/n)}{\sin(\pi i_\omega m_\ell/n)} \exp^{-\mathrm{j}\pi i_\omega m_\ell k/n} \right|$$

$$= \max_{i_\omega \in \{1,\dots,n-1\}} \left| \frac{1}{k+1} \frac{\sin(\pi i_\omega m_\ell(k+1)/n)}{\sin(\pi i_\omega m_\ell/n)} \right|, \qquad (\ell \in \{0, 1, \dots, \tau-1\}).$$

Thus, the expected consensus rate (product of $\tau$ mixing matrices) can be calculated as

$$p = 1 - \Lambda_2(\boldsymbol{W}^{(0)}\boldsymbol{W}^{(1)}\cdots\boldsymbol{W}^{(\tau-1)})$$

$$= 1 - \max_{i_\omega \in \{1,\dots,n-1\}} \left| \prod_{\ell=0}^{\tau-1} \frac{1}{k+1} \frac{\sin(\pi i_\omega m_\ell(k+1)/n)}{\sin(\pi i_\omega m_\ell/n)} \exp^{-\mathrm{j}\pi i_\omega m_\ell k/n} \right|$$

$$= 1 - \max_{i_\omega \in \{1,\dots,n-1\}} \left| \prod_{\ell=0}^{\tau-1} \frac{1}{k+1} \frac{\sin(\pi i_\omega m_\ell(k+1)/n)}{\sin(\pi i_\omega m_\ell/n)} \right|. \tag{10}$$

In our $k$-peer exponential graph, the exponential expansion $m_\ell = (k+1)^\ell$ is employed. Substituting $m_\ell = (k+1)^\ell$ into (10) results in:

$$p = 1 - \max_{i_\omega \in \{1,\dots,n-1\}} \left| \frac{1}{(k+1)^\tau} \frac{\sin(\pi i_\omega (k+1)^1/n)}{\sin(\pi i_\omega (k+1)^0/n)} \frac{\sin(\pi i_\omega (k+1)^2/n)}{\sin(\pi i_\omega (k+1)^1/n)} \cdots \frac{\sin(\pi i_\omega (k+1)^\tau/n)}{\sin(\pi i_\omega (k+1)^{\tau-1}/n)} \right|$$

$$= 1 - \max_{i_\omega \in \{1,\dots,n-1\}} \left| \frac{1}{(k+1)^\tau} \frac{\sin(\pi i_\omega (k+1)^\tau/n)}{\sin(\pi i_\omega/n)} \right|. \tag{11}$$

For specific $n$ (a power of $k+1$; namely, $n = (k+1)^\tau$), substituting it into (11) results in:

$$p = 1 - \max_{i_\omega \in \{1,\dots,n-1\}} \left| \frac{1}{(k+1)^\tau} \frac{\sin(\pi i_\omega (k+1)^\tau/(k+1)^\tau)}{\sin(\pi i_\omega/(k+1)^\tau)} \right|$$

$$= 1 - \max_{i_\omega \in \{1,\dots,n-1\}} \left| \frac{1}{(k+1)^\tau} \frac{\sin(\pi i_\omega)}{\sin(\pi i_\omega/(k+1)^\tau)} \right| = 1.$$

Otherwise, for any $n$, the expected consensus rate results in:

$$p = 1 - \max_{i_\omega \in \{1,\dots,n-1\}} \left| \frac{1}{(k+1)^\tau} \frac{\sin(\pi i_\omega (k+1)^\tau/n)}{\sin(\pi i_\omega/n)} \right|, \qquad (\tau = \lfloor \log_{k+1}(n) \rfloor).$$

These results are summarized in Theorem 2. Thanks to the exponential expansion $m_\ell = (k+1)^\ell$, division is recursively simplified (via cancellation of units) in the first reformulation of (11), which we refer to as the **exponential expansion trick**. This is crucial for demonstrating the efficiency of the $k$-peer exponential graph across $n$ and was not explored in the previous study [43].

### A.3  Calculation of null frequencies for circulant matrix employing MAFs

First, we review the principles underlying the 1-peer exponential graph explored in Sec. 3.2. Consider a sequence of length $(c_\ell + 1)$, denoted by $\{h^{(\ell)}(0), h^{(\ell)}(1), \dots, h^{(\ell)}(c_\ell)\}$, for $\ell \in \{0, 1, \dots, \tau-1\}$. Following Definition 7, the expanded sequence with order $m_\ell$, resulting in an $n$-length sequence $\{h_{\uparrow m_\ell}^{(\ell)}(0), h_{\uparrow m_\ell}^{(\ell)}(1), \dots, h_{\uparrow m_\ell}^{(\ell)}(n-1)\}$ is

$$h_{\uparrow m_\ell}^{(\ell)}(im_\ell) = \begin{cases} h^{(\ell)}(i) & i = 0, \dots, c_\ell \\ 0 & \text{otherwise} \end{cases}.$$

A circulant mixing matrix employing this sequence is given by

$$\boldsymbol{W}^{(\ell)} = \mathrm{circ}(h_{\uparrow m_\ell}^{(\ell)}(0), h_{\uparrow m_\ell}^{(\ell)}(1), \dots, h_{\uparrow m_\ell}^{(\ell)}(n-1)) = \begin{bmatrix} h_{\uparrow m_\ell}^{(\ell)}(0) & h_{\uparrow m_\ell}^{(\ell)}(n-1) & \cdots & h_{\uparrow m_\ell}^{(\ell)}(1) \\ h_{\uparrow m_\ell}^{(\ell)}(1) & h_{\uparrow m_\ell}^{(\ell)}(0) & \cdots & h_{\uparrow m_\ell}^{(\ell)}(2) \\ h_{\uparrow m_\ell}^{(\ell)}(2) & h_{\uparrow m_\ell}^{(\ell)}(1) & \cdots & h_{\uparrow m_\ell}^{(\ell)}(3) \\ \vdots & \ddots & \ddots & \vdots \\ h_{\uparrow m_\ell}^{(\ell)}(n-1) & h_{\uparrow m_\ell}^{(\ell)}(n-2) & \cdots & h_{\uparrow m_\ell}^{(\ell)}(0) \end{bmatrix}.$$

As performed in (5), eigenvalues of this is obtained by applying DFT, as

$$\boldsymbol{W}^{(\ell)} = \boldsymbol{D}^{\top} \boldsymbol{H}_{\uparrow m_\ell}^{(\ell)} \boldsymbol{D},$$

where $\boldsymbol{D}$ denotes the DFT matrix and the eigenvalue matrix is $\boldsymbol{H}_{\uparrow m_\ell}^{(\ell)} =$ $\mathrm{diag}([H_{\uparrow m_\ell}^{(\ell)}(0), \ldots, H_{\uparrow m_\ell}^{(\ell)}(n-1)])$, with

$$H_{\uparrow m_\ell}^{(\ell)}(i_\omega) = \sum_{i=0}^{n-1} h_{\uparrow m_\ell}^{(\ell)}(i) e^{-2\pi \mathrm{j} i_\omega i/n}, \qquad i_\omega \in \{0, 1, \ldots, n-1\}. \tag{12}$$

As mentioned in the main paper, the use of a doubly stochastic sequence ensures that $\sum_{i=0}^{n-1} h_{\uparrow m_\ell}^{(\ell)}(i) = 1$, preserving the DC component as $H_{\uparrow m_\ell}^{(\ell)}(0) = 1$. This is the maximum eigenvalue, since the sequence $\{h_{\uparrow m_\ell}^{(\ell)}(0), \ldots, h_{\uparrow m_\ell}^{(\ell)}(n-1)\}$ is positive-definite.

**MAF is used as a sequence $h^{(\ell)}(i)$**

Next, we identify the null frequencies when applying the MAFs discussed in Sec. 3.2. Suppose that the MAF with $(c_\ell + 1)$-length is employed as filter sequence:

$$h^{(\ell)}(i) = \frac{1}{c_\ell + 1} \qquad (i = 0, \ldots, c_\ell).$$

Associated with this, the expanded sequence with order $m_\ell$ results in an $n$-length filter sequence $\{h_{\uparrow m_\ell}^{(\ell)}(0), h_{\uparrow m_\ell}^{(\ell)}(1), \ldots, h_{\uparrow m_\ell}^{(\ell)}(n-1)\}$. As illustrated in (8), the frequency response regarding (continuous) angular frequency $\omega$ is given by

$$H_{\uparrow m_\ell}^{(\ell)}(e^{\mathrm{j}\omega}) = \frac{1}{c_\ell + 1} \frac{\sin(\omega m_\ell(c_\ell + 1)/2)}{\sin(\omega m_\ell/2)} \exp^{-\mathrm{j}\omega m c_\ell/2},$$

where frequency index $i_\omega$ and angular frequency $\omega$ are related by $\omega = 2\pi i_\omega/n$. As proven in Sec. B.1, null frequencies can be identified as follows:

$$\mathcal{N}_{\mathrm{MAF}}^{(\ell)} = \left\{ \text{Null frequency } i_\omega \text{ is multiples of } \frac{n}{m_\ell(c_\ell + 1)}, \text{ excluding multiples of } \frac{n}{m_\ell} \text{ within } i_\omega \in [0, n) \right\}. \tag{13}$$

## A.4 Examples of $k$-peer exponential graph

**Example 1** ($k$-peer exponential graph). *Given $(n, k) = (16, 1)$, the $1$-peer exponential graph is constructed using $\tau = \log_2(16) = 4$ circulant matrices:*

$$\boldsymbol{W}^{(0)} = \operatorname{circ}(\tfrac{1}{2}, \tfrac{1}{2}, \underbrace{0, 0, \dots, 0}_{14 \text{ zeros}}),$$

$$\boldsymbol{W}^{(1)} = \operatorname{circ}(\tfrac{1}{2}, 0, \tfrac{1}{2}, 0, \underbrace{0, \dots, 0}_{12 \text{ zeros}}),$$

$$\boldsymbol{W}^{(2)} = \operatorname{circ}(\tfrac{1}{2}, 0, 0, 0, \tfrac{1}{2}, 0, 0, 0, \underbrace{0, \dots, 0}_{8 \text{ zeros}}),$$

$$\boldsymbol{W}^{(3)} = \operatorname{circ}(\tfrac{1}{2}, \underbrace{0, 0, \dots, 0}_{7 \text{ zeros}}, \tfrac{1}{2}, \underbrace{0, 0, \dots, 0}_{7 \text{ zeros}}).$$

*Next, null indices are identified for each mixing matrix $\{0, 1, 2, 3\} \in \ell$ using (13), as:*

$$\mathcal{N}^{(0)} \ni \{8\}, \qquad \text{(MAF with } n = 16, c_0 = k = 1, m_0 = 2^0 = 1\text{)},$$

$$\mathcal{N}^{(1)} \ni \{4, 12\}, \qquad \text{(MAF with } n = 16, c_1 = k = 1, m_1 = 2^1 = 2\text{)},$$

$$\mathcal{N}^{(2)} \ni \{2, 6, 10, 14\}, \qquad \text{(MAF with } n = 16, c_2 = k = 1, m_2 = 2^2 = 4\text{)},$$

$$\mathcal{N}^{(3)} \ni \{1, 3, 5, 7, 9, 11, 13, 15\}, \qquad \text{(MAF with } n = 16, c_3 = k = 1, m_3 = 2^3 = 8\text{)}.$$

*Combining the above four null frequency sets satisfies (6), confirming that Example 1 satisfies finite-time convergence.*

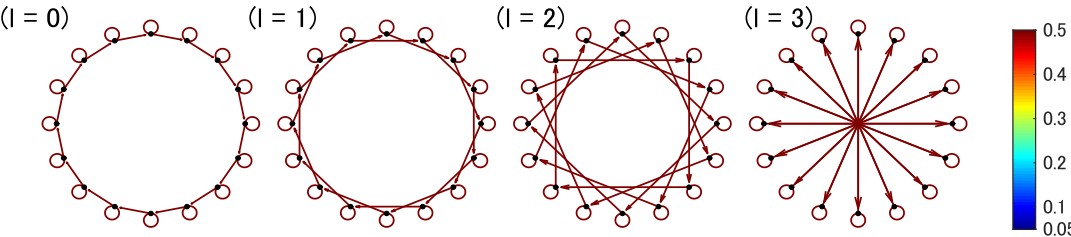

Figure 5: Mixing matrices of $k$-peer exponential graph with $(n, k) = (16, 1)$. Each graph consists of $n = 16$ nodes, depicted as black dots arranged in a circle and interconnected by lines. The colors of these lines indicate the mixing weights between nodes in $\boldsymbol{W}^{(\ell)}$.

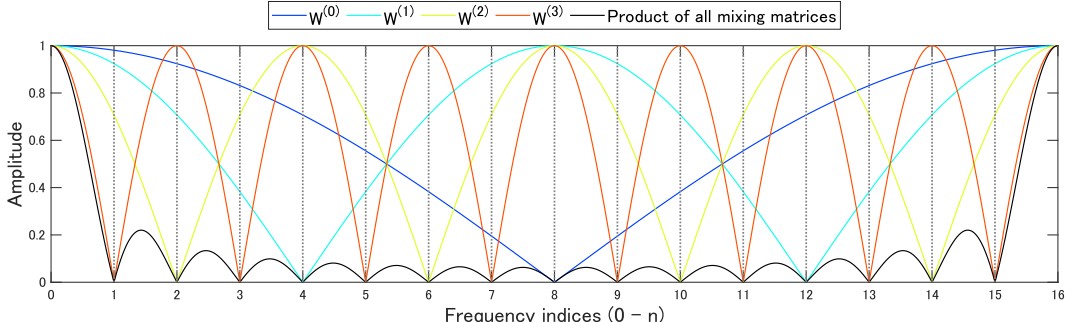

Figure 6: Frequency characteristics (amplitude) of each sequence.

**Example 2** (*k*-peer exponential graph)**.** *Given* $(n,k) = (27,1)$, *the* $1$-*peer exponential graph is constructed using* $\tau = \lfloor \log_2(24) \rfloor = 4$ *circulant matrices:*

$$\boldsymbol{W}^{(0)} = \mathrm{circ}(\tfrac{1}{2}, \tfrac{1}{2}, \underbrace{0, 0, \ldots, 0}_{25 \text{ zeros}}),$$

$$\boldsymbol{W}^{(1)} = \mathrm{circ}(\tfrac{1}{2}, 0, \tfrac{1}{2}, 0, \underbrace{0, \ldots, 0}_{23 \text{ zeros}}),$$

$$\boldsymbol{W}^{(2)} = \mathrm{circ}(\tfrac{1}{2}, 0, 0, 0, \tfrac{1}{2}, 0, 0, 0, \underbrace{0, \ldots, 0}_{19 \text{ zeros}}),$$

$$\boldsymbol{W}^{(3)} = \mathrm{circ}(\tfrac{1}{2}, \underbrace{0, 0, \ldots, 0}_{7 \text{ zeros}}, \tfrac{1}{2}, \underbrace{0, 0, \ldots, 0}_{18 \text{ zeros}}).$$

*However, null indices for each mixing matrix* $\{0, 1, 2, 3, 4\} \in \ell$ *using* (13) *form the following:*

$$\mathcal{N}^{(0)} \ni \{\tfrac{27}{2}\}, \qquad\qquad \textit{(MAF with } n = 27, c_0 = k = 1, m_0 = 2^0 = 1),$$

$$\mathcal{N}^{(1)} \ni \{\tfrac{27}{4}, \tfrac{81}{4}\}, \qquad\qquad \textit{(MAF with } n = 27, c_1 = k = 1, m_1 = 2^1 = 2),$$

$$\mathcal{N}^{(2)} \ni \{\tfrac{27}{8}, \tfrac{81}{8}, \tfrac{135}{8}, \tfrac{189}{8}\}, \qquad \textit{(MAF with } n = 27, c_2 = k = 1, m_2 = 2^2 = 4),$$

$$\mathcal{N}^{(3)} \ni \{\tfrac{27}{16}, \tfrac{81}{16}, \tfrac{135}{16}, \tfrac{189}{16}, \tfrac{243}{16}, \tfrac{297}{16}, \tfrac{351}{16}, \tfrac{405}{16}\}, \textit{(MAF with } n = 27, c_3 = k = 1, m_3 = 2^3 = 8),$$

*Since* (6) *is not satisfied, Example 2 is not a finite-time convergence graph.*

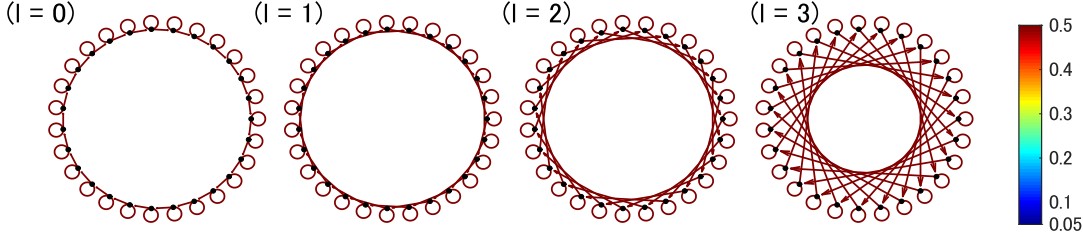

Figure 7: Mixing matrices of $k$-peer exponential graph with $(n, k) = (27, 1)$. Each graph consists of $n = 27$ nodes, depicted as black dots arranged in a circle and interconnected by lines. The colors of these lines indicate the mixing weights between nodes in $\boldsymbol{W}^{(\ell)}$.

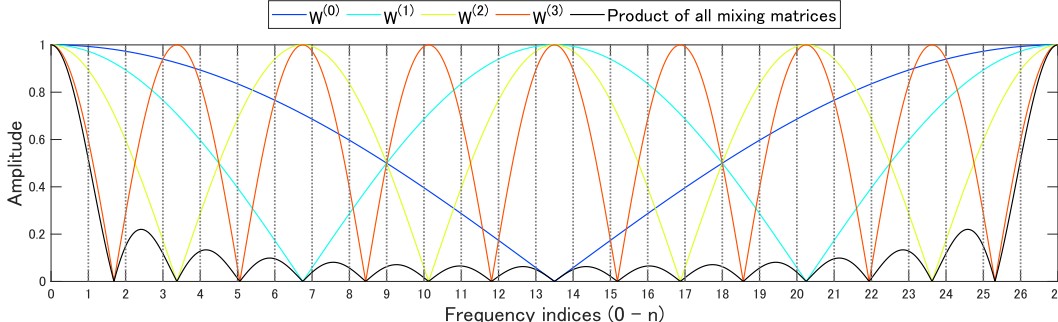

Figure 8: Frequency characteristics (amplitude) of each sequence.

**Example 3** (*k*-peer exponential graph). *For $(n, k) = (27, 2)$, the 2-peer exponential graph consists of $\tau = \log_3(27) = 3$ mixing matrices:*

$$\boldsymbol{W}^{(0)} = \operatorname{circ}\big(\tfrac{1}{3}, \tfrac{1}{3}, \tfrac{1}{3}, \underbrace{0, 0, \dots, 0}_{24 \text{ zeros}}\big),$$

$$\boldsymbol{W}^{(1)} = \operatorname{circ}\big(\tfrac{1}{3}, 0, 0, \tfrac{1}{3}, 0, 0, \tfrac{1}{3}, 0, 0, \underbrace{0, \dots, 0}_{18 \text{ zeros}}\big),$$

$$\boldsymbol{W}^{(2)} = \operatorname{circ}\big(\tfrac{1}{3}, \underbrace{0, 0, \dots, 0}_{8 \text{ zeros}}, \tfrac{1}{3}, \underbrace{0, 0, \dots, 0}_{8 \text{ zeros}}, \tfrac{1}{3}, \underbrace{0, 0, \dots, 0}_{8 \text{ zeros}}\big).$$

*Next, null frequency sets are identified for each mixing matrix $\{0, 1, 2\} \in \ell$ using* (13)*:*

$\mathcal{N}^{(0)} \ni \{9, 18\},$      *(MAF with $n = 27, c_0 = k = 2, m_0 = 3^0 = 1$),*

$\mathcal{N}^{(1)} \ni \{3, 6, 12, 15, 21, 24\},$      *(MAF with $n = 27, c_1 = k = 2, m_1 = 3^1 = 3$),*

$\mathcal{N}^{(2)} \ni \{1, 2, 4, 5, 7, 8, 10, 11, 13, 14, 16, 17, 19, 20, 22, 23, 25, 26\},$      *(MAF with $n = 27, c_2 = k = 2, m_2 = 3^2 = 9$),*

*Since this also fulfills* (6)*, Example 3 is confirmed to be a finite-time convergent graph.*

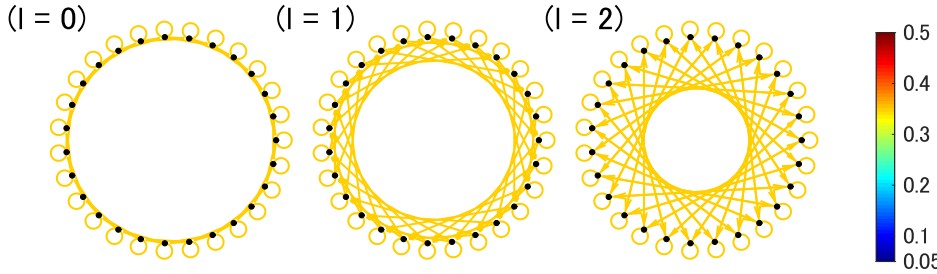

Figure 9: Mixing matrices of $k$-peer exponential graph with $(n, k) = (27, 2)$. Each graph consists of $n = 27$ nodes, depicted as black dots arranged in a circle and interconnected by lines. The colors of these lines indicate the mixing weights between nodes in $\boldsymbol{W}^{(\ell)}$.

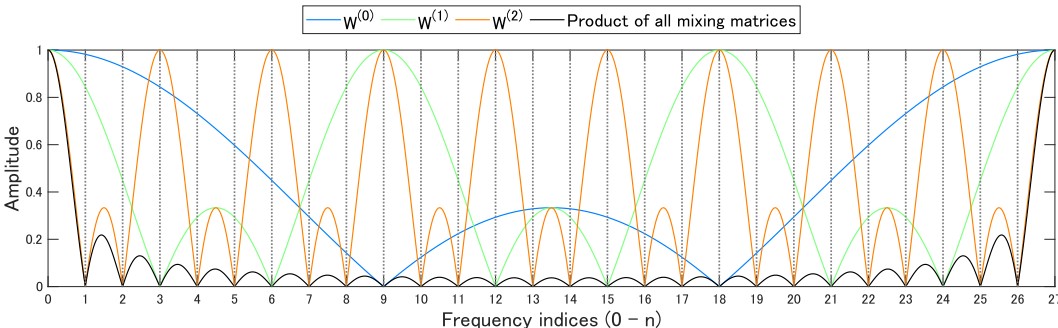

Figure 10: Frequency characteristics (amplitude) of each sequence.

# B  Technical details for null-cascade graph in Sec. 4

Technical proofs of lemmas and theorems in Sec. 4 are shown in Sec. B.1–B.3. Section B.4 provides the invariant property regarding the circulant shifting, which involves assigning maximum weight to the targeted node. Subroutines used in Alg. 1 are detailed in Sec. B.5. Associated example graphs are illustrated in Sec. B.6.

## B.1  Proof of Lemma 2

In (8), the frequency response of MAF with communication order $c_\ell$ and expansion order $m_\ell$ is formulated. To determine the frequency indices at which the frequency response of the expanded MAF becomes zero (nulls), we need to find the angular frequencies $\omega$ such that the numerator of (8) equals zero; namely, $\sin(\omega m_\ell(c_\ell+1)/2) = 0$. This satisfies when $\omega m_\ell(c_\ell+1)/2 = \phi\pi$ where $\phi$ is an integer. This yields $\omega = \frac{2\phi\pi}{m_\ell(c_\ell+1)}$. Considering $\phi$ ranges from 0 to $m_\ell(c_\ell+1) - 1$. However, since the DC component ($\omega = 0$) is not null and periodicity due to the expansion from Definition 7, $\phi$ can be determined such that multiples of 1, excluding multiples of $(c_\ell + 1)$, consisting of $\phi \in \{1, \ldots, (c_\ell + 1) - 1, (c_\ell + 1) + 1, \ldots, 2(c_\ell + 1) - 1, 2(c_\ell + 1) + 1, \ldots, (m_\ell - 1)(c_\ell + 1) - 1, (m_\ell - 1)(c_\ell + 1) + 1, \ldots, m_\ell(c_\ell + 1) - 1\}$. Thus, nulls can be obtained at multiples of $\frac{2\pi}{m_\ell(c_\ell+1)}$, excluding multiples of $\frac{2\pi}{m_\ell}$. From the fact that the angular frequency $\omega$ and frequency indices $i_\omega$ are related by $\omega = 2\pi i_\omega/n$, this can be reformulated: $i_\omega$ can be determined such that multiples of $\frac{n}{m_\ell(c_\ell+1)}$, excluding multiples of $\frac{n}{m_\ell}$ within $i_\omega \in [0, 2\pi)$:

$$\mathcal{N}_{\text{MAF}}^{(\ell)} = \Big\{ \text{Null frequency } i_\omega \text{ is multiples of } \frac{n}{m_\ell(c_\ell+1)}, \text{ excluding multiples of } \frac{n}{m_\ell} \text{ within } i_\omega \in [0, n) \Big\}.$$

## B.2  Proof of Lemma 3

Recall from Definition 8 the fundamental property of SNF, which forms $c_\ell = 2\lfloor k/2 \rfloor$ nulls at $q_{\ell,\psi}$ and their conjugates $q_{\ell,\psi}^* = n - q_{\ell,\psi}$ for $\psi \in \{1, ..., c_\ell/2\}$. From Definition 7, the expanded SNF with order $m_\ell$, null frequencies can be shifted by $1/m_\ell$. Therefore, the null frequencies are identified as follows:

$$\mathcal{N}_{\text{SNF}}^{(\ell)} = \Big\{ \text{Null frequency } i_\omega \text{ is } \frac{q_{\ell,\psi}+\phi n}{m_\ell} \text{ and } \frac{q_{\ell,\psi}^*+\phi n}{m_\ell} \text{ for } \psi = \{1, ..., c_\ell/2\}, \phi \in \{0, ..., m_\ell - 1\} \text{ within } i_\omega \in [0, n) \Big\}.$$

## B.3  Proof of Theorem 3

First, we explain our proof strategy demonstrating that the null-cascade graph achieves finite-time convergence for any $n$. As explained in Sec 3, when a graph consists of $\tau$ circulant mixing matrices as in Definition 1, it achieves finite-time convergence when cascading these $\tau$ mixing matrices results in null formation at all non-zero frequencies $i_\omega \in \{1, \ldots, n-1\}$, as demonstrated in (6). Utilizing this principle, if we calculate a set of null frequency indices for each $\boldsymbol{W}^{(\ell)}(\ell \in \{0, \ldots, \tau - 1\})$ within the null-cascade graph and demonstrate that the union of these sets covers all non-zero frequencies, the finite-time convergence property in the null-cascade graph can be established.

Suppose $n$ is factorized as $n = \prod_{i=1}^{\lambda}(\nu_i + 1)$, where $\nu_1 \leq ... \leq \nu_\kappa \leq k < \nu_{\kappa+1} \leq ... \leq \nu_\lambda$, with each $\nu_i \in \mathbb{N}$ (line 2 in Alg. 1). Initially, the remaining frequencies to be nullified, $i_\omega \in \{1, \ldots, n-1\}$, are equivalently represented as multiples of $b = 1$ within the interval $[0, n)$ (line 1 in Alg. 1). Given the factorization of $n$, the task of nullifying all non-zero frequencies can be interpreted as applying mixing matrices to form nulls at multiples of $b$, excluding multiples of $b(\nu_i + 1)$—referred to as **frequencies associated with** $\nu_i$ in the main paper. This process is repeated sequentially until $b$ reaches $n$. Once $b$ equals $n$, nullification at non-zero frequencies $i_\omega \in \{1, \ldots, n-1\}$ is accomplished. Thus, the focus is on generating filters to form nulls specifically at multiples of $b$ excluding multiples of $b(\nu_i + 1)$.

In Step 1 (lines 4-11) of Alg. 1, we utilize MAFs with dynamic selection of communication order $c_\ell$ and expansion order $m_\ell$ to nullify frequencies associated with $\{\nu_1, \ldots, \nu_\kappa\}$ satisfying $\nu_i \leq k$. According to Lemma 2, forming nulls at frequencies associated with $\nu_i(\leq k)$ is straightforward. In

line 4 in Alg. 1, we sequentially select from $\nu_\kappa$ down to $\nu_1$ as $c_\ell$ ($\ell \in \{0, ..., \kappa - 1\}$), because many nulls can be efficiently formed by choosing the largest possible values for both $c_\ell$ and $m_\ell$. In lines 5 and 6 in Alg. 1, $(c_0, m_0)$ are selected (with initialization $b = 1$) as

$$c_0 = \nu_\kappa, \quad m_0 = \frac{n}{(c_0+1)b} = \frac{\prod_{i=1}^{\lambda}(\nu_i+1)}{(\nu_0+1)b} = \frac{\prod_{i=1}^{\lambda}(\nu_i+1)}{\nu_\kappa+1} = \prod_{i \in \{1,...,\kappa-1,\kappa+1,...,\lambda\}}(\nu_i + 1).$$

After generating an expanded MAF with $(c_0, m_0)$, a corresponding mixing matrix $\boldsymbol{W}^{(0)}$ is calculated (lines 7-8 in Alg. 1). According to Lemma 2, this mixing matrix enables to formation of nulls at frequencies that are multiples of $\frac{n}{m_0(c_0+1)} = b$, while excluding multiples of $\frac{n}{m_0} = (c_0 + 1)b$. Consequently, the base number to be nullified is updated to $b \leftarrow b(\nu_\kappa + 1) = \nu_\kappa + 1$ (line 10 in Alg. 1). This process eliminates the frequencies associated with a factor $\nu_\kappa$.

In Step 1, this process is repeated to sequentially nullify the frequencies associated with factors $\{\nu_\kappa, \ldots, \nu_1\}$ satisfying $\nu_i \leq k$. For each factor $i \in \{1, \ldots, \kappa\}$, $(c_i, m_i, b)$ given by Alg. 1 can be generalized as

$$c_\ell = \nu_{\kappa-\ell}, \quad m_\ell = \prod_{i \in \{1,...,(\kappa-\ell-1),(\kappa+1),...,\lambda\}}(\nu_i + 1), \quad b = \prod_{i \in \{(\kappa-\ell),...,\kappa\}}(\nu_i + 1).$$

Repeating this process (Step 1) $\kappa$ times results in

$$c_{\kappa-1} = \nu_1, \quad m_{\kappa-1} = \prod_{i \in \{(\kappa+1),...,\lambda\}}(\nu_i + 1), \quad b = \prod_{i \in \{1,...,\kappa\}}(\nu_i + 1).$$

This indicates that the remaining frequencies to be nullified after the cascade of MAFs are identified as multiples of $b = \prod_{i \in \{1,...,\kappa\}}(\nu_i + 1)$.

To nullify the remaining frequencies associated with the factors $\{\nu_{\kappa+1}, \ldots, \nu_\lambda\}$ satisfying $\nu_i > k$, Step 2 involves generating SNFs with a fixed communication order $c_j = 2\lfloor \frac{k}{2} \rfloor$ as specified in Definition 8. According to Lemma 3, nulls can be formed at frequencies $\frac{q_{\ell,\psi}+\phi n}{m_\ell}$ and their conjugates $\frac{q_{\ell,\psi}^* + \phi n}{m_\ell}$, where $\psi = \{1, \ldots, c_\ell/2\}$ and $\phi \in \{0, \ldots, m_\ell - 1\}$. While each SNF can generate many nulls depending on the choice of $(m_\ell, q_{\ell,\psi})$, we particularly focus on the frequencies when setting $\phi = 0$; specifically, $(\frac{q_{\ell,\psi}}{m_\ell}, \frac{q_{\ell,\psi}^*}{m_\ell})$, where $(\psi \in \{1, \ldots, c_\ell/2\})$. The selection of $(m_\ell, q_{\ell,\psi})$ in Alg. 4 is strategically chosen to align $\frac{q_{\ell,\psi}}{m_\ell}$ with the remaining target frequencies associated with $\nu_i(> k)$ (multiples of $b$). To generate nulls at multiples of $b_i$, $\tau_{\mathrm{snf}}(\nu_i) = \lceil \nu_i/c_\ell \rceil$ SNFs are required to form nulls at remaining frequencies associated with $\nu_i(> k)$, since $c_j = 2\lfloor \frac{k}{2} \rfloor$. After these frequencies are nullified, the base number to be nullified is updated to $b = \prod_{i \in \{1,...,\kappa,\lambda-i+1,...,\lambda\}}(\nu_i + 1)$ (line 19 in Alg. 1). By repeating this nullification process with SNFs, $b_i$ eventually becomes reaches as $b = \prod_{i \in \{1,...,\lambda\}}(\nu_i + 1) = n$. This comprehensive nullification at non-zero frequencies ensures that finite-time convergence is achieved within the null-cascade graph.

### B.4 Invariant frequency amplitude regarding circulant-shifting

When using non-uniform weights (e.g., SNFs introduced in Sec. 4), assigning the maximum weight to the local model held by each node would be effective in decentralized learning. To this aim, the following lemma is effective for circulant-shifting of the filter sequence (e.g., used in Alg. 3).

**Lemma 4** (Invariant frequency amplitude regarding circulant shifting). *Suppose $n$-length sequence $\{h(0), h(1), \ldots, h(n-1)\}$ and its frequency response $H(e^{\mathrm{j}\omega})$. Its circulant shifted sequence with order $u$ is defined by $h_{\to u}(i) = h((i - u) \mod n)$ and its frequency response is denoted by $H_{\to u}(e^{\mathrm{j}\omega})$. The frequency response (amplitude) of the circulant shifted sequence is invariant for any $u$:*

$$|H_{\to u}(e^{\mathrm{j}\omega})| = |H(e^{\mathrm{j}\omega})|.$$

*Proof.* Suppose that $n$-length sequence $\{h(0), h(1), \ldots, h(n-1)\}$ and its frequency response $H(e^{\mathrm{j}\omega})$ are given by

$$H(e^{\mathrm{j}\omega}) = \sum_{i=0}^{n-1} h(i) \exp^{-\mathrm{j}\omega i}.$$

The circulant shift with shifting order $u$ is defined by $h_{\to u}(i) = h((i - u) \mod n)$, which is is equivalent to generate the following sequence: $\{h_{\to u}(0), h_{\to u}(1), \ldots, h_{\to u}(n-1)\} = \{h(u), h(u+1), \ldots, h(n-1), h(0), \ldots, h(u-1)\}$. The frequency response of it is given by:

$$
\begin{aligned}
H_{\to u}(e^{j\omega}) &= \sum_{i=0}^{n-1} h_{\to u}(i) \exp^{-j\omega i} \\
&= \sum_{i=0}^{n-1} h_{\to u}(i+u) \exp^{-j\omega(i+u)} \\
&= \exp^{-j\omega u} \sum_{i=0}^{n-1} h_{\to u}(i+u) \exp^{-j\omega i} \\
&= H(e^{j\omega}) \exp^{-j\omega u}.
\end{aligned}
$$

Since $|H_{\to u}(e^{j\omega})| = |H(e^{j\omega})e^{-\omega u}| = |H(e^{j\omega})|$, the frequency amplitude regarding circulant shifting is invariant.

$\square$

Based on Lemma 4, Alg. 2 is constructed. This is a subroutine of Alg. 3.

---

**Algorithm 2** Circulant shifting function

---

1: **function** $h_{\to u}(i) = \text{CirculantShifting}(h(i))$
2: ▷ Ciurculant shifting to assign maximum weight to local node
3: $u = \arg\max_i(h(i))$
4: $[h_{\to u}(0), h_{\to u}(1), \ldots, h_{\to u}(n-1)] = [h(u), \ldots, h(n-1), h(0), \ldots, h(u-1)]$

---

## B.5 Subroutines to compute SNFs

A subroutine used in Alg. 1 is outlined in Alg. 3. The aim of this subroutine is to nullify the remaining frequencies associated with the factors $\{\nu_{\kappa+1}, \ldots, \nu_\lambda\}$ satisfying $\nu_i > k$ by generating SNFs. As described in Definition 8, SNFs that form nulls at target frequencies can be obtained by solving a polynomial expression. Specifically, line 2 (calling the subroutine in Alg. 4) selects $(m_\ell, p_{\ell,\psi})$ for $\psi \in \{1, \ldots, c_\ell/2\}$, and lines 4-6 compute the conjugate frequency indices. Following (9), the polynomial is expressed with roots corresponding to frequency indices $\{q_{\ell,1}, \ldots, q_{\ell,c_\ell/2}\}$ and their conjugate frequency indices $\{q_{\ell,1}^*, \ldots, q_{\ell,c_\ell/2}^*\}$ in line 8, yielding a $c_\ell$-order polynomial equation of the form: $\gamma(0)x^{c_\ell} + \cdots + \gamma(c_\ell-1)x + \gamma(c_\ell)$. To ensure the doubly stochastic property in a $(c_\ell+1)$-length filter sequence, the normalized filter sequence is computed in lines 10-12. In line 14, thanks to the invariant frequency amplitude regarding circulant shifting, detailed in Lemma 4, the maximum weight assigned to the local model held by each node would be effective in decentralized learning.

Next, we briefly explain the core functionality of Alg. 4 for selecting $(m_\ell, p_{\ell,\psi})$ to nullify frequencies associated with $\nu_i$. As discussed in Sec. B.3, the selection of $(m_\ell, p_{\ell,\psi})$ is not uniquely determined. Therefore, an implementation for selecting $(m_\ell, p_{\ell,\psi})$ is introduced in Alg. 4, which can be improved in future work. As outlined in Definition 8, communication order is set as $c_j = 2\lfloor \frac{k}{2} \rfloor$ in line 6 for two reasons: (i) the desire to use an even value of $c_\ell$ to form pairwise nulls in the conjugate relationship for real-valued filter sequence, and (ii) the goal of achieving as many nulls as possible. According to Lemma 3, nulls can be formed at frequencies $\frac{q_{\ell,\psi}+\phi n}{m_\ell}$ and their conjugates $\frac{q_{\ell,\psi}^*+\phi n}{m_\ell}$, where $\psi = \{1, \ldots, c_\ell/2\}$ and $\phi \in \{0, \ldots, m_\ell - 1\}$. While each SNF can generate many nulls depending on the choice of $(m_\ell, q_{\ell,\psi})$, we particularly focus on the frequencies when setting $\phi = 0$; specifically, $(\frac{q_{\ell,\psi}}{m_\ell}, \frac{q_{\ell,\psi}^*}{m_\ell})$, where $(\psi \in \{1, \ldots, c_\ell/2\})$. To generate nulls at multiples of $b_i$, $\tau_{\text{snf}}(\nu_i) = \lceil \nu_i/c_\ell \rceil$ SNFs are required to form nulls at remaining frequencies associated with $\nu_i(> k)$, since $c_j = 2\lfloor \frac{k}{2} \rfloor$. After these frequencies are nullified, the base number to be nullified is updated to $b = \prod_{i \in \{1, \ldots, \kappa, \lambda-i+1, \ldots, \lambda\}} (\nu_i + 1)$ (line 19 in Alg. 1).

Finally, runtime cost and implementation of Alg. 1-4 is briefly noted. Firstly our source code associated with Alg. 1-4 is available on GitHub (see Abstract). Alg. 1- 3 are straightforward to implement and incur negligible runtime overhead. Alg. 4, which is used to select the roots $q_\ell$, expansions $m_\ell$, and communication orders $c_\ell$ for computing SNFs, may be complex to implement and may introduce noticeable runtime costs. This potential complexity stems from the fact that combinations of $(q_\ell, m_\ell, c_\ell)$ are not uniquely determined, requiring a greedy search to identify suitable candidates. While we have not exhaustively evaluated the runtime cost for all possible $(n, k)$-configurations, we confirm that for those configurations tested in our experiments (as reported in Section 5 and Appendix F), the runtime costs were negligible.

---

**Algorithm 3** SNF: a subroutine of Alg. 1

---

1: **function** $\{\widetilde{\gamma}_j(0),...,\widetilde{\gamma}_j(n-1)\}_{j \in \{1,...,\tau_{\text{snf}}(\nu_i)\}} = \text{SNF}(n, k, \nu_i, b)$
2: $\{c_j, m_j, p_{j,1}, \ldots, p_{j,c_j/2}\}_{j=1,\ldots,\tau_{\text{snf}}(\nu_i)} = \text{SelectOrders}(n, k, \nu_i, b)$     /* Alg. 4 */
3: **for** $j = 1, \ldots, \tau_{\text{snf}}(\nu_i)$ **do**
4:     **for** $\psi = 1, \ldots, c_j/2$ **do**
5:        $q_{j,\psi}^* = n - q_{j,\psi}$     /* conjugate frequency index to be nullified */
6:     **end for**
7:     ▷ compute coefficients of a polynomial equation
8:     $\prod_{\psi=1}^{c_j/2}(x - e^{\mathrm{j}2\pi q_{j,\psi}/n})(x - e^{\mathrm{j}2\pi q_{j,\psi}^*/n}) = \gamma_j(0)x^{c_j} + \cdots + \gamma_j(c_j-1)x + \gamma_j(c_j)$
9:     ▷ compute the normalized sequence
10:     **for** $i = 0, \ldots, c_j$ **do**
11:        $\overline{\gamma}_j(i) = \gamma_j(i)/\sum_{l=0}^{c_j} \gamma_j(l)$
12:     **end for**
13:     ▷ circulant shifting after expansion with order $m_j$
14:     $\widetilde{\gamma}_j(i) = \text{CirculantShifting}(\overline{\gamma}_{j,\uparrow m_j}(i))$     /* Alg. 2 */
15: **end for**

---

**Algorithm 4** An implementation of subroutine to select roots, expansions, and communication orders

1: **function** $\{c_j, m_j, q_{j,1}, \ldots, q_{j,c_j/2}\}_{j=1,\ldots,\tau_{\text{snf}}(\nu_i)} = \text{SelectOrders}(n, k, \nu_i, b)$

2:  $\beta \in \{\frac{n}{\nu_i+1}, \frac{2n}{\nu_i+1}, \ldots, \frac{\lfloor \nu_i/2 \rfloor n}{\nu_i+1}\}$   /* frequency indices to be nullified */

3:  $m_{\text{base}} = \frac{n}{(\nu_i+1)b}$    /* base expansion order corresponding to factorization number $\nu_i$ */

4:  $j = 0$         /* SNF index to be incremented */

5: **while** (1) **do**

6:    $c_j = 2\lfloor \frac{k}{2} \rfloor$    /* we use an even number less than $k$ is used as communication order */

7:    $m_{\text{tmp}} = 1$    /* temporary expansion order to incrementally search */

8:    ▷ compute roots of a polynomial equation and expansion order

9:    $\{q_{j,\psi}\}_{\psi \in \{1,\ldots,c_j/2\}} = \text{SelectFreqIndices}(\beta, c_j/2)$    /* $c_j/2$ frequency indices to be nullified are selected from the higher indices of $\beta$. If $|\beta| \neq c_j/2$, other frequencies such that interpolates $\beta$ are selected. */

10:    **while** (1) **do**

11:      **if** $m_{\text{tmp}} \geq 2$ **then**

12:        **for** $\psi = 1, \ldots, c_j/2$ **do**

13:          $q_{j,\psi} = m_{\text{tmp}} \cdot q_{j,\psi}$    /* expanded SNF is used */

14:        **end for**

15:      **end if**

16:      **for** $\psi = 1, \ldots, c_j/2$ **do**

17:        $q^*_{j,\psi} = n - q_{j,\psi}$    /* conjugate frequency index to be nullified */

18:      **end for**

19:      ▷ compute coefficients of a polynomial equation

20:      $\prod_{\psi=1}^{c_j/2}(x - e^{\mathrm{j}2\pi q_{j,\psi}/n})(x - e^{\mathrm{j}2\pi q^*_{j,\psi}/n}) = \gamma_j(0)x^{c_j} + \cdots + \gamma_j(c_j-1)x + \gamma_j(c_j)$

21:      **if** $\{\gamma_j(0), \ldots, \gamma_j(c_j)\}$ are positive definite real numbers **then**

22:        $\beta = \beta \setminus \{q_{j,\psi}/m_{\text{tmp}}\}_{\psi \in \{1,\ldots,c_j/2\}}$   /* remove nullified frequency indices */

23:        **break**

24:      **else**

25:        $m_{\text{tmp}} = m_{\text{tmp}} + 1$  /* expansion order increment */

26:      **end if**

27:    **end while**

28:    ▷ set expansion order $m_j$

29:    $m_j = m_{\text{base}} \cdot m_{\text{tmp}}$

30:    $j = j + 1$         /* SNF index increment */

31:    **if** $m_{\text{tmp}} \geq 2$ **then**

32:      Factorization of $m_{\text{tmp}} = \mu_1, \ldots, \mu_v$, where $\mu_1 \leq \cdots \leq \mu_v$ and $\mu_1 = 1$

33:      **for** $s = v, \ldots, 1$ **do**

34:        $c_j = 2\lfloor \frac{k}{2} \rfloor$

35:        $m_j = m_{\text{base}} \cdot \mu_s$

36:        **for** $\psi = 1, \ldots, c_j/2$ **do**

37:          $q_{j,\psi} = q_{j-1,\psi}$

38:        **end for**

39:        $\beta = \beta \setminus \{q_{j,\psi}/\mu_s\}_{\psi \in \{1,\ldots,c_j/2\}}$   /* remove nullified frequency indices */

40:        $j = j + 1$    /* SNF index increment */

41:      **end for**

42:    **end if**

43:    **if** $|\beta| = 0$  (If a set of frequencies to be nullified is empty) **then**

44:      **break**

45:    **end if**

46: **end while**

47: $\tau_{\text{snf}}(\nu_i) = j$    /* number of mixing matrices employing SNFs */

## B.6 Examples of null-cascade graph

**Example 4** (Null-cascade graph). *Given $(n, k) = (15, 2)$, where $n$ can be factorized as $n = 15 = 5 \times 3$. Since $k = 2$, $n$ is not a composite number that can be factorized using only integers less than or equal to $k + 1 = 3$. The null-cascade graph consists of $\tau = 4$ circulant matrices, as*

$$\boldsymbol{W}^{(0)} = \mathrm{circ}(\tfrac{1}{3}, \underbrace{0, \ldots, 0}_{4 \text{ zeros}}, \tfrac{1}{3}, \underbrace{0, \ldots, 0}_{4 \text{ zeros}}, \tfrac{1}{3}, \underbrace{0, \ldots, 0}_{4 \text{ zeros}}),$$

$$\boldsymbol{W}^{(1)} = \mathrm{circ}(\tfrac{-2\cos(2\pi \cdot (6/15))}{2 - 2\cos(2\pi \cdot (6/15))}, 0, \tfrac{1}{2 - 2\cos(2\pi \cdot (6/15))}, \underbrace{0, \ldots, 0}_{12 \text{ zeros}}, \tfrac{1}{2 - 2\cos(2\pi \cdot (6/15))}, 0),$$

$$\boldsymbol{W}^{(2)} = \mathrm{circ}(\tfrac{-2\cos(2\pi \cdot (6/15))}{2 - 2\cos(2\pi \cdot (6/15))}, \tfrac{1}{2 - 2\cos(2\pi \cdot (6/15))}, \underbrace{0, \ldots, 0}_{12 \text{ zeros}}, \tfrac{1}{2 - 2\cos(2\pi \cdot (6/15))}),$$

*where null responses for each $\ell \in \{0, 1, 2\}$ are obtained at following frequency indices:*

$$\mathcal{N}^{(0)} \ni \{1, 2, 4, 5, 7, 8, 10, 11, 13, 14\}, \quad \text{(Step 1: MAF with } n = 15, c_0 = 2, m_0 = 5),$$

$$\mathcal{N}^{(1)} \ni \{3, \tfrac{9}{2}, \tfrac{21}{2}, 12\}, \quad \text{(Step 2: SNF with } n = 15, c_1 = 2, m_1 = 2, (q_{1,1}, q_{1,1}^*) = (6, 9)),$$

$$\mathcal{N}^{(2)} \ni \{6, 9\}, \quad \text{(Step 2: SNF with } n = 15, c_2 = 2, m_2 = 1, (q_{2,1}, q_{2,1}^*) = (6, 9)).$$

*Combining the above four null frequency sets satisfies* (6)*, confirming that Example 4 satisfies finite-time convergence.*

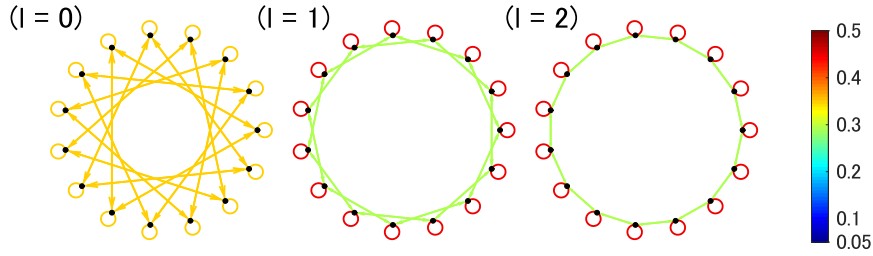

Figure 11: Mixing matrices of null-cascade graph with $(n, k) = (15, 2)$. Each graph consists of $n = 15$ nodes, depicted as black dots arranged in a circle and interconnected by lines. The colors of these lines indicate the mixing weights between nodes in $\boldsymbol{W}^{(\ell)}$.

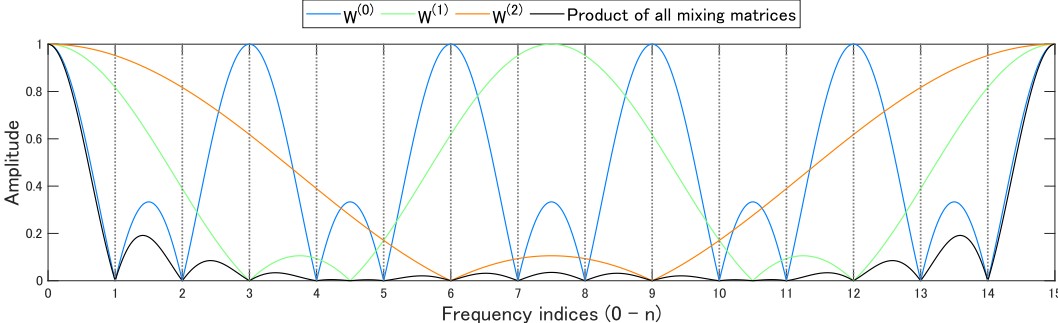

Figure 12: Frequency characteristics (amplitude) of each sequence.

**Example 5** (Null-cascade graph). *Given $(n, k) = (17, 2)$, where $n$ can be factorized as $n = 17$. Since $k = 2$, $n$ is not a composite number that can be factorized using only integers less than or equal to $k + 1 = 3$. The null-cascade graph consists of $\tau = 8$ circulant matrices, as*

$$\boldsymbol{W}^{(0)} = \mathrm{circ}\big(\tfrac{1}{2-2\cos(2\pi\cdot(5/17))}, 0, 0, 0, 0, \tfrac{-2\cos(2\pi\cdot(5/17))}{2-2\cos(2\pi\cdot(5/17))}, 0, 0, 0, 0, \tfrac{1}{2-2\cos(2\pi\cdot(5/17))}, 0, 0, 0, 0, 0, 0\big),$$

$$\boldsymbol{W}^{(1)} = \mathrm{circ}\big(\tfrac{1}{2-2\cos(2\pi\cdot(5/17))}, \tfrac{1}{2-2\cos(2\pi\cdot(5/17))}, \tfrac{-2\cos(2\pi\cdot(5/17))}{2-2\cos(2\pi\cdot(5/17))}, 0, 0, 0, 0, 0, 0, 0, 0, 0, 0, 0, 0, 0, 0\big),$$

$$\boldsymbol{W}^{(2)} = \mathrm{circ}\big(\tfrac{-2\cos(2\pi\cdot(6/17))}{2-2\cos(2\pi\cdot(6/17))}, 0, 0, \tfrac{1}{2-2\cos(2\pi\cdot(6/17))}, 0, 0, 0, 0, 0, 0, 0, 0, 0, \tfrac{1}{2-2\cos(2\pi\cdot(6/17))}, 0, 0\big),$$

$$\boldsymbol{W}^{(3)} = \mathrm{circ}\big(\tfrac{-2\cos(2\pi\cdot(6/17))}{2-2\cos(2\pi\cdot(6/17))}, \tfrac{1}{2-2\cos(2\pi\cdot(6/17))}, 0, 0, 0, 0, 0, 0, 0, 0, 0, 0, 0, 0, 0, 0, \tfrac{1}{2-2\cos(2\pi\cdot(6/17))}\big),$$

$$\boldsymbol{W}^{(4)} = \mathrm{circ}\big(\tfrac{-2\cos(2\pi\cdot(6/17))}{2-2\cos(2\pi\cdot(6/17))}, 0, \tfrac{1}{2-2\cos(2\pi\cdot(6/17))}, 0, 0, 0, 0, 0, 0, 0, 0, 0, 0, 0, 0, \tfrac{1}{2-2\cos(2\pi\cdot(6/17))}, 0\big),$$

$$\boldsymbol{W}^{(5)} = \mathrm{circ}\big(\tfrac{-2\cos(2\pi\cdot(8/17))}{2-2\cos(2\pi\cdot(8/17))}, 0 \tfrac{1}{2-2\cos(2\pi\cdot(8/17))}, 00, 0, 0, 0, 0, 0, 0, 0, 0, 0, \tfrac{1}{2-2\cos(2\pi\cdot(8/17))}, 0\big),$$

$$\boldsymbol{W}^{(6)} = \mathrm{circ}\big(\tfrac{-2\cos(2\pi\cdot(8/17))}{2-2\cos(2\pi\cdot(8/17))}, \tfrac{1}{2-2\cos(2\pi\cdot(8/17))}, 0, 0, 0, 0, 0, 0, 0, 0, 0, 0, 0, 0, 0, \tfrac{1}{2-2\cos(2\pi\cdot(8/17))}\big),$$

$$\boldsymbol{W}^{(7)} = \mathrm{circ}\big(\tfrac{-2\cos(2\pi\cdot(7/17))}{2-2\cos(2\pi\cdot(7/17))}, \tfrac{1}{2-2\cos(2\pi\cdot(7/17))}, 0, 0, 0, 0, 0, 0, 0, 0, 0, 0, 0, 0, 0, \tfrac{1}{2-2\cos(2\pi\cdot(7/17))}\big),$$

*where null responses for each $\ell \in \{0, 1, \dots, 7\}$ are obtained at the following frequency indices:*

$\mathcal{N}^{(0)} \ni \{1, \frac{12}{5}, \frac{22}{5}, \frac{29}{5}, \frac{39}{5}, \frac{46}{5}, \frac{56}{5}, \frac{63}{5}, \frac{73}{5}, 16\}$,  *(Step 2: SNF with $n = 17$, $c_0 = 2$, $m_0 = 5$, $(q_{0,1}, q_{0,1}^*) = (5, 12)$),*

$\mathcal{N}^{(1)} \ni \{5, 12\}$  *(Step 2: SNF with $n = 17$, $c_1 = 2$, $m_1 = 1$, $(q_{1,1}, q_{1,1}^*) = (5, 12)$),*

$\mathcal{N}^{(2)} \ni \{2, \frac{11}{3}, \frac{23}{3}, \frac{28}{3}, \frac{40}{3}, 15\}$  *(Step 2: SNF with $n = 17$, $c_2 = 2$, $m_2 = 3$, $(q_{2,1}, q_{2,1}^*) = (6, 11)$),*

$\mathcal{N}^{(3)} \ni \{6, 11\}$  *(Step 2: SNF with $n = 17$, $c_3 = 2$, $m_3 = 1$, $(q_{3,1}, q_{3,1}^*) = (6, 11)$),*

$\mathcal{N}^{(4)} \ni \{3, \frac{11}{2}, \frac{23}{2}, 14\}$  *(Step 2: SNF with $n = 17$, $c_4 = 2$, $m_4 = 2$, $(q_{4,1}, q_{4,1}^*) = (6, 11)$),*

$\mathcal{N}^{(5)} \ni \{4, \frac{9}{2}, \frac{25}{2}, 13\}$  *(Step 2: SNF with $n = 17$, $c_5 = 2$, $m_5 = 2$, $(q_{5,1}, q_{5,1}^*) = (8, 9)$),*

$\mathcal{N}^{(6)} \ni \{8, 9\}$  *(Step 2: SNF with $n = 17$, $c_6 = 2$, $m_6 = 1$, $(q_{6,1}, q_{6,1}^*) = (8, 9)$),*

$\mathcal{N}^{(7)} \ni \{7, 10\}$  *(Step 2: SNF with $n = 17$, $c_7 = 2$, $m_7 = 1$, $(q_{7,1}, q_{7,1}^*) = (7, 10)$).*

*Combining the above four null frequency sets satisfies* (6)*, confirming that Example 5 satisfies finite-time convergence.*

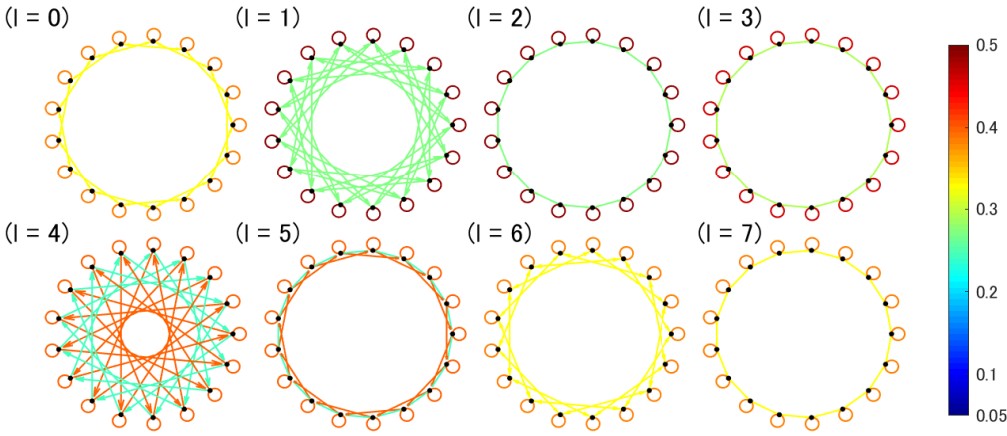

Figure 13: Mixing matrices of null-cascade graph with $(n, k) = (17, 2)$. Each graph consists of $n = 17$ nodes, depicted as black dots arranged in a circle and interconnected by lines. The colors of these lines indicate the mixing weights between nodes in $\boldsymbol{W}^{(\ell)}$.

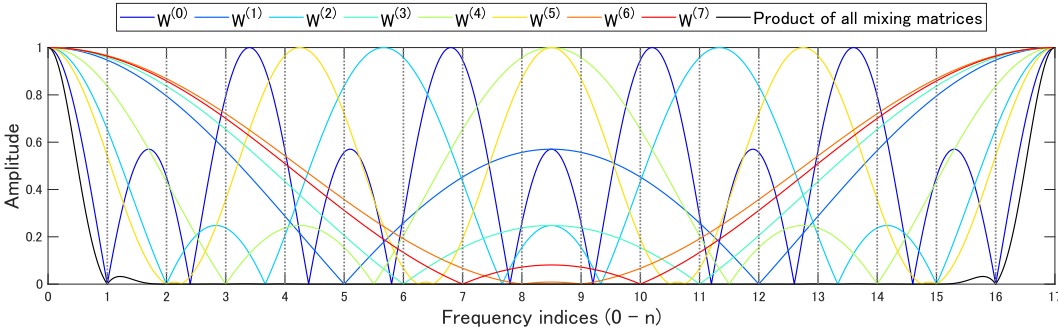

Figure 14: Frequency characteristics (amplitude) of each sequence.

**Example 6** (Null-cascade graph). *Given $(n, k) = (30, 2)$, where $n$ can be factorized as $n = 30 = 5 \times 3 \times 2$. Since $k = 2$, $n$ is not a composite number that can be factorized using only integers less than or equal to $k + 1 = 3$. The null-cascade graph consists of $\tau = 4$ circulant matrices, as*

$$\boldsymbol{W}^{(0)} = \operatorname{circ}(\tfrac{1}{3}, \underbrace{0, \ldots, 0}_{9 \text{ zeros}}, \tfrac{1}{3}, \underbrace{0, \ldots, 0}_{9 \text{ zeros}}, \tfrac{1}{3}, \underbrace{0, \ldots, 0}_{9 \text{ zeros}}),$$

$$\boldsymbol{W}^{(1)} = \operatorname{circ}(\tfrac{1}{2}, \underbrace{0, \ldots, 0}_{4 \text{ zeros}}, \tfrac{1}{2}, \underbrace{0, \ldots, 0}_{4 \text{ zeros}}, \underbrace{0, \ldots, 0}_{20 \text{ zeros}}),$$

$$\boldsymbol{W}^{(2)} = \operatorname{circ}(\tfrac{-2\cos(2\pi \cdot (12/30))}{2 - 2\cos(2\pi \cdot (12/30))}, 0, \tfrac{1}{2 - 2\cos(2\pi \cdot (12/30))}, \underbrace{0, \ldots, 0}_{25 \text{ zeros}}, \tfrac{1}{2 - 2\cos(2\pi \cdot (12/30))}, 0),$$

$$\boldsymbol{W}^{(3)} = \operatorname{circ}(\tfrac{-2\cos(2\pi \cdot (12/30))}{2 - 2\cos(2\pi \cdot (12/30))}, \tfrac{1}{2 - 2\cos(2\pi \cdot (12/30))}, \underbrace{0, \ldots, 0}_{27 \text{ zeros}}, \tfrac{1}{2 - 2\cos(2\pi \cdot (12/30))}),$$

*where null responses for each $\ell \in \{0, 1, 2, 3\}$ are obtained at the following frequency indices:*

$$\mathcal{N}^{(0)} \ni \{1, 2, 4, 5, 7, 8, 10, 11, 13, 14, 16, 17, 19, 20, 22, 23, 25, 26, 28, 29\},$$

*(Step 1: MAF with $n = 30, c_0 = 2, m_0 = 10$,*

$$\mathcal{N}^{(1)} \ni \{3, 9, 15, 21, 27\}, \quad \text{(Step 1: MAF with } n = 30, c_1 = 2, m_1 = 5,$$

$$\mathcal{N}^{(2)} \ni \{6, 9, 21, 24\}, \quad \text{(Step 2: SNF with } n = 30, c_2 = 2, m_2 = 2, (q_{2,1}, q_{2,1}^*) = (12, 18)),$$

$$\mathcal{N}^{(3)} \ni \{12, 18\}, \quad \text{(Step 2: SNF with } n = 30, c_3 = 2, m_3 = 1, (q_{3,1}, q_{3,1}^*) = (12, 18)).$$

*Combining the above four null frequency sets satisfies* (6), *confirming that Example 6 satisfies finite-time convergence.*

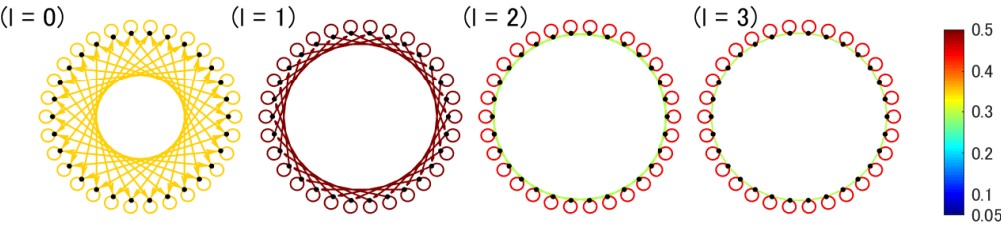

Figure 15: Mixing matrices of null-cascade graph with $(n, k) = (30, 2)$. Each graph consists of $n = 30$ nodes, depicted as black dots arranged in a circle and interconnected by lines. The colors of these lines indicate the mixing weights between nodes in $\boldsymbol{W}^{(\ell)}$.

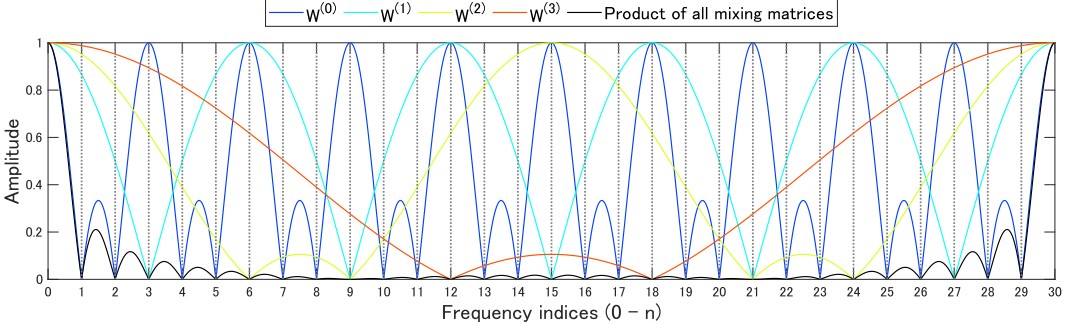

Figure 16: Frequency characteristics (amplitude) of each sequence.

**Example 7** (Null-cascade graph). *Given $(n, k) = (60, 4)$, where $n$ can be factorized as $n = 60 = 5 \times 4 \times 3$. Since $k = 4$, $n$ is a composite number that can be factorized using integers less than or equal to $k + 1 = 5$. The null-cascade graph consists of $\tau = 3$ circulant matrices, as*

$$\boldsymbol{W}^{(0)} = \text{circ}(\tfrac{1}{5}, \underbrace{0, \ldots, 0}_{11 \text{ zeros}}, \tfrac{1}{5}, \underbrace{0, \ldots, 0}_{11 \text{ zeros}}, \tfrac{1}{5}, \underbrace{0, \ldots, 0}_{11 \text{ zeros}}, \tfrac{1}{5}, \underbrace{0, \ldots, 0}_{11 \text{ zeros}}, \tfrac{1}{5}, \underbrace{0, \ldots, 0}_{11 \text{ zeros}}),$$

$$\boldsymbol{W}^{(1)} = \text{circ}(\tfrac{1}{4}, 0, 0, \tfrac{1}{4}, 0, 0, \tfrac{1}{4}, 0, 0, \tfrac{1}{4}, 0, 0, \underbrace{0, \ldots, 0}_{48 \text{ zeros}}),$$

$$\boldsymbol{W}^{(2)} = \text{circ}(\tfrac{1}{3}, \tfrac{1}{3}, \tfrac{1}{3}, \underbrace{0, \ldots, 0}_{57 \text{ zeros}}),$$

*where null responses for each $\ell \in \{0, 1, 2\}$ are obtained at the following frequency indices*

$$\mathcal{N}^{(0)} \ni \{1, 2, 3, 4, 6, 7, 8, , 9, 11, 12, 13, 14, 16, 17, 18, 19, 21, 22, 23, 24, 26, 27, 28, 29, 31, 32, 33, 34,$$
$$36, 37, 38, 39, 41, 42, 43, 44, 46, 47, 48, 49, 51, 52, 53, 54, 56, 57, 58, 59\},$$

*(Step 1: MAF with $n = 60$, $c_0 = 4$, $m_0 = 12$),*

$$\mathcal{N}^{(1)} \ni \{5, 10, 15, 25, 30, 35, 45, 50, 55\}, \quad \textit{(Step 1: MAF with } n = 60, c_1 = 3, m_1 = 3),$$

$$\mathcal{N}^{(2)} \ni \{20, 40\}, \quad \textit{(Step 1: MAF with } n = 60, c_2 = 2, m_2 = 1).$$

*Combining the above three null frequency sets satisfies* (6)*, confirming that Example 7 satisfies finite-time convergence.*

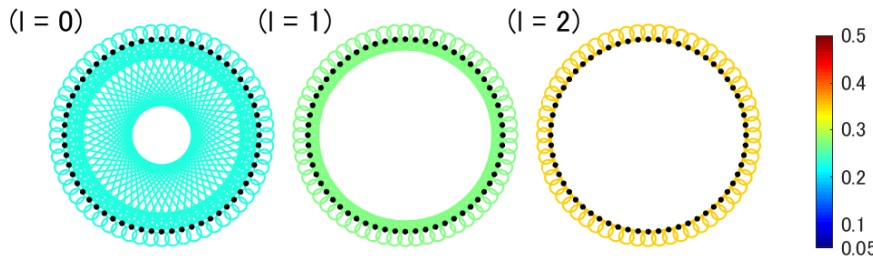

Figure 17: Mixing matrices of null-cascade graph with $(n, k) = (60, 4)$. Each graph consists of $n = 60$ nodes, depicted as black dots arranged in a circle and interconnected by lines. The colors of these lines indicate the mixing weights between nodes in $\boldsymbol{W}^{(\ell)}$.

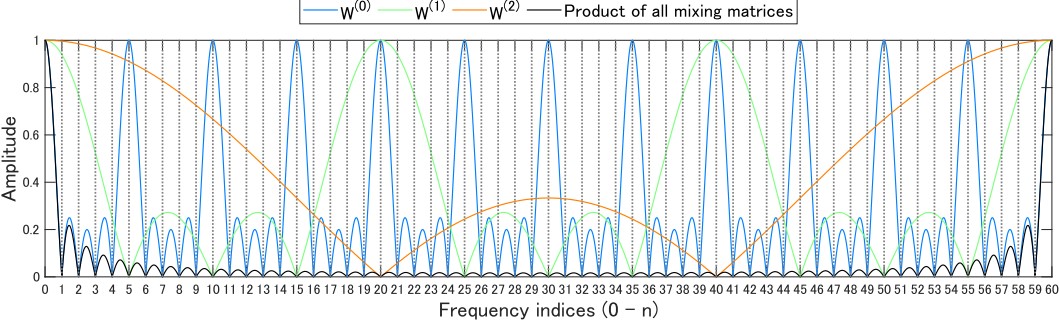

Figure 18: Frequency characteristics (amplitude) of each sequence.

**Example 8** (Null-cascade graph). *Given $(n, k) = (60, 3)$, where $n$ can be factorized as $n = 60 = 5 \times 4 \times 3$. Since $k = 3$, $n$ is not a composite number that can be factorized using only integers less than or equal to $k + 1 = 4$. The null-cascade graph consists of $\tau = 4$ circulant matrices, as*

$$\boldsymbol{W}^{(0)} = \mathrm{circ}(\tfrac{1}{4}, \underbrace{0, \ldots, 0}_{14 \text{ zeros}}, \tfrac{1}{4}, \underbrace{0, \ldots, 0}_{14 \text{ zeros}}, \tfrac{1}{4}, \underbrace{0, \ldots, 0}_{14 \text{ zeros}}, \tfrac{1}{4}, \underbrace{0, \ldots, 0}_{14 \text{ zeros}}).$$

$$\boldsymbol{W}^{(1)} = \mathrm{circ}(\tfrac{1}{3}, 0, 0, 0, 0, \tfrac{1}{3}, 0, 0, 0, 0, \tfrac{1}{3}, 0, 0, 0, 0, \underbrace{0, \ldots, 0}_{45 \text{ zeros}}),$$

$$\boldsymbol{W}^{(2)} = \mathrm{circ}(\tfrac{-2\cos(2\pi\cdot(24/60))}{2 - 2\cos(2\pi\cdot(24/60))}, 0, \tfrac{1}{2 - 2\cos(2\pi\cdot(24/60))}, \underbrace{0, \ldots, 0}_{55 \text{ zeros}}, \tfrac{1}{2 - 2\cos(2\pi\cdot(24/60))}, 0),$$

$$\boldsymbol{W}^{(3)} = \mathrm{circ}(\tfrac{-2\cos(2\pi\cdot(24/60))}{2 - 2\cos(2\pi\cdot(24/60))}, \tfrac{1}{2 - 2\cos(2\pi\cdot(24/60))}, \underbrace{0, \ldots, 0}_{57 \text{ zeros}}, \tfrac{1}{2 - 2\cos(2\pi\cdot(24/60))}),$$

*where null responses for each $\ell \in \{0, 1, 2, 3\}$ are obtained at the following frequency indices*

$$\mathcal{N}^{(0)} \ni \{1, 2, 3, 5, 6, 7, 9, 10, 11, 13, 14, 15, 17, 18, 19, 21, 22, 23, 25, 26, 27, 29, 30, 31, 33, 34, 35,$$
$$37, 38, 39, 41, 42, 43, 45, 46, 47, 49, 50, 51, 53, 54, 55, 57, 58, 59\},$$

*(Step 1: MAF with $n = 60$, $c_0 = 3$, $m_0 = 15$),*

$$\mathcal{N}^{(1)} \ni \{4, 8, 16, 20, 28, 32, 40, 44, 52, 56\}, \qquad \textit{(Step 1: MAF with } n = 60, c_1 = 2, m_1 = 5\textit{)},$$

$$\mathcal{N}^{(2)} \ni \{12, 18, 42, 48\}, \qquad \textit{(Step 2: SNF with } n = 60, c_2 = 2, m_2 = 2, (q_{2,1}, q_{2,1}^*) = (24, 36)\textit{)},$$

$$\mathcal{N}^{(3)} \ni \{24, 36\}, \qquad \textit{(Step 2: SNF with } n = 60, c_3 = 2, m_3 = 1, (q_{3,1}, q_{3,1}^*) = (24, 36)\textit{)}.$$

*Combining the above four null frequency sets satisfies* (6)*, confirming that Example 8 satisfies finite-time convergence.*

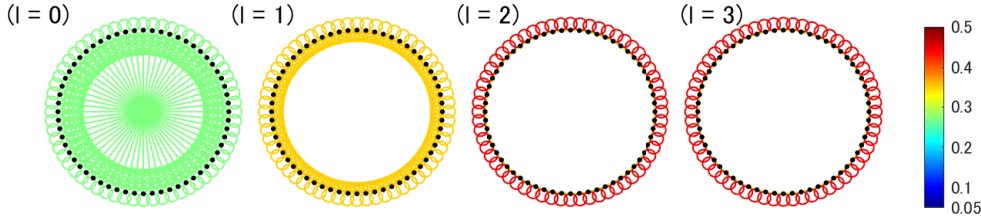

Figure 19: Mixing matrices of null-cascade graph with $(n, k) = (60, 3)$. Each graph consists of $n = 60$ nodes, depicted as black dots arranged in a circle and interconnected by lines. The colors of these lines indicate the mixing weights between nodes in $\boldsymbol{W}^{(\ell)}$.

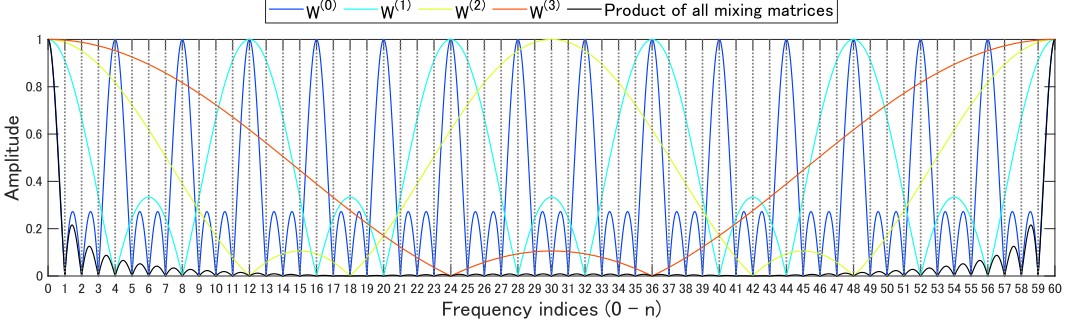

Figure 20: Frequency characteristics (amplitude) of each sequence.

**Example 9** (Null-cascade graph). *Given $(n, k) = (60, 2)$, where $n$ can be factorized as $n = 60 = 5 \times 3 \times 2 \times 2$. Since $k = 2$, $n$ is not a composite number that can be factorized using only integers less than or equal to $k + 1 = 3$. The null-cascade graph consists of $\tau = 5$ circulant matrices, as*

$$\boldsymbol{W}^{(0)} = \operatorname{circ}(\tfrac{1}{3}, \underbrace{0, \ldots, 0}_{19 \text{ zeros}}, \tfrac{1}{3}, \underbrace{0, \ldots, 0}_{19 \text{ zeros}}, \tfrac{1}{3}, \underbrace{0, \ldots, 0}_{19 \text{ zeros}}),$$

$$\boldsymbol{W}^{(1)} = \operatorname{circ}(\tfrac{1}{2}, \underbrace{0, \ldots, 0}_{9 \text{ zeros}}, \tfrac{1}{2}, \underbrace{0, \ldots, 0}_{9 \text{ zeros}}, \underbrace{0, \ldots, 0}_{40 \text{ zeros}}),$$

$$\boldsymbol{W}^{(2)} = \operatorname{circ}(\tfrac{1}{2}, 0, 0, 0, 0, \tfrac{1}{2}, 0, 0, 0, 0, \underbrace{0, \ldots, 0}_{50 \text{ zeros}}),$$

$$\boldsymbol{W}^{(3)} = \operatorname{circ}(\tfrac{-2\cos(2\pi \cdot (24/60))}{2 - 2\cos(2\pi \cdot (24/60))}, 0, \tfrac{1}{2 - 2\cos(2\pi \cdot (24/60))}, \underbrace{0, \ldots, 0}_{55 \text{ zeros}} \tfrac{1}{2 - 2\cos(2\pi \cdot (24/60))}, 0),$$

$$\boldsymbol{W}^{(4)} = \operatorname{circ}(\tfrac{-2\cos(2\pi \cdot (24/60))}{2 - 2\cos(2\pi \cdot (24/60))}, \tfrac{1}{2 - 2\cos(2\pi \cdot (24/60))}, \underbrace{0, \ldots, 0}_{57 \text{ zeros}}, \tfrac{1}{2 - 2\cos(2\pi \cdot (24/60))}),$$

*where null responses for each $\ell \in \{0, 1, 2, 3, 4\}$ are obtained at the following frequency indices*

$$\mathcal{N}^{(0)} \ni \{1, 2, 4, 5, 7, 8, 10, 11, 13, 14, 16, 17, 19, 20, 22, 23, 25, 26, 28, 29, 31, 32, 34, 35, 37, 38, 40,$$
$$41, 43, 44, 46, 47, 49, 50, 52, 53, 55, 56, 58, 59\}, \quad \textit{(Step 1: MAF with } n = 60, c_0 = 2, m_0 = 20\textit{)},$$

$$\mathcal{N}^{(1)} \ni \{3, 9, 15, 21, 27, 33, 39, 45, 51, 57\}, \quad \textit{(Step 1: MAF with } n = 60, c_1 = 1, m_1 = 10\textit{)},$$

$$\mathcal{N}^{(2)} \ni \{6, 18, 30, 42, 54\}, \quad \textit{(Step 1: MAF with } n = 60, c_2 = 1, m_2 = 5\textit{)},$$

$$\mathcal{N}^{(3)} \ni \{12, 18, 42, 48\}, \quad \textit{(Step 2: SNF with } n = 60, c_3 = 2, m_3 = 2, (q_{3,1}, q_{3,1}^*) = (24, 36)\textit{)},$$

$$\mathcal{N}^{(4)} \ni \{24, 36\}, \quad \textit{(Step 2: SNF with } n = 60, c_4 = 2, m_4 = 1, (q_{4,1}, q_{4,1}^*) = (24, 36)\textit{)}.$$

*Combining the above five null frequency sets satisfies* (6)*, confirming that Example 9 satisfies finite-time convergence.*

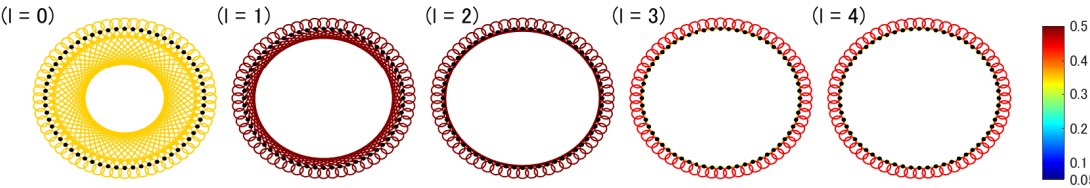

Figure 21: Mixing matrices of null-cascade graph with $(n, k) = (60, 2)$. Each graph consists of $n = 60$ nodes, depicted as black dots arranged in a circle and interconnected by lines. The colors of these lines indicate the mixing weights between nodes in $\boldsymbol{W}^{(\ell)}$.

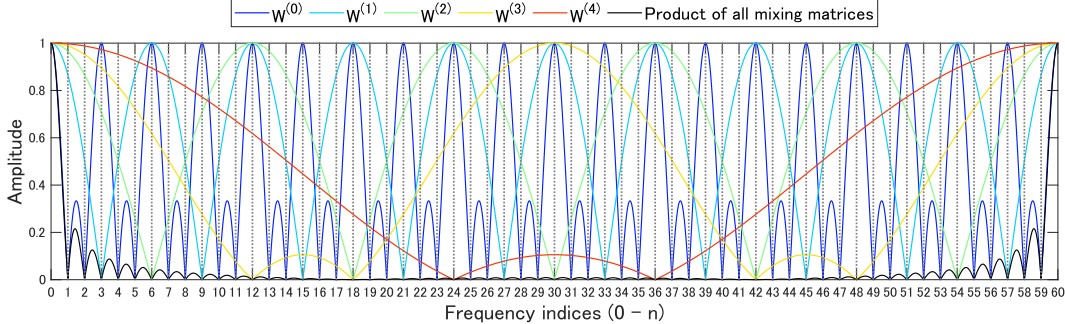

Figure 22: Frequency characteristics (amplitude) of each sequence.

# C   Simplified algorithm to construct base-$(k+1)$ graph

This section consists of the following:

**(i) Interpretation of the base-$(k+1)$ graph.** We summarized our interpretation of the base-$(k+1)$ graph based on the principles underlying the 1-peer exponential graph in Sec. C.1. We found that: i) applying the simple base-$(k+1)$ graph , which is included in the base-$(k+1)$ graph, effectively nullifies frequencies except for those that are multiples of a specific number, and ii) the remaining non-zero frequencies can be eliminated by utilizing circulant matrices employing MAFs with an appropriate choice of length $c_\ell$ and expansion order $m_\ell$.

**(ii) Simplified algorithm to construct the base-$(k+1)$ graph**: Based on the insights in Sec. C.1, a new and simple algorithm to compute mixing matrices in the base-$(k+1)$ graph is developed in Sec. C.2.

## C.1   Our interpretation of the base-$(k+1)$ graph

### (i-a) Brief introduction of the simple base-$(k+1)$ graph

First, we briefly explain the simple base-$(k+1)$ graph, which is included in the base-$(k+1)$ graph. Since the algorithm to compute mixing matrices in the simple base-$(k+1)$ graph is complex (see Alg. 2 in [34]), we aim to provide an intuitive explanation of the simple base-$(k+1)$ graph through an illustrative example:

**Example 10** (Simple base-$(k+1)$ graph with $(n, k) = (3, 1)$). *First, let us decompose $n$ using the power of $(k + 1)$. Consider the case when $(n, k) = (3, 1)$. Then, $n$ can be decomposed as $n = 3 = (1 + 1)^1 + (1 + 1)^0 = 2^1 + 2^0$, resulting in the generation of two subgroups:* $\{\underbrace{\{\mathbf{x}_1, \mathbf{x}_2\}}_{S_1:2^1 \ nodes}, \ \underbrace{\{\mathbf{x}_3\}}_{S_2:2^0 \ node} \ \}$*. The simple base-$(k+1)$ graph with $(n, k) = (3, 1)$ consists of $\tau = 3$ mixing matrices $\{\mathbf{W}^{(0)}, \mathbf{W}^{(1)}, \mathbf{W}^{(2)}\}$:*

$$\begin{bmatrix} \frac{\mathbf{x}_1+\mathbf{x}_2+\mathbf{x}_3}{3} \\ \frac{\mathbf{x}_1+\mathbf{x}_2+\mathbf{x}_3}{3} \\ \frac{\mathbf{x}_1+\mathbf{x}_2+\mathbf{x}_3}{3} \end{bmatrix} = \overbrace{\begin{bmatrix} \frac{1}{2} & \frac{1}{2} & 0 \\ \frac{1}{2} & \frac{1}{2} & 0 \\ 0 & 0 & 1 \end{bmatrix}}^{\mathbf{W}^{(2)}} \overbrace{\begin{bmatrix} 1 & 0 & 0 \\ 0 & \frac{1}{3} & \frac{2}{3} \\ 0 & \frac{2}{3} & \frac{1}{3} \end{bmatrix}}^{\mathbf{W}^{(1)}} \overbrace{\begin{bmatrix} \frac{1}{2} & \frac{1}{2} & 0 \\ \frac{1}{2} & \frac{1}{2} & 0 \\ 0 & 0 & 1 \end{bmatrix}}^{\mathbf{W}^{(0)}} \begin{bmatrix} \mathbf{x}_1 \\ \mathbf{x}_2 \\ \mathbf{x}_3 \end{bmatrix}.$$

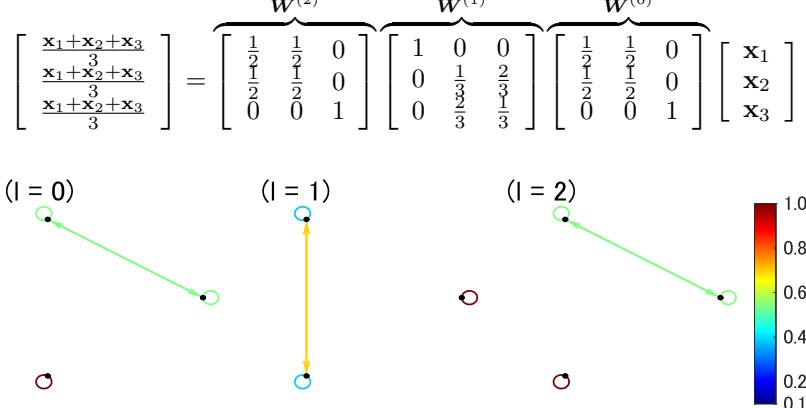

Figure 23: Mixing matrices of simple base-$(k+1)$ graph with $(n, k) = (3, 1)$. Each graph consists of $n = 3$ nodes, depicted as black dots arranged in a circle and interconnected by lines. The colors of these lines indicate the mixing weights between nodes in $\mathbf{W}^{(\ell)}$.

**Example 11** (Simple base-$(k+1)$ graph with $(n,k) = (5,3)$). *First, let us decompose $n$ using the power of $(k+1)$. Consider the case when $(n,k) = (5,3)$. Then, $n$ can be decomposed as $n = 5 = (3+1)^1 + (3+1)^0 = 4^1 + 4^0$, resulting in the generation of two subgroups: $\{\underbrace{\{\mathbf{x}_1,\mathbf{x}_2,\mathbf{x}_3,\mathbf{x}_4\}}_{S_1:4^1\ nodes}, \underbrace{\{\mathbf{x}_5\}}_{S_2:4^0\ node}\}$. The simple base-$(k+1)$ graph with $(n,k) = (5,3)$ consists of $\tau = 3$*
*mixing matrices $\{\boldsymbol{W}^{(0)}, \boldsymbol{W}^{(1)}, \boldsymbol{W}^{(2)}\}$:*

$$
\begin{bmatrix}
\frac{x_1+x_2+x_3+x_4+x_5}{5} \\
\frac{x_1+x_2+x_3+x_4+x_5}{5} \\
\frac{x_1+x_2+x_3+x_4+x_5}{5} \\
\frac{x_1+x_2+x_3+x_4+x_5}{5} \\
\frac{x_1+x_2+x_3+x_4+x_5}{5}
\end{bmatrix}
=
\overbrace{\begin{bmatrix}
\frac{1}{4} & \frac{1}{4} & \frac{1}{4} & \frac{1}{4} & 0 \\
\frac{1}{4} & \frac{1}{4} & \frac{1}{4} & \frac{1}{4} & 0 \\
\frac{1}{4} & \frac{1}{4} & \frac{1}{4} & \frac{1}{4} & 0 \\
\frac{1}{4} & \frac{1}{4} & \frac{1}{4} & \frac{1}{4} & 0 \\
0 & 0 & 0 & 0 & 1
\end{bmatrix}}^{\boldsymbol{W}^{(2)}}
\overbrace{\begin{bmatrix}
\frac{1}{3} & \frac{1}{3} & \frac{1}{3} & 0 & 0 \\
\frac{1}{3} & \frac{1}{3} & \frac{1}{3} & 0 & 0 \\
\frac{1}{3} & \frac{1}{3} & \frac{1}{3} & 0 & 0 \\
0 & 0 & 0 & \frac{1}{5} & \frac{4}{5} \\
0 & 0 & 0 & \frac{4}{5} & \frac{1}{5}
\end{bmatrix}}^{\boldsymbol{W}^{(1)}}
\overbrace{\begin{bmatrix}
\frac{1}{4} & \frac{1}{4} & \frac{1}{4} & \frac{1}{4} & 0 \\
\frac{1}{4} & \frac{1}{4} & \frac{1}{4} & \frac{1}{4} & 0 \\
\frac{1}{4} & \frac{1}{4} & \frac{1}{4} & \frac{1}{4} & 0 \\
\frac{1}{4} & \frac{1}{4} & \frac{1}{4} & \frac{1}{4} & 0 \\
0 & 0 & 0 & 0 & 1
\end{bmatrix}}^{\boldsymbol{W}^{(0)}}
\begin{bmatrix}
x_1 \\ x_2 \\ x_3 \\ x_4 \\ x_5
\end{bmatrix}
$$

*Note that this example corresponds to Fig. 2 in Sec. 5.*

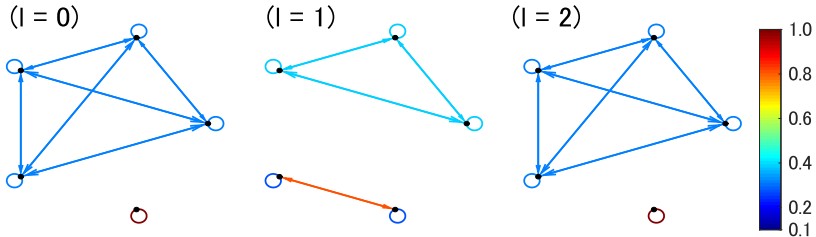

Figure 24: Mixing matrices of simple base-$(k+1)$ graph with $(n,k) = (5,3)$. Each graph consists of $n = 5$ nodes, depicted as black dots arranged in a circle and interconnected by lines. The colors of these lines indicate the mixing weights between nodes in $\boldsymbol{W}^{(\ell)}$.

It is important to note that, in the simple base-$(k+1)$ graph , the use of mixing matrices is not limited to circulant matrices. Thus, the commutativity of the $\tau$ mixing matrices does not hold.

Examples with other $(n,k)$ configurations are illustrated in Sec. C.3. The simple base-$(k+1)$ graph, included in the base-$(k+1)$ graph, is basically structured by three steps: (1a): intra-subgroup averaging within each subgroup using the $k$-peer hypercube graph, which is implemented by $\boldsymbol{W}^{(0)}$, (2): inter-subgroup mixing of the models across subgroups, which is implemented by $\boldsymbol{W}^{(1)}$, and (1b): intra-subgroup averaging using the $k$-peer hypercube graph, which is implemented by $\boldsymbol{W}^{(2)}$.

Although the simple base-$(k+1)$ graph achieves finite-time convergence for any $(n,k)$ combination, two issues remain: i) the periodic interval $\tau$ tends to be large, mainly because the twice repetition of intra-subgroup averaging, resulting in $\mathcal{O}(2\lceil \log_{k+1}(n) \rceil)$ [4], and ii) commutativity among the mixing matrices does not hold.

**(i-b) Our interpretation of the base-$(k+1)$ graph**

Next, we delve into our interpretation of base-$(k+1)$ graph. The algorithm currently used to compute mixing matrices within the base-$(k+1)$ graph, including the simple base-$(k+1)$ graph, is complex (refer to Alg. 3 in [34]). Our primary aim in this section is to develop a new algorithm in the following subsection that will calculate mixing matrices in the base-$(k+1)$ graph. This new algorithm will be based on the principles underlying the 1-peer exponential graph in Sec. 3, presenting a different formulation of base-$(k+1)$ graph. As an initial step, we explain our interpretation regarding base-$(k+1)$ graph in detail below.

Let us recall that the simple base-$(k+1)$ graph ensures finite-time convergence for any $(n,k)$ configuration. The base-$(k+1)$ graph is employed to average local models within each subgroup. Suppose that $n$ can be factorized as $n = \mu_1\mu_2$, which results in the formation of $\mu_1$ subgroups, each containing $\mu_2$ nodes. Assuming that local models within

---

[4]The original paper [34] states the order of the periodic interval $\tau$ as $\mathcal{O}(\log_{k+1}(n))$. This representation remains correct since coefficients multiplied by the main component $\log_{k+1}(n)$ are general. However, we apply a coefficient of 2 to estimate $\tau$ more accurately, reflecting the twice iteration of intra-subgroup averaging.

each subgroup are averaged/identified using the simple base-$(k+1)$ graph, this results in $\{\underbrace{\tilde{\mathbf{x}}_1, \tilde{\mathbf{x}}_1, \ldots, \tilde{\mathbf{x}}_1}_{\mu_2 \text{ nodes}}, \underbrace{\tilde{\mathbf{x}}_2, \tilde{\mathbf{x}}_2, \ldots, \tilde{\mathbf{x}}_2}_{\mu_2 \text{ nodes}}, \cdots, \underbrace{\tilde{\mathbf{x}}_{\mu_1}, \tilde{\mathbf{x}}_{\mu_1}, \ldots, \tilde{\mathbf{x}}_{\mu_1}}_{\mu_2 \text{ nodes}}\}$. This is equivalent to a permutation that repeats $\mu_1$-length sequence $\{\tilde{\mathbf{x}}_1, \tilde{\mathbf{x}}_2, \ldots, \tilde{\mathbf{x}}_{\mu_1}\}$ by $\mu_2$ times, referred to as repeated sequence. The arrangement appears as follows: $\{\underbrace{\tilde{\mathbf{x}}_1, \tilde{\mathbf{x}}_2, \ldots, \tilde{\mathbf{x}}_{\mu_1}}_{1 \text{ time}}, \underbrace{\tilde{\mathbf{x}}_1, \tilde{\mathbf{x}}_2, \ldots, \tilde{\mathbf{x}}_{\mu_1}}_{2 \text{ times}}, \ldots, \underbrace{\tilde{\mathbf{x}}_1, \tilde{\mathbf{x}}_2, \ldots, \tilde{\mathbf{x}}_{\mu_1}}_{\mu_2 \text{ times}}\}$.

For simple notation, suppose that $\mu_1$-length sequence $\{h(0), h(1), \ldots, h(\mu_1 - 1)\}$ and its $\mu_2$-times repeated sequence $\{h'(0), h'(1), \ldots, h(\mu_1 \mu_2 - 1)\} = \{h(0), h(1), \ldots, h(\mu_1 - 1), h(0), h(1), \ldots, h(\mu_1 - 1), \ldots, h(0), h(1), \ldots, h(\mu_1 - 1)\}$ are given. The DFT of the original sequence $h(i)$ is given by

$$H(i) = \sum_{\psi=0}^{\mu_1 - 1} h(\psi) e^{-2\pi \mathrm{j} i \psi / \mu_1},$$

where $i \in \{0, \ldots, \mu_1 - 1\}$ in $H(i)$ is frequency index. Since the repeated sequence can be equivalently formulated using the original sequence, as $h'(i) = h(i \bmod \mu_1)$. Thus, the DFT of the repeated sequence is given by

$$
\begin{aligned}
H'(i) &= \sum_{\psi=0}^{\mu_1 \mu_2 - 1} h'(\psi) e^{-2\pi \mathrm{j} i \psi / (\mu_1 \mu_2)} \\
&= \sum_{\psi=0}^{\mu_1 \mu_2 - 1} h(\psi \bmod \mu_1) e^{-2\pi \mathrm{j} i \psi / (\mu_1 \mu_2)} \\
&= \sum_{\psi_1=0}^{\mu_2 - 1} \sum_{\psi_2=0}^{\mu_1 - 1} h(\psi_2) e^{-2\pi \mathrm{j} i (\psi_1 \mu_1 + \psi_2) / (\mu_1 \mu_2)} \\
&= \sum_{\psi_2=0}^{\mu_1 - 1} h(\psi_2) e^{-2\pi \mathrm{j} i \psi_2 / (\mu_1 \mu_2)} \underbrace{\sum_{\psi_1=0}^{\mu_2 - 1} e^{-2\pi \mathrm{j} i \psi_1 \mu_1 / (\mu_1 \mu_2)}}_{T_1},
\end{aligned}
$$

where $T_1$ takes $\mu_2$ when $i$ is a multiple of $\mu_1 \mu_2$; otherwise, it is zero. Consequently, we obtain

$$H'(i) = \begin{cases} \mu_2 H(\frac{i}{\mu_2}) & (\text{if } i \bmod \mu_2 = 0) \\ 0 & (\text{otherwise}) \end{cases}, \tag{14}$$

where $i \in \{0, \ldots, \mu_1 \mu_2 - 1\}$ in $H'(i)$ denotes the frequency index. This indicates that many frequency components are nullified if $n = \mu_1 \mu_2$-length sequence is composed of $\mu_2$ times repeats of $\mu_1$-length sequence. This is applicable to the local models through the simple base-$(k+1)$ graph, denoted by $\{\underbrace{\tilde{\mathbf{x}}_1, \tilde{\mathbf{x}}_2, \ldots, \tilde{\mathbf{x}}_{\mu_1}}_{1 \text{ time}}, \ldots, \underbrace{\tilde{\mathbf{x}}_1, \tilde{\mathbf{x}}_2, \ldots, \tilde{\mathbf{x}}_{\mu_1}}_{\mu_2 \text{ times}}\}$. Since this is also a repeated sequence, we can recognize the effect of the simple base-$(k+1)$ graph as null forming except the frequency indices satisfying $(i \bmod \mu_2 = 0)$.

Noticing that the remaining frequencies to be nullified are multiples of $\mu_2$, MAFs with a suitable choice of $c_\ell$-length and $m_\ell$-order expansion prove beneficial. Recall that the null frequency indices with MAFs with $c_\ell$-length and $m_\ell$-order expansion are theoretically analyzed in Sec. 3 and Appendix A, specifically in (13). Moreover, this approach is efficiently utilized in null-cascade graph in Sec. 4, with a dynamic/suitable choice of $\{c_\ell, m_\ell\}$, resulting in the forming of nulls at non-zero frequencies. This strategy is also applicable to nullify the remaining frequencies. In the next subsection, building on the ideas discussed so far, a new and simplified algorithm to construct mixing matrices in the base-$(k+1)$ graph will be developed.

## C.2 Simplified algorithm to construct the base-$(k+1)$ graph

The original algorithm to construct the base-$(k+1)$ graph is outlined in Alg. 3 in [34]; however, it is somewhat complex. Assuming that the algorithm to construct the simple base-$(k+1)$ graph is given

---

**Algorithm 5** New algorithm to construct base-$(k+1)$ graph

---

1: ▷ Given $(n, k)$ and $\ell = 0$

2: ▷ Factorization of $n$ as $n = \prod_{i=1}^{\lambda}(\nu_i + 1)$, where $\nu_1 \leq \cdots \leq \nu_\kappa \leq k \leq \nu_{\kappa+1} \leq \cdots \leq \nu_\lambda$, where $\nu_i \in \mathbb{N}$.

3: ▷ (Step 1) Applying simple base-$(k+1)$ graph to be repeated sequence (which is equivalent to form nulls as in (14))

4: **for** $i = \lambda, ..., \kappa + 1$ **do**

5:    ▷ Generate mixing matrices $\boldsymbol{Z} \in \mathbb{R}_+^{(n \times n) \times \tau_{\mathrm{sb}}}$ to be repeated sequence

6:    $\{\boldsymbol{Z}^{(0)}, \ldots, \boldsymbol{Z}^{(\tau_{\mathrm{sb}}-1)}\} = \mathrm{SimpleBaseGraphRepeatedSequence}(\nu_i + 1, n, k)$

7:    **for** $j = 0, ..., \tau_{\mathrm{sb}} - 1$ **do**

8:      $\boldsymbol{W}^{(\ell)} = \boldsymbol{Z}^{(j)}$

9:      $\ell = \ell + 1$   /* mixing matrix index increment */

10:    **end for**

11: **end for**

12: ▷ Remaining frequency to be eliminated (multiples of $b$ from (14))

13: $b = \prod_{i=\kappa+1}^{\lambda}(\nu_i + 1)$    /* base of frequencies to be nullified */

14: ▷ (Step 2) Null forming using expanded MAFs

15: **for** $i = \kappa, \ldots, 1$ **do**

16:    $(c_\ell, m_\ell) = (\nu_i, \frac{n}{(\nu_i+1)b})$

17:    $h^{(\ell)}(j) = \begin{cases} 1/(c_\ell + 1) & (j = 0, \ldots, c_\ell) \\ 0 & (\text{otherwise}) \end{cases}$

18:    $\boldsymbol{W}^{(\ell)} = \mathrm{circ}([h_{\uparrow m_\ell}^{(\ell)}(0), \ldots, h_{\uparrow m_\ell}^{(\ell)}(n-1)])$

19:    $\ell = \ell + 1$     /* mixing matrix index increment */

20:    $b = b(\nu_i + 1)$    /* base of frequencies to be eliminated */

21: **end for**

---

22: **function** $\{\boldsymbol{Z}^{(0)}, \ldots, \boldsymbol{Z}^{(\tau_{\mathrm{sb}}-1)}\} = \mathrm{SimpleBaseGraphRepeatedSequence}(\nu_j + 1, n, k)$

23: ▷ Generate simple base-$(k+1)$ graph $\boldsymbol{Y} \in \mathbb{R}_+^{((\nu_j+1)\times(\nu_j+1)) \times \tau_{\mathrm{sb}}}$ to make consensus among $(\nu_j + 1)$ nodes.

24: $\{\boldsymbol{Y}^{(0)}, \ldots, \boldsymbol{Y}^{(\tau_{\mathrm{sb}}-1)}\} = \mathrm{SimpleBaseGraph}(\nu_j + 1, k)$     /* Alg. 2 in [34] */

25: ▷ Generate mixing matrices $\boldsymbol{Z}^{(r)} \in \mathbb{R}_+^{(n \times n) \times \tau_{\mathrm{sb}}}$ to be repeated sequence

26: **for** $i = 0, \ldots, \tau_{\mathrm{sb}} - 1$ **do**

27:    $\boldsymbol{Z}^{(i)} = \displaystyle\sum_{q=0}^{n/(\nu_j+1)-1} \left( \left( \boldsymbol{Y}^{(i)} \otimes [\underbrace{0, \ldots, 0}_{q}, 1, \underbrace{0, \ldots, 0}_{n/(\nu_j+1)-q-1}]^\top \right) \otimes [\underbrace{0, \ldots, 0}_{q}, 1, \underbrace{0, \ldots, 0}_{n/(\nu_j+1)-q-1}] \right)$

28: **end for**

---

(Alg. 2 in [34]), a new and simplified algorithm for constructing the base-$(k+1)$ graph is presented, leveraging the insights discussed in Sec. C.1.

Our new algorithm for computing the base-$(k+1)$ graphis detailed in Alg. 5. In line 1, $n$ is factorized into $\kappa (\leq 1)$ components as $n = \prod_{i=0}^{\kappa-1}(c_i + 1)$, where $c_i \in \mathbb{N}$ are ordered as $c_0 \leq \cdots \leq c_{\nu-1} \leq k \leq c_\nu \leq \cdots \leq c_{\kappa-1}$. This factorization is critical to determine the integration strategy of the simple base-$(k+1)$ graph and circulant matrices using MAFs. For $c_i > k$ for $i \in \{\nu, \ldots, \kappa - 1\}$, the simple base-$(k+1)$ graph is applied (step 1). Conversely, for $c_i \leq k$ for $i \in \{0, \ldots, \nu - 1\}$, circulant matrices using MAFs are employed (step 2). Lines 3 to 11 compute mixing matrices based on the simple base-$(k+1)$ graph . Through $\tau_{\mathrm{sb}}$ mixing matrices, a repeated sequence of local models $\{\underbrace{\tilde{\mathbf{x}}_1, \tilde{\mathbf{x}}_2, \ldots, \tilde{\mathbf{x}}_{n/(c_r+1)}}_{1 \text{ time}}), \ldots, \underbrace{\tilde{\mathbf{x}}_1, \tilde{\mathbf{x}}_2, \ldots, \tilde{\mathbf{x}}_{n/(c_r+1)}}_{(c_r+1) \text{ times}}\}$ is generated. The application of the simple base-$(k+1)$ graph is repeated for $c_i > k$ for $i \in \{\nu, \ldots, \kappa - 1\}$. From (14), this cascade of the simple base-$(k+1)$ graph remains non-nullified frequency components $i$, which are multiples of $b = \prod_{i=\nu}^{\kappa-1}(c_i + 1)$ within $i \in \{0, \ldots, n\}$. For nullifying the remaining frequencies in step 2,

circulant matrices employing MAFs with $c_r$-length and expansion order $m_r$ are generated in lines 14 to 21. We used the techniques in the null-cascade graph in Alg. 1.

Although our Alg. 5 diverges significantly from the original algorithm (Alg. 3 in [34]), they generate equivalent mixing matrices. These intriguing outcomes stem from our theoretical analysis of the principles underlying the 1-peer exponential graph.

## C.3 Example illustrations

### C.3.1 Simple base-$(k+1)$ graph

**Example 12** (Simple base-$(k+1)$ graph). *First, let us decompose $n$ using the power of $(k+1)$. Consider the case when $(n, k) = (5, 1)$. Then, $n$ can be decomposed as $n = 5 = 2 \cdot (1+1)^1 + (1+1)^0$, resulting in the generation of three subgroups:* $\{ \underbrace{\{\mathbf{x}_1, \mathbf{x}_2}_{S_1:2^1 \text{ nodes}}, \underbrace{\mathbf{x}_3, \mathbf{x}_4\}}_{S_2:2^1 \text{ nodes}}, \underbrace{\{\mathbf{x}_5\}}_{S_3:2^0 \text{ node}} \}$. *The simple base-$(k+1)$ graph with $(n, k) = (5, 1)$ consists of $\tau = 5$ mixing matrices $\{\mathbf{W}^{(0)}, \ldots, \mathbf{W}^{(4)}\}$:*

$$
\begin{bmatrix}
\frac{\mathbf{x}_1+\mathbf{x}_2+\mathbf{x}_3+\mathbf{x}_4+\mathbf{x}_5}{5} \\
\frac{\mathbf{x}_1+\mathbf{x}_2+\mathbf{x}_3+\mathbf{x}_4+\mathbf{x}_5}{5} \\
\frac{\mathbf{x}_1+\mathbf{x}_2+\mathbf{x}_3+\mathbf{x}_4+\mathbf{x}_5}{5} \\
\frac{\mathbf{x}_1+\mathbf{x}_2+\mathbf{x}_3+\mathbf{x}_4+\mathbf{x}_5}{5} \\
\frac{\mathbf{x}_1+\mathbf{x}_2+\mathbf{x}_3+\mathbf{x}_4+\mathbf{x}_5}{5}
\end{bmatrix}
=
\overbrace{\begin{bmatrix}
\frac{1}{2} & 0 & \frac{1}{2} & 0 & 0 \\
0 & \frac{1}{2} & 0 & \frac{1}{2} & 0 \\
\frac{1}{2} & 0 & \frac{1}{2} & 0 & 0 \\
0 & \frac{1}{2} & 0 & \frac{1}{2} & 0 \\
0 & 0 & 0 & 0 & 1
\end{bmatrix}}^{\mathbf{W}^{(4)}}
\overbrace{\begin{bmatrix}
\frac{1}{2} & \frac{1}{2} & 0 & 0 & 0 \\
\frac{1}{2} & \frac{1}{2} & 0 & 0 & 0 \\
0 & 0 & \frac{1}{2} & \frac{1}{2} & 0 \\
0 & 0 & \frac{1}{2} & \frac{1}{2} & 0 \\
0 & 0 & 0 & 0 & 1
\end{bmatrix}}^{\mathbf{W}^{(3)}}
\overbrace{\begin{bmatrix}
\frac{1}{2} & \frac{1}{2} & 0 & 0 & 0 \\
\frac{1}{2} & \frac{1}{2} & 0 & 0 & 0 \\
0 & 0 & 1 & 0 & 0 \\
0 & 0 & 0 & \frac{1}{5} & \frac{4}{5} \\
0 & 0 & 0 & \frac{4}{5} & \frac{1}{5}
\end{bmatrix}}^{\mathbf{W}^{(2)}}
$$

$$
\cdot
\overbrace{\begin{bmatrix}
\frac{1}{2} & 0 & \frac{1}{2} & 0 & 0 \\
0 & \frac{1}{2} & 0 & \frac{1}{2} & 0 \\
\frac{1}{2} & 0 & \frac{1}{2} & 0 & 0 \\
0 & \frac{1}{2} & 0 & \frac{1}{2} & 0 \\
0 & 0 & 0 & 0 & 1
\end{bmatrix}}^{\mathbf{W}^{(1)}}
\overbrace{\begin{bmatrix}
\frac{1}{2} & \frac{1}{2} & 0 & 0 & 0 \\
\frac{1}{2} & \frac{1}{2} & 0 & 0 & 0 \\
0 & 0 & \frac{1}{2} & \frac{1}{2} & 0 \\
0 & 0 & \frac{1}{2} & \frac{1}{2} & 0 \\
0 & 0 & 0 & 0 & 1
\end{bmatrix}}^{\mathbf{W}^{(0)}}
\begin{bmatrix}
\mathbf{x}_1 \\
\mathbf{x}_2 \\
\mathbf{x}_3 \\
\mathbf{x}_4 \\
\mathbf{x}_5
\end{bmatrix}.
$$

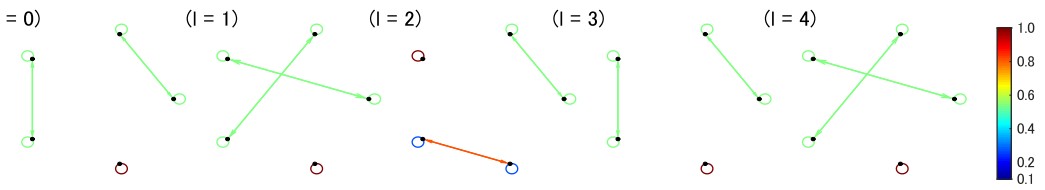

Figure 25: Mixing matrices of simple base-$(k+1)$ graph with $(n, k) = (5, 1)$. Each graph consists of $n = 5$ nodes, depicted as black dots arranged in a circle and interconnected by lines. The colors of these lines indicate the mixing weights between nodes in $\mathbf{W}^{(\ell)}$.

**Example 13** (Simple base-$(k+1)$ graph)**.** *First, let us decompose $n$ using the power of $(k+1)$.*
*Consider the case when $(n,k) = (5,2)$. Then, $n$ can be decomposed as $n = 5 = (2+1)^1 + 2 \cdot (2+1)^0$,*
*resulting in the generation of two subgroups:* $\{\underbrace{\{\mathbf{x}_1, \mathbf{x}_2, \mathbf{x}_3,}_{S_1 : 3^1 \text{ nodes}} \underbrace{\mathbf{x}_4, \mathbf{x}_5\}}_{S_2 : 2 \cdot 3^0 \text{ nodes}} \}$. *The simple base-$(k+1)$*
*graph with $(n,k) = (5,2)$ consists of $\tau = 3$ mixing matrices $\{\mathbf{W}^{(0)}, \ldots, \mathbf{W}^{(2)}\}$:*

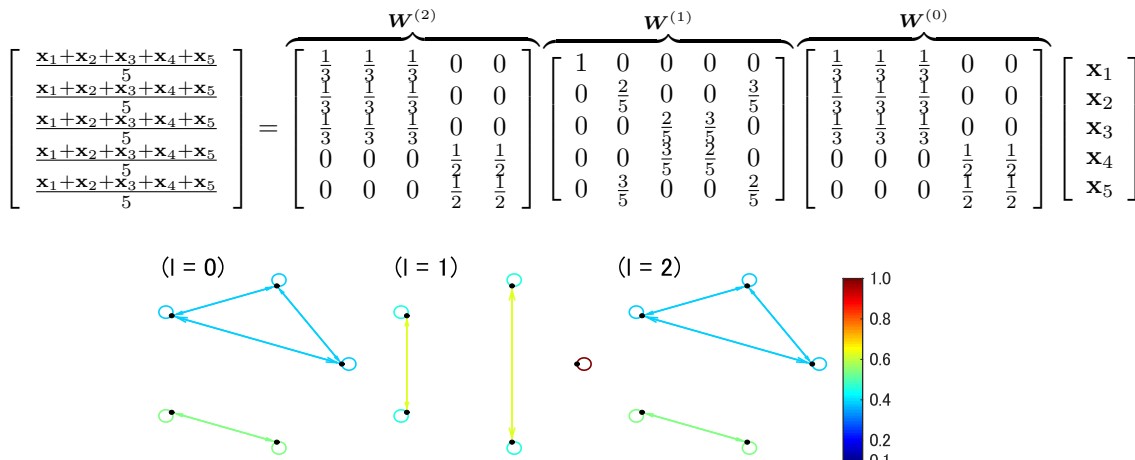

$$
\begin{bmatrix} \frac{\mathbf{x}_1+\mathbf{x}_2+\mathbf{x}_3+\mathbf{x}_4+\mathbf{x}_5}{5} \\ \frac{\mathbf{x}_1+\mathbf{x}_2+\mathbf{x}_3+\mathbf{x}_4+\mathbf{x}_5}{5} \\ \frac{\mathbf{x}_1+\mathbf{x}_2+\mathbf{x}_3+\mathbf{x}_4+\mathbf{x}_5}{5} \\ \frac{\mathbf{x}_1+\mathbf{x}_2+\mathbf{x}_3+\mathbf{x}_4+\mathbf{x}_5}{5} \\ \frac{\mathbf{x}_1+\mathbf{x}_2+\mathbf{x}_3+\mathbf{x}_4+\mathbf{x}_5}{5} \end{bmatrix}
=
\begin{bmatrix} \frac{1}{3} & \frac{1}{3} & \frac{1}{3} & 0 & 0 \\ \frac{1}{3} & \frac{1}{3} & \frac{1}{3} & 0 & 0 \\ \frac{1}{3} & \frac{1}{3} & \frac{1}{3} & 0 & 0 \\ 0 & 0 & 0 & \frac{1}{2} & \frac{1}{2} \\ 0 & 0 & 0 & \frac{1}{2} & \frac{1}{2} \end{bmatrix}
\begin{bmatrix} 1 & 0 & 0 & 0 & 0 \\ 0 & \frac{2}{5} & 0 & 0 & \frac{3}{5} \\ 0 & 0 & \frac{2}{5} & \frac{3}{5} & 0 \\ 0 & 0 & \frac{3}{5} & \frac{2}{5} & 0 \\ 0 & \frac{3}{5} & 0 & 0 & \frac{2}{5} \end{bmatrix}
\begin{bmatrix} \frac{1}{3} & \frac{1}{3} & \frac{1}{3} & 0 & 0 \\ \frac{1}{3} & \frac{1}{3} & \frac{1}{3} & 0 & 0 \\ \frac{1}{3} & \frac{1}{3} & \frac{1}{3} & 0 & 0 \\ 0 & 0 & 0 & \frac{1}{2} & \frac{1}{2} \\ 0 & 0 & 0 & \frac{1}{2} & \frac{1}{2} \end{bmatrix}
\begin{bmatrix} \mathbf{x}_1 \\ \mathbf{x}_2 \\ \mathbf{x}_3 \\ \mathbf{x}_4 \\ \mathbf{x}_5 \end{bmatrix}
$$

Figure 26: Mixing matrices of simple base-$(k+1)$ graph with $(n,k) = (5,2)$. Each graph consists of $n = 5$ nodes, depicted as black dots arranged in a circle and interconnected by lines. The colors of these lines indicate the mixing weights between nodes in $\mathbf{W}^{(\ell)}$.

## C.3.2  Base-$(k+1)$ graph

Following Alg. 5, several illustrations of the base-$(k+1)$ graph are shown.

**Example 14** (Base-$(k+1)$ graph)**.** *First, the factorization of $n$ is given as: $n = 15 = (2+1)\cdot(4+1)$; namely, $\nu_1 = 2, \nu_2 = 4$. Since $k = 2$, $n$ includes a factorization number $\nu_2 = 4$ such that $\nu_i > k$. Following Alg. 5, the simple base-$(k+1)$ graph with $(n, k) = (5, 2)$ is computed. From Example 13, $\{\boldsymbol{Y}^{(i)}\}_{i\in\{0,1,2\}}$ are prepared.*

$$\boldsymbol{Y}^{(0)} = \begin{bmatrix} \frac{1}{3} & \frac{1}{3} & \frac{1}{3} & 0 & 0 \\ \frac{1}{3} & \frac{1}{3} & \frac{1}{3} & 0 & 0 \\ \frac{1}{3} & \frac{1}{3} & \frac{1}{3} & 0 & 0 \\ 0 & 0 & 0 & \frac{1}{2} & \frac{1}{2} \\ 0 & 0 & 0 & \frac{1}{2} & \frac{1}{2} \end{bmatrix}, \quad \boldsymbol{Y}^{(1)} = \begin{bmatrix} 1 & 0 & 0 & 0 & 0 \\ 0 & \frac{2}{5} & 0 & 0 & \frac{3}{5} \\ 0 & 0 & \frac{2}{5} & \frac{3}{5} & 0 \\ 0 & 0 & \frac{3}{5} & \frac{2}{5} & 0 \\ 0 & \frac{3}{5} & 0 & 0 & \frac{2}{5} \end{bmatrix}, \quad \boldsymbol{Y}^{(2)} = \begin{bmatrix} \frac{1}{3} & \frac{1}{3} & \frac{1}{3} & 0 & 0 \\ \frac{1}{3} & \frac{1}{3} & \frac{1}{3} & 0 & 0 \\ \frac{1}{3} & \frac{1}{3} & \frac{1}{3} & 0 & 0 \\ 0 & 0 & 0 & \frac{1}{2} & \frac{1}{2} \\ 0 & 0 & 0 & \frac{1}{2} & \frac{1}{2} \end{bmatrix}.$$

*The base-$(k+1)$ graph consists of $\tau = 4$ circulant matrices as*

$$\boldsymbol{W}^{(0)} = \sum_{q=0}^{2}\left(\left(\boldsymbol{Y}^{(0)} \otimes [\underbrace{0,\dots,0}_{q}, 1, \underbrace{0,\dots,0}_{2-q}]^{\top}\right) \otimes [\underbrace{0,\dots,0}_{q}, 1, \underbrace{0,\dots,0}_{2-q}]\right),$$

$$\boldsymbol{W}^{(1)} = \sum_{q=0}^{2}\left(\left(\boldsymbol{Y}^{(1)} \otimes [\underbrace{0,\dots,0}_{q}, 1, \underbrace{0,\dots,0}_{2-q}]^{\top}\right) \otimes [\underbrace{0,\dots,0}_{q}, 1, \underbrace{0,\dots,0}_{2-q}]\right),$$

$$\boldsymbol{W}^{(2)} = \sum_{q=0}^{2}\left(\left(\boldsymbol{Y}^{(2)} \otimes [\underbrace{0,\dots,0}_{q}, 1, \underbrace{0,\dots,0}_{2-q}]^{\top}\right) \otimes [\underbrace{0,\dots,0}_{q}, 1, \underbrace{0,\dots,0}_{2-q}]\right),$$

$$\boldsymbol{W}^{(3)} = \mathrm{circ}(\tfrac{1}{3}, \tfrac{1}{3}, \tfrac{1}{3}, \underbrace{0,\dots,0}_{12 \text{ zeros}}),$$

*where null responses for each $\ell \in \{0,1,2,3,4\}$ are obtained at following frequency indices*

$$\bigcup_{\ell=0}^{2}\mathcal{N}^{(\ell)} \ni \{1,2,3,4,6,7,8,9,11,12,13,14\},$$

*(Null frequency indices are multiples of 1, excluding multiples of 5, from* (14) *characterizing the frequency response of the simple base-$(k+1)$ graph. )*

$$\mathcal{N}^{(3)} \ni \{5, 10\}, \quad \textit{(MAF with } n = 15, c_3 = 1, m_3 = 1).$$

*Combining the above null frequency sets satisfies* (6)*, confirming that this satisfies finite-time convergence.*

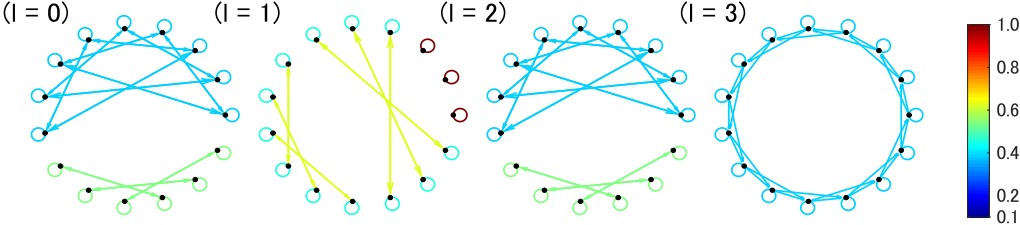

Figure 27: Mixing matrices of the simple base-$(k+1)$ graph with $(n, k) = (15, 2)$. Each graph consists of $n = 15$ nodes, depicted as black dots arranged in a circle and interconnected by lines. The colors of these lines indicate the mixing weights between nodes in $\boldsymbol{W}^{(\ell)}$.

**Example 15** (Base-$(k+1)$ graph). *First, the factorization of $n$ is given as: $n = 30 = (1+1) \cdot (2+1) \cdot (4+1)$; namely, $\nu_1 = 1, \nu_2 = 2, \nu_3 = 4$. Since $k = 2$, $n$ includes a factorization number $\nu_3 = 4$ such that $\nu_i > k$. Following Alg. 5, the simple base-$(k+1)$ graph with $(n, k) = (5, 2)$ is computed. From Example 13, $\{\boldsymbol{Y}^{(i)}\}_{i \in \{0,1,2\}}$ are prepared.*

$$\boldsymbol{Y}^{(0)} = \begin{bmatrix} \frac{1}{3} & \frac{1}{3} & \frac{1}{3} & 0 & 0 \\ \frac{1}{3} & \frac{1}{3} & \frac{1}{3} & 0 & 0 \\ \frac{1}{3} & \frac{1}{3} & \frac{1}{3} & 0 & 0 \\ 0 & 0 & 0 & \frac{1}{2} & \frac{1}{2} \\ 0 & 0 & 0 & \frac{1}{2} & \frac{1}{2} \end{bmatrix}, \quad \boldsymbol{Y}^{(1)} = \begin{bmatrix} 1 & 0 & 0 & 0 & 0 \\ 0 & \frac{2}{5} & 0 & 0 & \frac{3}{5} \\ 0 & 0 & \frac{2}{5} & \frac{3}{5} & 0 \\ 0 & 0 & \frac{3}{5} & \frac{2}{5} & 0 \\ 0 & \frac{3}{5} & 0 & 0 & \frac{2}{5} \end{bmatrix}, \quad \boldsymbol{Y}^{(2)} = \begin{bmatrix} \frac{1}{3} & \frac{1}{3} & \frac{1}{3} & 0 & 0 \\ \frac{1}{3} & \frac{1}{3} & \frac{1}{3} & 0 & 0 \\ \frac{1}{3} & \frac{1}{3} & \frac{1}{3} & 0 & 0 \\ 0 & 0 & 0 & \frac{1}{2} & \frac{1}{2} \\ 0 & 0 & 0 & \frac{1}{2} & \frac{1}{2} \end{bmatrix}.$$

*The base-$(k+1)$ graph consists of $\tau = 4$ circulant matrices as*

$$\boldsymbol{W}^{(0)} = \sum_{q=0}^{5} \left( \left( \boldsymbol{Y}^{(0)} \otimes [\underbrace{0, \ldots, 0}_{q}, 1, \underbrace{0, \ldots, 0}_{5-q}]^{\top} \right) \otimes [\underbrace{0, \ldots, 0}_{q}, 1, \underbrace{0, \ldots, 0}_{5-q}] \right),$$

$$\boldsymbol{W}^{(1)} = \sum_{q=0}^{5} \left( \left( \boldsymbol{Y}^{(1)} \otimes [\underbrace{0, \ldots, 0}_{q}, 1, \underbrace{0, \ldots, 0}_{5-q}]^{\top} \right) \otimes [\underbrace{0, \ldots, 0}_{q}, 1, \underbrace{0, \ldots, 0}_{5-q}] \right),$$

$$\boldsymbol{W}^{(2)} = \sum_{q=0}^{5} \left( \left( \boldsymbol{Y}^{(2)} \otimes [\underbrace{0, \ldots, 0}_{q}, 1, \underbrace{0, \ldots, 0}_{5-q}]^{\top} \right) \otimes [\underbrace{0, \ldots, 0}_{q}, 1, \underbrace{0, \ldots, 0}_{5-q}] \right),$$

$$\boldsymbol{W}^{(3)} = \mathrm{circ}(\tfrac{1}{3}, 0, \tfrac{1}{3}, 0, \tfrac{1}{3}, 0, \underbrace{0, \ldots, 0}_{24 \text{ zeros}}),$$

$$\boldsymbol{W}^{(4)} = \mathrm{circ}(\tfrac{1}{2}, \tfrac{1}{2}, \underbrace{0, \ldots, 0}_{28 \text{ zeros}}),$$

*where null responses for each $\ell \in \{0, 1, 2, 3, 4\}$ are obtained at following frequency indices*

$$\bigcup_{\ell=0}^{2} \mathcal{N}^{(\ell)} \ni \{1, 2, 3, 4, 6, 7, 8, 9, 11, 12, 13, 14, 16, 17, 18, 19, 21, 22, 23, 24, 26, 27, 28, 29\},$$

*(Null frequency indices are multiples of 1, excluding multiples of 5, from (14) characterizing the frequency response of the simple base-$(k+1)$ graph. )*

$$\mathcal{N}^{(3)} \ni \{5, 10, 20, 25\}, \qquad \textit{(MAF with } n = 30, c_3 = 2, m_3 = 2\textit{)}.$$
$$\mathcal{N}^{(4)} \ni \{15\}, \qquad \textit{(MAF with } n = 30, c_4 = 1, m_4 = 1\textit{)}.$$

*Combining the above null frequency sets satisfies (6), confirming that this satisfies finite-time convergence.*

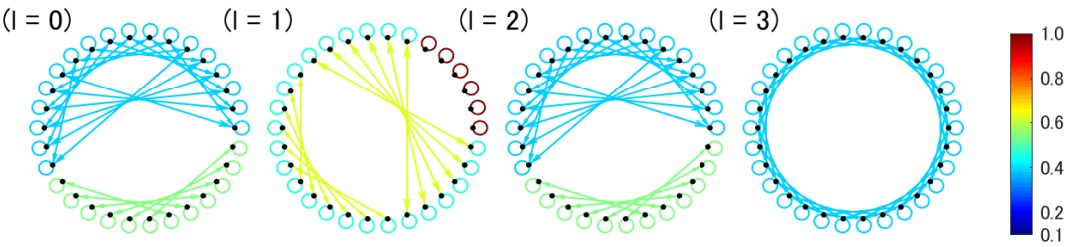

Figure 28: Mixing matrices of simple base-$(k+1)$ graph with $(n, k) = (30, 2)$. Each graph consists of $n = 30$ nodes, depicted as black dots arranged in a circle and interconnected by lines. The colors of these lines indicate the mixing weights between nodes in $\boldsymbol{W}^{(\ell)}$.

# D   A consideration regarding the benefit of commutativity

A brief consideration of commutativity (Definition 4) follows. Reducing the discrepancy between the global model $\overline{\mathbf{x}}^{(r)}$ and local models $\mathbf{x}_i^{(r)}$, specifically, minimizing $\sum_{i=1}^n \|\overline{\mathbf{x}}^{(r)} - \mathbf{x}_i^{(r)}\|$ is critical for accelerating decentralized learning. This term appears explicitly in the convergence analysis of DSGD (implicitly included in Theorem 1). Enforcing commutativity of the graph mixing matrices can reduce this discrepancy.

To illustrate this, we reformulate the update rules of DSGD recursively, as follows:

$$
\begin{aligned}
\mathbf{X}^{(r+1)} &= \mathbf{X}^{(r)} \boldsymbol{W}^{(r)} - \eta \nabla F(\mathbf{X}^{(r)}; \xi^{(r)}) \boldsymbol{W}^{(r)} \\
&= \mathbf{X}^{(r-1)} \boldsymbol{W}^{(r-1)} \boldsymbol{W}^{(r)} - \eta \big( \nabla F(\mathbf{X}^{(r-1)}; \xi^{(r-1)}) \boldsymbol{W}^{(r-1)} \boldsymbol{W}^{(r)} + \nabla F(\mathbf{X}^{(r)}; \xi^{(r)}) \boldsymbol{W}^{(r)} \big) \\
&\;\;\vdots \\
&= \mathbf{X}^{(0)} \boldsymbol{W}^{(0)} \cdots \boldsymbol{W}^{(r)} - \eta \big( \nabla F(\mathbf{X}^{(0)}; \xi^{(0)}) \boldsymbol{W}^{(0)} \cdots \boldsymbol{W}^{(r)} + \cdots + \nabla F(\mathbf{X}^{(r)}; \xi^{(r)}) \boldsymbol{W}^{(r)} \big) \\
&= \mathbf{X}^{(0)} \prod_{s=0}^r \boldsymbol{W}^{(s)} - \eta \sum_{s_1=0}^r \Big( \nabla F(\mathbf{X}^{(s_1)}; \xi^{(s_1)}) \prod_{s_2=s_1}^r \boldsymbol{W}^{(s_2)} \Big).
\end{aligned}
$$

Next, we investigate the property of $\prod_{s_2=s_1}^r \boldsymbol{W}^{(s_2)}$ for finite-time convergent graph with periodic interval $\tau$.

**[Case 1: Commutativity of mixing matrices is satisfied]** (e.g., $k$-peer exponential graph and null-cascade graph)

When commutativity of mixing matrices is satisfied, $\boldsymbol{W}^{(\rho(0))} \boldsymbol{W}^{(\rho(1))} \cdots \boldsymbol{W}^{(\rho(\tau-1))} = \frac{1}{n} \mathbf{1}_n \mathbf{1}_n^\top$ for any permutation $\rho$. The product $\prod_{s_2=s_1}^r \boldsymbol{W}^{(s_2)}$ can be categorized into two cases:

(Case 1a) When $s_1 \leq r - \tau + 1$, $\prod_{s_2=s_1}^r \boldsymbol{W}^{(s_2)} = \frac{1}{n} \mathbf{1}_n \mathbf{1}_n^\top$.

(Case 1b) When $s_1 \geq r - \tau + 2$, $\prod_{s_2=s_1}^r \boldsymbol{W}^{(s_2)} \neq \frac{1}{n} \mathbf{1}_n \mathbf{1}_n^\top$.

These results indicate that no discrepancy among $n$ nodes can occur for terms with index $s_1 \in [0, r-\tau+1]$ (in Case 1a). In contrast, discrepancy among $n$ nodes can occur for terms with index $s_1 \in [r-\tau+2, r]$ (in Case 1b), which leads to degradation in the convergence rate.

**[Case 2: Commutativity of mixing matrices is NOT satisfied]** (e.g., base-$(k+1)$ graph)

When we denote $\mod(r, \tau) = \varsigma$, the product $\prod_{s_2=s_1}^r \boldsymbol{W}^{(s_2)}$ can be categorized into two cases:

(Case 2a) When $s_1 \leq r - \tau - \varsigma + 1$, $\prod_{s_2=s_1}^r \boldsymbol{W}^{(s_2)} = \frac{1}{n} \mathbf{1}_n \mathbf{1}_n^\top$.

(Case 2b) When $s_1 \geq r - \tau - \varsigma + 2$, $\prod_{s_2=s_1}^r \boldsymbol{W}^{(s_2)} \neq \frac{1}{n} \mathbf{1}_n \mathbf{1}_n^\top$.

These results indicate that no discrepancy among $n$ nodes can occur for terms with index $s_1 \in [0, r-\tau-\varsigma+1]$ (in case 2a). In contrast, discrepancy among $n$ nodes can occur for terms with index $s_1 \in [r-\tau-\varsigma+2, r]$ (in case 2b), which leads to degradation in the convergence rate.

Comparing these two cases, the components that contribute to discrepancies among the $n$ nodes tend to be more significant when $\mod(r, \tau) = \varsigma$ is non-zero. Although this difference may be small and therefore difficult to observe empirically, we have theoretically demonstrated the advantage of using commutative mixing matrices.

# E   Convergence rate comparison using various graphs

As a theoretical contribution, the convergence rates of DSGD over various graphs are summarized in Table 3. These rates are derived by substituting the expected spectral gap $p$ and periodic interval $\tau$ specific to each graph into the convergence rate of DSGD in Theorem 1.

Table 3: Comparison of convergence rates of DSGD over various graphs.

| Graphs | Convergence rate of DSGD | Maximum degree $k$ | # of nodes $n$ |
|---|---|---|---|
| Ring [24] | $\mathcal{O}\left(\dfrac{\sigma^2}{n\epsilon^2} + \dfrac{\zeta n^2 + \sigma n}{\epsilon^{3/2}} + \dfrac{n^2}{\epsilon}\right) \cdot Lf_0$ | 2 | $\forall n \in \mathbb{N}$ |
| Torus [24] | $\mathcal{O}\left(\dfrac{\sigma^2}{n\epsilon^2} + \dfrac{\zeta n + \sigma\sqrt{n}}{\epsilon^{3/2}} + \dfrac{n}{\epsilon}\right) \cdot Lf_0$ | 4 | $\forall n \in \mathbb{N}$ |
| Exp. [43] | $\mathcal{O}\left(\dfrac{\sigma^2}{n\epsilon^2} + \dfrac{\zeta \log_2(n) + \sigma\sqrt{\log_2(n)}}{\epsilon^{3/2}} + \dfrac{\log_2(n)}{\epsilon}\right) \cdot Lf_0$ | $\lceil\log_2(n)\rceil$ | $\forall n \in \mathbb{N}$ |
| 1-peer exp. [43] | $\begin{cases}\mathcal{O}\left(\dfrac{\sigma^2}{n\epsilon^2} + \dfrac{\zeta \log_2(n) + \sigma\sqrt{\log_2(n)}}{\epsilon^{3/2}} + \dfrac{\log_2(n)}{\epsilon}\right) \cdot Lf_0 \\ \text{n.a.}\end{cases}$ | 1 | $\begin{cases}\text{A power of } 2 \\ \forall n \in \mathbb{N}\end{cases}$ |
| 1-peer hypercube [31] | $\mathcal{O}\left(\dfrac{\sigma^2}{n\epsilon^2} + \dfrac{\zeta \log_2(n) + \sigma\sqrt{\log_2(n)}}{\epsilon^{3/2}} + \dfrac{\log_2(n)}{\epsilon}\right) \cdot Lf_0$ | 1 | A power of 2 |
| Base-$(k+1)$ [35] | $\mathcal{O}\left(\dfrac{\sigma^2}{n\epsilon^2} + \dfrac{\zeta \tau_{\text{base}} + \sigma\sqrt{\tau_{\text{base}}}}{\epsilon^{3/2}} + \dfrac{\tau_{\text{base}}}{\epsilon}\right) \cdot Lf_0$ *1 | $\forall k \in \mathbb{N}$ | $\forall n \in \mathbb{N}$ |
| $k$-peer exp. | $\mathcal{O}\left(\dfrac{\sigma^2}{n\epsilon^2} + \dfrac{\zeta \log_{k+1}(n) + \sigma\sqrt{p_{\text{kpexp}}\ \log_{k+1}(n)}}{p_{\text{kpexp}}\ \epsilon^{3/2}} + \dfrac{\log_{k+1}(n)}{p_{\text{kpexp}}\ \epsilon}\right) \cdot Lf_0$ *2 | $\forall k \in \mathbb{N}$ | $\forall n \in \mathbb{N}$ |
| Null-cascade | $\mathcal{O}\left(\dfrac{\sigma^2}{n\epsilon^2} + \dfrac{\zeta \tau_{\text{null}} + \sigma\sqrt{\tau_{\text{null}}}}{\epsilon^{3/2}} + \dfrac{\tau_{\text{null}}}{\epsilon}\right) \cdot Lf_0$ *3 | $\forall k (\geq 2)$ | $\forall n \in \mathbb{N}$ |

*1: $\tau_{\text{base}} = \kappa + \sum_{i=\kappa+1}^{\lambda} 2\lceil\log_{k+1}(\nu_i)\rceil$

*2: $p_{\text{kpexp}} = 1 - \max_{i\in\{1,\dots,n-1\}}\left|\dfrac{1}{(k+1)^{\lfloor\log_{k+1}(n)\rfloor}}\dfrac{\sin(\pi i(k+1)^{\lfloor\log_{k+1}(n)\rfloor}/n)}{\sin(\pi i/n)}\right|$ shown in Theorem 2

*3: $\tau_{\text{null}} = \kappa + \sum_{i=\kappa+1}^{\lambda} \nu_i/(2\lfloor k/2\rfloor)$, where $n$ is factorized as $n = \prod_{i=1}^{\lambda}(\nu_i+1)$, where $\nu_1 \leq \dots \leq \nu_\kappa \leq k < \nu_{\kappa+1} \leq \dots \leq \nu_\lambda$, and each $\nu_i$ is a natural number, with $\kappa$ factors being less than or equal to $k$.

# F    Relationships between expected consensus rate, periodic interval and $(n, k)$ configurations

In Sec. F.1, the relationships between the expected consensus rate $p$, the periodic interval $\tau$, and the number of nodes $n$ are illustrated for various values of $k$. These illustrations reveal that the periodic interval $\tau$ required for achieving finite-time convergence $(p = 1)$ in the null-cascade graph tends to increase when the factorization of $n$ includes large prime numbers, as discussed in Sec. 4. Nonetheless, experimental results presented in Sec. 5 indicate that performance is not compromised even under such scenarios. The primal reason is discussed in Sec. 5, with additional considerations provided in Sec. F.2.

## F.1    Illustrations of expected consensus rate $p$, periodic interval $\tau$, and $(n, k)$ configurations

The expected consensus rate $p$, the periodic interval $\tau$, and $(n, k)$ configurations are numerically investigated across several graphs. This examination covers a broad range of $n$ and $k = \{2, 3, 4\}$, as depicted in Fig. 29. For the $k$-peer exponential graph, the expected consensus rate becomes one (indicating finite-time convergence) for specific $n$ (a power of $k + 1$). As $n$ deviates from this condition, the expected consensus rate increases, as observed with configurations like $(n, k) = (17, 2)$ discussed in Sec. 5. The periodic interval $\tau$ for the $k$-peer exponential graph typically remains small, given by $\tau = \lfloor \log_{k+1}(n) \rfloor$. On the other hand, the consensus rate of both base-$(k+1)$ graph and null-cascade graph consistently reaches one, indicating they achieve finite-time convergence for any $n$. However, the required periodic intervals $\tau$ to achieve finite-time convergence differ between these two graphs. When the factorization of $n$ includes large prime numbers, $\tau$ in the null-cascade graph tends to increase; if not, $\tau$ in the null-cascade graph is generally equal to or less than that in the base-$(k+1)$ graph.

## F.2    Additional discussion

As shown in Fig. 29, the periodic interval $\tau$ for the null-cascade graph increases when the factorization of $n$ includes large prime numbers. Nonetheless, the experimental results presented in Sec. 5 demonstrated that the performance is not degraded even under such scenarios. The prime reason for this is discussed in the last paragraph of Sec. 4. This suggests that balanced mixing among $n$ nodes in the null-cascade graph leads to better experimental results.

Additionally, there is the issue of potential redundant counting of $\tau$ in the null-cascade graph. This arises because $\tau$ in the null-cascade graph is counted as the number of mixing matrices required to achieve finite-time convergence. However, a smaller $\tau$ could potentially be set by accepting a modest expected consensus rate. Furthermore, repeating a graph that achieves finite-time convergence twice results in a doubled periodic interval, yet the expected consensus rate remains unchanged. Even though the periodic interval can be doubled, this does not necessarily slow the convergence rate. Such observations indicate that a rigorous determination of $\tau$ in Assumption 4 cannot yet be definitively established, pointing to a direction for future research.

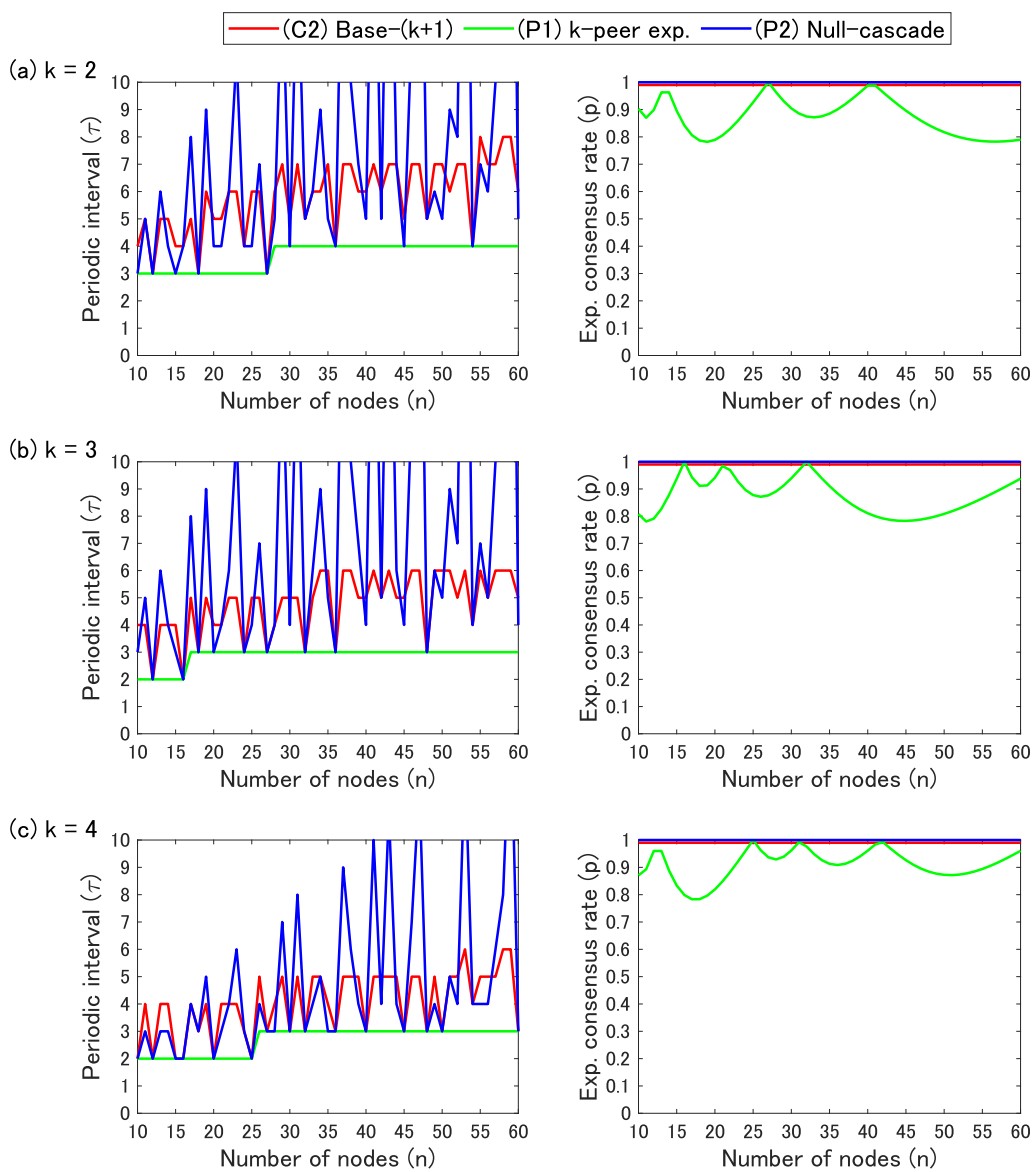

Figure 29: Relationships between expected consensus rate $p$, periodic interval $\tau$, and $n$ with $k = \{2, 3, 4\}$.

# G   Additional experiments

**Data distributions of CIFAR-10 and CIFAR-100.**   As described in Sec. 5, the training dataset of CIFAR-10 and CIFAR-100 [5] were divided into $n$ local datasets $\mathcal{D}_i$ ($i \in \{1, \dots, n\}$) to follow a Dirichlet distribution with concentration hyperparameter $\alpha$, using the source code used in [38][6]. We set $\alpha = 0.1$, representing a scenario with significant data heterogeneity. The distributions of the dataset for $n = \{15, 17, 30\}$ are illustrated in Figs. 30 and 31.

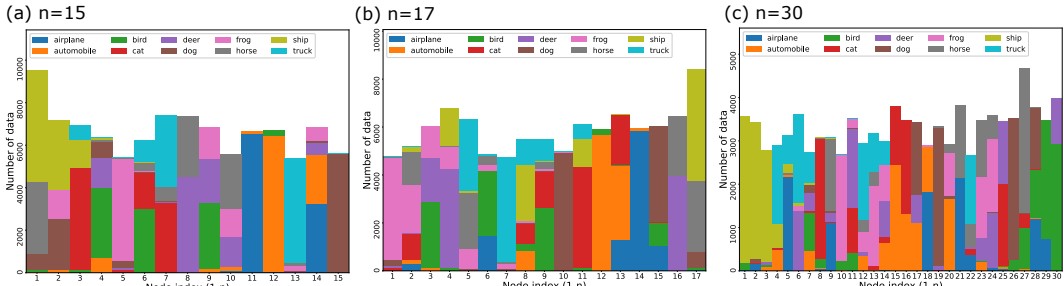

Figure 30: Data distributions of CIFAR-10 training dataset when (a) $n = 15$, (b) $n = 17$, and (c) $n = 30$.

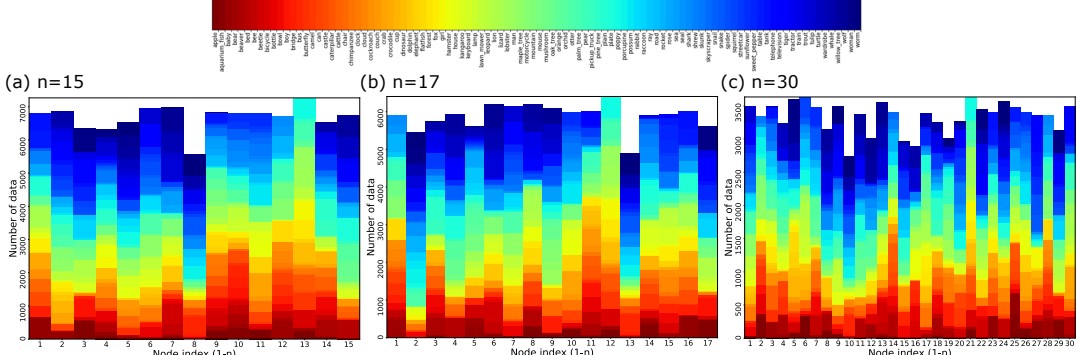

Figure 31: Data distributions of CIFAR-100 training dataset when (a) $n = 15$, (b) $n = 17$, and (c) $n = 30$.

**Update rules based on DSGD.** Our DSGD-based update rules that allow multiple local updates are summarized in Alg. 6. We set the number of inner multiple local updates to $T = 100$ for CIFAR-10 and $T = 10$ for CIFAR-100. The minibatch size to compute stochastic gradient was 64. As noted in the main paper, we computed the test accuracy of the global parameter $\overline{\mathbf{x}} = \frac{1}{n} \sum_{i=1}^{n} \mathbf{x}_i$ once every dozen communication rounds, for the evaluation metric.

---

**Algorithm 6** DSGD used in the experiments in Sec. 5

---

1: ▷ Initialization $\mathbf{x}_1 =, \dots, = \mathbf{x}_n, \eta, R, T, \mathcal{D}_i$ ($i \in [n]$)
2: ▷ Set mixing matrices $\{\boldsymbol{W}^{(0)}, \dots, \boldsymbol{W}^{(\tau-1)}\}$
3: **for** (Outer communication round) $r = 0, \dots, R - 1$ **do**
4:     $\boldsymbol{x}_i^{(r,0)} = \mathbf{x}_i^{(r)}$
5:     **for** (Inner local updates) $t = 0, \dots, T - 1$ **do**
6:        $\boldsymbol{x}_i^{(r,t+1)} = \boldsymbol{x}_i^{(r,t)} - \eta \nabla f_i(\boldsymbol{x}_i^{(r,t)}; \xi_i^{(r,t)})$          /* minibatch sampling $\xi_i^{(r,t)} \sim \mathcal{D}_i$ */
7:     **end for**
8:     $\mathbf{x}_i^{(r+\frac{1}{2})} = \boldsymbol{x}_i^{(r,T)}$
9:     ▷ Partial averaging
10:    $\mathbf{x}_i^{(r+1)} = \sum_{j=1}^{n} W_{ij}^{(\text{mod}(r,\tau))} \mathbf{x}_i^{(r+\frac{1}{2})}$
11: **end for**

---

[5] https://www.cs.toronto.edu/~kriz/cifar.html
[6] https://github.com/epfml/relaysgd

**Hyperparameter tuning.** For Alg. 6, the learning rate $\eta$ was initially pre-tuned for each dataset-model combination. To maintain consistency across various graphs and ensure a fair comparison, we used a common $\eta$ across all tested graphs. For this aim, we investigated $\eta$, which maximizes the test accuracy for each dataset-model combination, based on the results from single-node model training. Details of the associated hyperparameter settings are provided in Table 4. Through this hyperparameter tuning process, a learning rate of $\eta = 0.01$, which gradually reduces to $\eta = 0.001$ through cosine annealing, was chosen for all combinations of datasets (CIFAR-10 / CIFAR-100) and the model (ResNet-18).

Table 4: Hyperparameter settings for CIFAR-10/CIFAR-100 with ResNet-18.

| | |
|---|---|
| Data augmentation | RandomCrop, RandomHorizontalFlip, RandomErasing of PyTorch |
| Learning rate $\eta$ | Grid search over $\{0.1, 0.01, 0.001\}$ |
| Momentum | 0.9 |
| Weight decay | 0.005 |
| Batch size | 64 |
| Learning rate scheduler | Cosine annealing |
| Learning warmup | 10 epochs ($\eta = 5e^{-6}$) |

**Graphs used in experiments.** Several graphs used in the experiments are summarized as follows:

**(C1) Ring graph.** This is a static graph ($\tau = 1$) and each node connects with neighboring two nodes. The mixing matrix is given by

$$\boldsymbol{W}^{(0)} = \mathrm{circ}(\tfrac{1}{3}, \tfrac{1}{3}, \underbrace{0, \dots, 0}_{n-3 \text{ zeros}}, \tfrac{1}{3}),$$

where this holds $k = 2$ independently of $n$. Thus, we used this as a comparison graph for the configurations in the main paper configurations $(n, k) = (15, 2), (17, 2), (30, 2)$.

**(C1') Extended ring graph.** This is a static graph ($\tau = 1$). To enable a fair comparison with other graphs that support $k \geq 2$ connections, we extended the ring graph by connecting each node not only to its two immediate neighbors (i.e., $k = 2$) but also to additional adjacent nodes. For instance, when $k = 3$, the mixing matrix is given by

$$\boldsymbol{W}^{(0)} = \mathrm{circ}(\tfrac{1}{4}, \tfrac{1}{4}, \tfrac{1}{4}, \underbrace{0, \dots, 0}_{n-4 \text{ zeros}}, \tfrac{1}{4}),$$

and when $k = 4$, the mixing matrix is given by

$$\boldsymbol{W}^{(0)} = \mathrm{circ}(\tfrac{1}{5}, \tfrac{1}{5}, \tfrac{1}{5}, \underbrace{0, \dots, 0}_{n-5 \text{ zeros}}, \tfrac{1}{5}, \tfrac{1}{5}).$$

**(C2) Base-$(k+1)$ graph.** This is a dynamic graph ($\tau \geq 2$) and a related discussion is given in Sec. C.3. We computed the $\tau$ mixing matrix in the base-$(k+1)$ graph using the source code on the website[7].

**(C3) Random graph.** This is a static graph ($\tau = 1$), and is modeled as in the Erdos-Rényi random graph [5], adjusted to ensure $k$ for each configuration.

**(C4) $(k+1)$-partite random match graph.** This is a dynamic graph that is a generalization of the bipartite random match graph used as a comparison graph in [43], to accommodate $k$-peer communication. In the bipartite random match graph, two randomly selected nodes are connected with undirected edges, with the assumption that $n$ is even. Unlike other graphs where connections follow a periodic pattern, the connections between two nodes vary over time without a fixed period, thus this graph does not maintain a consistent periodic interval $\tau$. To generate the bipartite random match graph to allow any $(n, k)$ configurations, i) the index $(1, 2, \dots, n)$ to $(\rho^{(r)}(1), \rho^{(r)}(2), \dots, \rho^{(r)}(n))$ was randomly permuted, where $\rho^{(r)}(\cdot)$ denotes the permutation function at communication round $r$, ii) the forming subgroups consisted of as many $(k+1)$ nodes as possible, and iii) partial average

---

[7]`https://github.com/yukiTakezawa/BaseGraph`

within each subgroup after exchanging local parameters was performed. To the best of our knowledge, the expected consensus rate of the $(k + 1)$-partite random match graph, has not been investigated.

**(P1) $k$-peer exponential graph.** This is a dynamic graph, whose details are explained in Sec. 3 and Appendix A.

**(P2) Null-cascade graph.** This is a dynamic graph, whose details are explained in Sec. 4 and Appendix B.

**Computing resource.** We used computing servers employing 8 GPUs (NVIDIA RTX 6000 Ada (48 GB)) and 2 CPUs (AMD EPYC 9354, 3.25 GHz, 32-Core Processor).

**Additional experimental results.** Firstly, due to the space limitation, we picked several configurations in Fig. 4 in Sec. 5, its complete version is illustrated in Fig. 32. The corresponding highest test accuracy is summarized in Table 2.

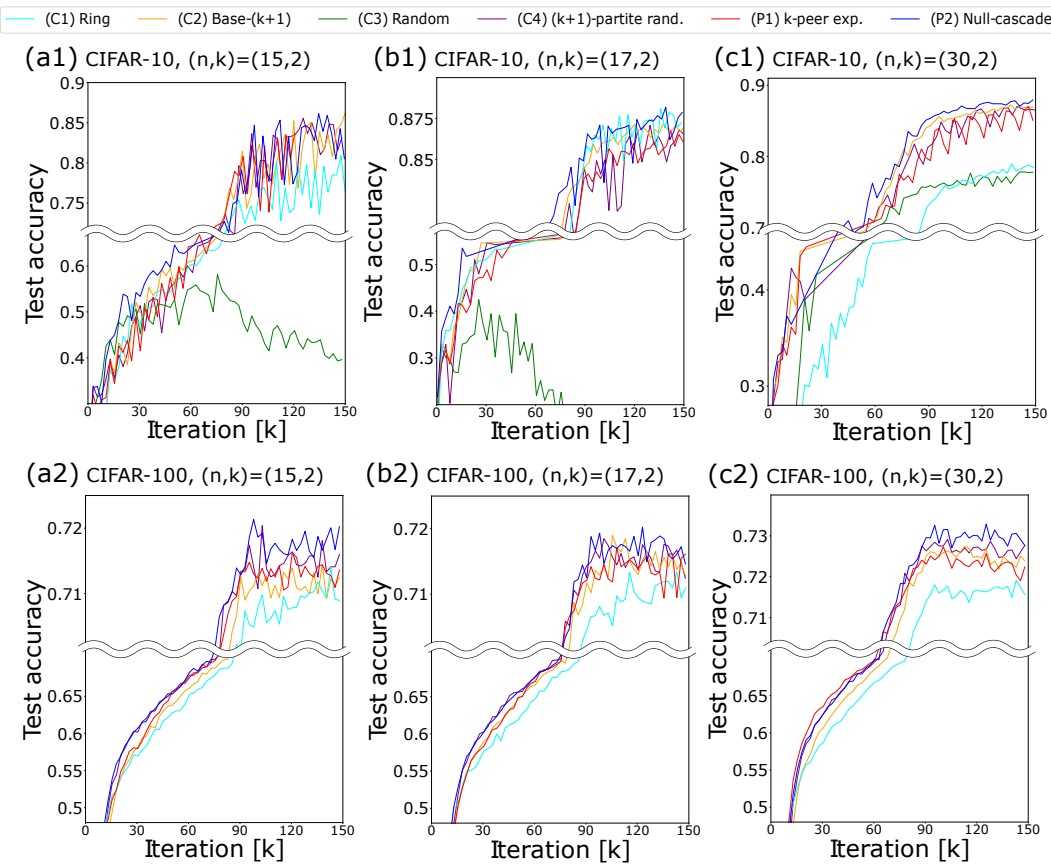

Figure 32: Convergence curves using test accuracy of global parameters for three $(n, k)$-configurations with two datasets (CIFAR-10/CIFAR-100). This figure is the complete version of Fig. 4 in Sec. 5.

Next, additional $(n, k)$-configurations were tested in this appendix section, particularly using $k \geq 3$. We conducted five additional $(n, k)$-configurations; namely (d1) $(n, k) = (15, 3)$, (e1) $(n, k) = (17, 3)$, (f1) $(n, k) = (30, 3)$, (g1) $(n, k) = (30, 4)$, (h1) $(n, k) = (31, 4)$ using CIFAR-10 classification task. The corresponding results are summarized in Table 5 and Fig. 33. Even among the additional $(n, k)$-configurations with $k \geq 3$, the proposed (P2) null-cascade graph consistently illustrated strong performance, including cases where $n$ is a large prime number. As discussed in the main paper, this can be attributed to the maximized expected spectral gap under uniform degree, although for some $(n, k)$-configurations, the periodic interval $\tau$ is increased. For $(n, k) = (30, 4)$, we observed that the mixing matrices consisting of the base-$(k+1)$ graph and the null-cascade graph are identical. As explained in Appendix C, this arises from the fact that the base-$(k+1)$ graph for certain $(n, k)$ configurations can be constructed following the design principle of the null-cascade

graph. Therefore, it is natural that the performance of the base-$(k+1)$ graph and the null-cascade graph is nearly identical in this case. To further investigate this, we conducted additional experiments with $(n,k) = (31,4)$, where $n$ is a large prime number. In this configuration, we observed that the null-cascade graph outperforms the base-$(k+1)$ graph. These additional experimental results highlight the effectiveness of the null-cascade graph across various $(n,k)$-configurations.

Table 5: Comparison of the highest test accuracy using global parameters for CIFAR-10 classification task for five additional $(n,k)$-configurations.

| $(n,k)$-configuration | (d1) $(n,k)=(15,3)$ | | (e1) $(n,k)=(17,3)$ | | (f1) $(n,k)=(30,3)$ | | (g1) $(n,k)=(30,4)$ | | (h1) $(n,k)=(31,4)$ | |
|---|---|---|---|---|---|---|---|---|---|---|
| Tested graphs | $\tau$ | test acc. | $\tau$ | test acc. | $\tau$ | test acc. | $\tau$ | test acc. | $\tau$ | test acc. |
| (C1') Extended ring | 1 | 0.8676 | 1 | 0.8756 | 1 | 0.8605 | 1 | 0.8544 | 1 | 0.8958 |
| (C2) Base-$(k+1)$ | 4 | 0.8467 | 5 | 0.8701 | 5 | 0.8640 | 3 | 0.8746 | 5 | 0.8974 |
| (C3) Random | 1 | 0.7560 | 1 | 0.8749 | 1 | 0.8747 | 1 | 0.8237 | 1 | 0.7576 |
| (C4) $(k+1)$-partite rand. match | – | 0.8574 | – | 0.8760 | – | 0.8760 | – | **0.8788** | – | 0.8999 |
| (P1) $k$-peer exponential | 2 | 0.8532 | 2 | 0.8789 | 3 | 0.8713 | 2 | 0.8780 | 2 | **0.9031** |
| (P2) Null-cascade | 3 | **0.8732** | 8 | **0.8817** | 3 | **0.8794** | 3 | 0.8746 | 8 | 0.9019 |

——— (C1') Extended Ring ——— (C2) Base-(k+1) ——— (C3) Random ——— (C4) (k+1)-partite rand.
——— (P1) k-peer exp. ——— (P2) Null-cascade

**(d1)** CIFAR-10, (n,k)=(15,3)

**(e1)** CIFAR-10, (n,k)=(17,3)

**(f1)** CIFAR-10, (n,k)=(30,3)

**(g1)** CIFAR-10, (n,k)=(30,4)

**(h1)** CIFAR-10, (n,k)=(31,4)

Figure 33:    Convergence curves using test accuracy of global parameters for five additional $(n,k)$-configurations.

**Further additional tests using large $n$.**

In the rebuttal phase, we performed additional experiments using large $n$. For $(n, k) = (69, 2)$, data distribution of CIFAR-10 with $\alpha = 0.1$ is illustrated as follow:

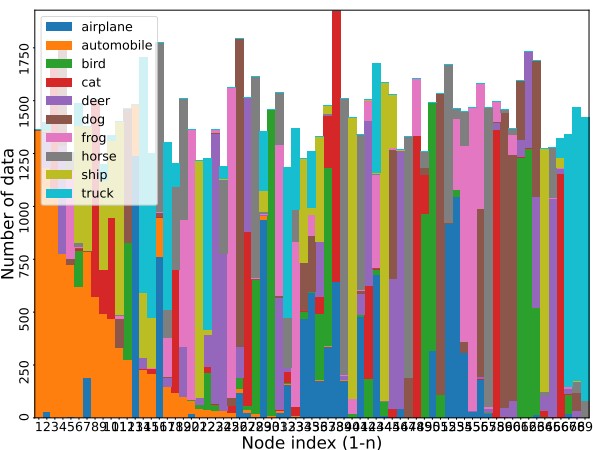

Figure 34: Data distributions of CIFAR-10 training dataset when $n = 69$.

The highest test accuracy and convergence curves are illustrated as follows:

Table 6: Comparison of the highest test accuracy using global parameters for CIFAR-10 classification task for $(n, k) = (69, 2)$.

| $(n, k)$-configuration | $(n, k) = (69, 2)$ | |
|---|---|---|
| Tested graphs | $\tau$ | test acc. |
| (C1') Extended ring | 1 | 0.8805 |
| (C2) Base-$(k{+}1)$ | 7 | 0.9213 |
| (C3) Random | 1 | 0.6267 |
| (C4) $(k{+}1)$-partite rand. match | — | **0.9229** |
| (P1) $k$-peer exponential | 4 | 0.9225 |
| (P2) Null-cascade | 12 | 0.9216 |

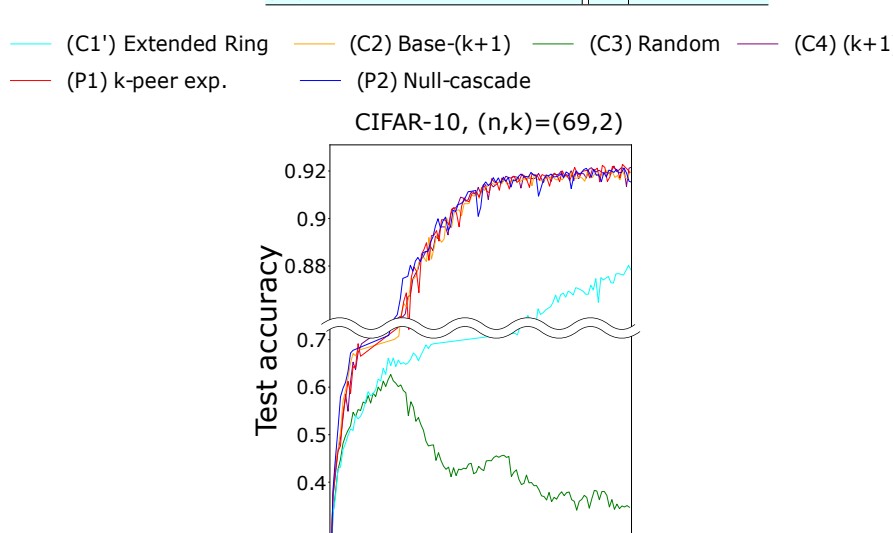

Figure 35: Convergence curves using test accuracy of global parameter for $(n, k) = (69, 2)$.

# H Related works

**(i) Centralized learning.** In distributed model parameter training, centralized learning such as Parallel SGD (PSGD), All-Reduce, and Federated Learning (FL) [23, 11, 10] often serve as the initial approach. FL, in particular, has seen significant efficiency gains due to client sampling by the center server and multiple local updates performed in each client. However, in scenarios involving extensive computing resources with many nodes/clients, a center server becomes a bottleneck due to the need to aggregate local model parameters from clients and distribute the global parameters back to them. To address this issue, research into decentralized learning approaches has been pursued, e.g., [16, 17].

**(ii) Decentralized learning algorithms.** The most widely used decentralized learning algorithm is DSGD [20, 26], and its extensions [16, 1, 2, 17]. Many researchers have integrated DSGD and well-known acceleration techniques, such as momentum [6, 18, 45, 46], communication compression techniques [9, 14, 21, 36, 39].

As illustrated in Theorem 1, the convergence rate of DSGD depends on data heterogeneity (Assumption 3). To mitigate this issue, many improved algorithms have been studied, such as Gradient Tracking (GT) and its extensions [3, 25, 30, 12, 33, 42, 48], $D^2$ [37], decentralized SCAFFOLD [19], primal-dual methods [47, 27, 28], etc.

In this paper, we primarily used DSGD as the fundamental decentralized learning algorithm. However, our proposed graphs are applicable to any decentralized learning algorithms.

**(iii) Other graphs for decentralized learning.** As illustrated in theoretical analysis for decentralized learning algorithms [13, 12, 15, 22, 40, 49, 33], network topology (graph) affects the convergence rate of many decentralized learning algorithms. Since the total communication costs are increased as the maximum degree $k$ increases [32, 43], constructing graphs to increase/maximize the (expected) consensus rate while maintaining a small maximum degree $k$ is essential for communication-efficient decentralized learning.

Several studies [32, 4, 40] have aimed to minimize communication costs by constraining the maximum degree of an underlying graph for communication-efficient decentralized learning. For instance, the ring graph, with its maximum degree $k$ of only 2, is simple yet communication-efficient. However, its consensus rate, $p = \mathcal{O}(1/n^2)$, as detailed in [24], suggests it performs poorly with large $n$. This observation aligns with the experimental findings presented in Sec. 5. When a larger $k$ is available, the (static) exponential graph [1] may be a good option. Its consensus rate is discussed in Sec. 2.2; although $k$ is dependent on $n$, limiting the flexibility of $(n, k)$ combinations. In the recent study by [44], the BTPP utilizes two spanning trees (static graphs) as communication graphs. This study analytically demonstrates finite-time convergence through multiple repeats of communication. However, it is important to note that not all combinations of $(n, k)$ are feasible with this method.

Nowadays, employing dynamic graphs for communication-efficient decentralized learning is a hot topic. We omit the discussion of works introduced in the main paper (e.g., [31, 43, 32, 4, 35]).

# I  Limitations and future work

First, the $k$-peer exponential graph achieves finite-time convergence for limited $n$ (a power of $k+1$). Despite this being a significant constraint, we calculated the expected consensus rate for the $k$-peer exponential graph for any $(n, k)$ configuration, as detailed in Sec. 3 and illustrated in Appendix F. Figure 29 in Appendix F shows that the expected consensus rate increases when $n$ deviates from conditions where $n$ is a power of $k+1$. Consequently, employing the $k$-peer exponential graph may be particularly advantageous for values of $n$ that are a power of $k+1$.

Second, it is necessary to have $k \geq 2$ in the null-cascade graph, as discussed in Sec. 4. This requirement arises from setting pairwise nulls in a conjugate relationship, which results in the polynomial equation in (9), ensuring that SNFs are real numbers. In an era characterized by substantial computational resources spread across multi-location data centers, the availability of a large number of nodes $n$ and a sufficiently large maximum degree $k$ is feasible. Our null-cascade graph is designed to be well-suited for this future scenario.

Third, the periodic interval $\tau$ required for achieving finite-time convergence in the null-cascade graph is not always small, as discussed in Sec. 4. Including large prime numbers $\nu_i$ in the factorization of $n$ increases in the periodic interval $\tau$ needed for finite-time convergence. Nevertheless, its impact would not be significant for the following reasons: i) uniform degree: the null-cascade graph consists of circulant matrices, resulting in a uniform degree such that every node has the same number of connections. Although the balanced mixing due to this is empirically known to influence outcomes, it has not yet been considered in the advanced convergence analysis (e.g., [13]); addressing this remains work for the future. The additional reason, ii) the possibility of redundant counting of $\tau$, can be found in Appendix F. Moreover, experimental results in Sec. 5 and Appendix G demonstrated that the null-cascade graph remains effective even when large prime numbers $\nu_i$ are included in the factorization of $n$.

Fourth, we report convergence curves with respect to communication rounds rather than run-time. This choice reflects our compute constraints—eight GPUs on a single server without NVLink—requiring us to simulate multiple nodes per GPU (e.g., for $n=30$, 3 or 4 nodes/GPU). Under this setup, accurately and fairly assessing communication overheads is difficult, and these costs grow especially large as the number of neighbors $k$ increases.

Another promising direction for future work involves leveraging non-zero spatial frequency components for personalized model training. Although this study focuses on training the global parameter (DC component), non-zero frequency components also contain valuable information that could significantly enhance model personalization, thus presenting a valuable avenue for future exploration.

# J  Impact statement

We present dynamic graphs for communication-efficient decentralized learning, which can be applied for training large-scale models (e.g., Large Language Models: LLMs) across extensive distributed computing resources, such as data centers. A potential risk of this technology is that it could enable a broader range of organizations to train large-scale models, which were previously restricted to organizations with access to substantial computing resources.

