# OpenReview forum: "Revisiting 1-peer exponential graph for enhancing decentralized learning efficiency"
_NeurIPS.cc/2025/Conference — NeurIPS 2025 poster_

### Official Review · Reviewer_6S3u · 2025-06-13

**Clarity:** 2
**Significance:** 2
**Originality:** 2
**Rating:** 5
**Confidence:** 3

**Summary:**

This paper re-visited well-studied one peer exponential graph and extended it to k peer. Using Steerable Nulling Filters, authors are able to propose the null-cascade graph which has finite-time convergence for any n. The authors present numerical experiments verifying their claims for both consensus errors and deep learning setting.

**Questions:**

1. In reference [26], it should be O(1) instead of o(1).
2. Adding more intuition and explanation about MAF and SNF would be helpful.
3. Adding the missing reference.

**Ethical Concerns:**

["NO or VERY MINOR ethics concerns only"]

**Final Justification:**

The authors have addressed all my concerns. I believe this work makes a meaningful contribution to the field of decentralized learning.

**Limitations:**

Yes

**Quality:**

2

**Strengths And Weaknesses:**

Strengths:
1. The authors re-stated some basic definitions/theorems from the one peer exponential graphs, which is a good refresher in case some readers are not familiar with decentralized learning.
2. Although the definition of the k-peer exponential graph is a trivial extension of the one peer case, the authors proposed a necessary condition using Fourier transform in the frequency space. They also found the expected spectral gap of the k-peer exponential graphs.
3. To support the case when n is not power of k+1, authors proposed null-cascade graph and showed they have expected spectral gap 1.
4. The authors tested their algorithm for consensus errors. The outcome is consistent with their claim. From the plots, one can clearly observe that it has finite time convergence. The null cascade graph also performed well on CIFAR10 and CIFAR10 in the deep learning setting.

Weaknesses:
1. The optimal rate for any n has been achieved in for example [1]. The authors should at least mention this work.
2. The extension from 1 peer exponential graph to k peer is quite straightforward.
3. The authors included little intuition and explanation about MAF. Specifically, "We notice that (4) is a MAF with (k + 1) uniform weights and its expansions" needs more details.
4. The authors only discussed spectral gap of the null cascade graph. While having some deep learning experiments, the authors did not prove the method converge in the deep learning setting, unlike many previous work, [1, 2, 3].


[1] Ding et al. DSGD-CECA: Decentralized SGD with Communication-Optimal Exact Consensus Algorithm, 2023.

[2] Ying et al. Exponential graph is provably efficient for decentralized deep training, 2021.

[3] Song et al. Communication efficient topologies for decentralized learning with O(1) consensus rate, 2023.

---

> ### Author Rebuttal · Authors · 2025-07-31
>
> First of all, we sincerely appreciate your insightful comments. We will carefully incorporate your feedback to improve the quality of this paper.
>
> __[W1] The optimal rate for any $n$ has been achieved in for example [1]. [1] Ding et al. DSGD-CECA: Decentralized SGD with Communication-Optimal Exact Consensus Algorithm, 2023.__
>
> As you suggested, a missing reference [1] will be cited in the revised version. We believe that our study offers advantages by proposing graphs that maximally leverage distributed computing resources capable of supporting multi-peer communication with $k \geq 2$.
>
> __[W2] The extension from 1-peer exponential graph to $k$-peer is quite straightforward.__
>
> We believe that Section 3, which extends from 1-peer exponential graph to $k$-peer exponential graph, plays a crucial role in establishing the connection between the expected spectral gap and the Discrete Fourier Transform (DFT). While, as you pointed out, the extension from the 1-peer to be $k$-peer exponential graph may not represent a major novelty on its own, __its importance lies in enabling a novel analysis of the expected spectral gap through the DFT of the employed filters,__ specifically, the Moving Average Filters (MAFs) that underlie the $k$-peer exponential graph.
>
> Building on this foundation, Section 4 extends this idea by incorporating Steerable Null Filters (SNFs) alongside MAFs, leading to the formulation of the proposed null-cascade graph. In this context, the $k$-peer exponential graph serves as a critical conceptual and technical milestone towards the development of the null-cascade graph.
>
> __[W3] [Q2] Adding more intuition and explanation about MAF and SNF would be helpful.__
>
> While the application of DFT to the filters (e.g., MAFs and SNFs) is standard practice in the signal processing community, it may be unfamiliar to many researchers in the machine learning field. Therefore, we will include more intuitive explanations of MAFs and SNFs, along with their frequency characteristics, in the Appendix of the revised version.
>
> __[W4] The authors only discussed spectral gap of the null-cascade graph.__
>
> __See Table 3 in Appendix D, which presents the convergence rates of DSGD with various graphs,__ including our proposed graphs: the $k$-peer exponential graph and the null-cascade graph. These convergence rates can be derived by substituting the expected spectral gap $p$ and the periodic interval $\tau$ into the convergence expression of DSGD given in Theorem 1. Due to space constraints, we have omitted these details from the main paper.
>
> __[Q1] In [26], it should be O(1) instead of o(1).__
>
> Thank you for pointing this out. This will be modified in the revised version.

---

> > ### Comment · Reviewer_6S3u · 2025-08-03
> >
> > Thank the authors for their clarification. I have increased my rating to 5.

---

### Official Review · Reviewer_Nrs7 · 2025-06-30

**Clarity:** 4
**Significance:** 3
**Originality:** 3
**Rating:** 5
**Confidence:** 4

**Summary:**

The paper examines the role of communication network topologies in decentralized learning, reinvestigating 1-peer exponential topologies and introducing an extended version: $k$-peer exponential networks followed by the nullcascade graph topology. The nullcascade topology demonstrates finite-time convergence for any possible number of nodes and connections while utilizing uniform degree subgraphs, unlike the base $(k+1)$ communication topology.

**Questions:**

see weaknesses

**Ethical Concerns:**

["NO or VERY MINOR ethics concerns only"]

**Final Justification:**

My initial score was 5, and from the first time I read the paper, I thought it had a good chance of being accepted, as it is well written and presents a novel and interesting approach. As someone working on decentralized learning, I believe their contribution is both significant and necessary for the field. Their response to my questions was also satisfactory.

**Limitations:**

The proposed approach requires predefined subgraph structures based on fixed $(n, k)$ values, assuming stable, fully connected topologies.

**Paper Formatting Concerns:**

no concern

**Quality:**

3

**Strengths And Weaknesses:**

#### **Strengths:**

- The paper is well written and easy to read, with extra information and clear examples provided in the appendix.
- The dependence of finite-time convergence on isolating the DC component while nullifying other non-zero frequencies is a novel and interesting finding.
- The design of MAF and SNF filters to nullify non-zero frequencies is a clear, effective approach.
- The paper includes solid experimental studies showing better performance compared to base $(k+1)$.

---

#### **Weaknesses and Questions:**

- The role of commutativity is not clear, even though the paper states that “commutativity necessitates strict lock-step synchronization to coordinate the periodic communication cycles among nodes.”
  - How does commutativity specifically help decentralized training?
  - Can asynchronous training be supported in these topologies?
  - How can practical implementations benefit from commutative network structures in real-world decentralized training systems?

- Can a filter be designed that directly zeros out undesired frequencies without cascading many filters? For example, instead of using $\tau$ circulant matrices, is it possible to define a filter that places the required zeros at the targeted frequencies in a single communication round, potentially with fewer or different graph connections?


- The proposed approach requires predefined subgraph structures based on fixed $(n, k)$ values, assuming stable, fully connected topologies. In dynamically changing networks where nodes may leave, join, or lose connectivity, the designed structures may no longer satisfy the required properties, limiting its practical applicability in dynamic decentralized learning environments. Is this correct?

- Algorithm 4 seems complex, but the paper does not discuss the implementation or runtime cost of Algorithms 3 and 4, which is important for assessing scalability.


- When using \paragraph{Text}, it is clearer to end the paragraph title with a period (i.e., \paragraph{Text.}), which visually separates the heading from the body text and improves readability.

---

> ### Author Rebuttal · Authors · 2025-07-31
>
> First of all, we sincerely appreciate your insightful comments. We will carefully incorporate your feedback to improve the quality of this paper.
>
> __[W1] Unclear role of commutativity.__
>
> __- Can asynchronous training be supported in these topologies?__
>
> Under __synchronized communication among $n$ nodes,__ the mixing matrices can be applied in an arbitrary order, as long as the corresponding graphs satisfy the commutativity of mixing matrices specified in Definition 4.
>
> __- How does commutativity specifically help decentralized training?__
>
> We agree this is an important question. Given your insightful comment, we reconsider the benefit of commutativity of mixing matrices in decentralized learning, and a possible idea which may mitigate your concern is summarized as follows:
>
> First, we would like to emphasize that reducing the discrepancy between the global model $\bar{x}^{(r)}$ and local models $x_{i}^{(r)}$, specifically, minimizing $\sum_{i=1}^{n} \| \bar{x}^{(r)} - x_{i}^{(r)} \|$ is crucial for improving the convergence rate of decentralized learning algorithms. This quantity explicitly appears in the convergence rate of DSGD, as stated in Theorem 1 (see proof in [12]). __Ensuring the commutativity of the mixing matrices within graphs would help reduce this discrepancy.__
>
> To illustrate this, we reformulate the update rules of DSGD recursively, as follows:
>
> $X^{(r+1)}
> = X^{(r)} W^{(r)} - \eta \nabla F(X^{(r)}; \xi^{(r)}) W^{(r)}$
>
> $= X^{(r-1)} W^{(r-1)} W^{(r)} - \eta \{ \nabla F(X^{(r-1)}; \xi^{(r-1)}) W^{(r-1)} W^{(r)} + \nabla F(X^{(r)}; \xi^{(r)}) W^{(r)} \}$
>
> $\vdots$
>
> $= X^{(0)} W^{(0)} \cdots W^{(r)} - \eta \{ \nabla F(X^{(0)}; \xi^{(0)}) W^{(0)} \cdots W^{(r)} + \cdots + \nabla F(X^{(r)}; \xi^{(r)}) W^{(r)} \}$
>
> $= X^{(0)} \prod_{s=0}^{r} W^{(s)} - \eta \sum_{s_{1}=0}^{r} \{ \nabla F(X^{(s_{1})}; \xi^{(s_{1})})  \prod_{s_{2}=s_{1}}^{r} W^{(s_{2})} \}.$
>
> Next, we investigate the property of $\prod_{s_{2} = s_{1}}^{r} W^{(s_{2})}$ for finite-time convergent graph with periodic interval $\tau$.
>
> __[Case 1: Commutativity of mixing matrices is satisfied]__
>
> When commutativity of mixing matrices is satisfied, $W^{(\rho(0))} W^{(\rho(1))} \cdots W^{(\rho(\tau-1))} = \frac{1}{n} 1 1^{T}$ for any permutation $\rho$. The product $\prod_{s_{2} = s_{1}}^{r} W^{(s_{2})}$ can be categorized into two cases:
>
> (Case 1a) When $s_{1} \leq r - \tau + 1$, $\prod_{s_{2} = s_{1}}^{r} W^{(s_{2})} = \frac{1}{n} 1 1^{T}$.
>
> (Case 1b) When $s_{1} \geq r - \tau + 2$, $\prod_{s_{2} = s_{1}}^{r} W^{(s_{2})} \neq \frac{1}{n} 1 1^{T}$.
>
> These results indicate that no discrepancy among $n$ nodes can occur for terms with index $s_{1} \in [0, r - \tau + 1]$ (in Case 1a). In contrast, discrepancy among $n$ nodes can occur for terms with index $s_{1} \in [r - \tau + 2, r ]$ (in Case 1b), which leads to degradation in the convergence rate.
>
> __[Case 2: Commutativity of mixing matrices is NOT satisfied]__
>
> When we denote $\mod (r, \tau) = \phi$, the product $\prod_{s_{2} = s_{1}}^{r} W^{(s_{2})}$ can be categorized into two cases:
>
> (Case 2a) When $s_{1} \leq r - \tau - \phi + 1$, $\prod_{s_{2} = s_{1}}^{r} W^{(s_{2})} = \frac{1}{n} 1 1^{T}$.
>
> (Case 2b) When $s_{1} \geq r - \tau - \phi + 2$, $\prod_{s_{2} = s_{1}}^{r} W^{(s_{2})} \neq \frac{1}{n} 1 1^{T}$.
>
> These results indicate that no discrepancy among $n$ nodes can occur for terms with index $s_{1} \in [0, r - \tau - \phi + 1]$ (in case 2a). In contrast, discrepancy among $n$ nodes can occur for terms with index $s_{1} \in [r - \tau - \phi + 2, r ]$ (in case 2b), which leads to degradation in the convergence rate.
>
> Comparing these two cases, __the components that contribute to discrepancies among the $n$ nodes tend to be more significant when $\mod (r, \tau) = \phi$ is non-zero.__ Although this difference may be small and therefore difficult to observe empirically, __we have theoretically demonstrated the advantage of using commutative mixing matrices.__ To reflect this discussion, we will revise the sentence in the Introduction regarding commutativity and add an Appendix section to explain the potential benefits of commutativity as discussed above.
>
> __[W2] Can a filter be designed that directly zeros out undesired frequencies without cascading many filters?__
>
> When a large value of $k$ is available, undesired frequencies can be effectively nullified without the need to cascade many filters. For instance, in the extreme case of $k = n - 1$, all non-zero frequencies can be eliminated; however, this corresponds to the complete graph, where each node is connected to all other nodes.
>
> We highlight a fundamental trade-off between the number of nullified frequencies and the value of $k$. In practical network settings, only a relatively small $k$ (i.e., $k \ll n$) is typically feasible. Under such constraints, __it is impossible to nullify undesired frequencies without cascading multiple filters.__
>
> __[W3] In dynamically changing networks where nodes may leave, join, or lose connectivity, the designed structures may no longer satisfy the required properties. Is this correct?__
>
> Yes, our problem setting is to construct dynamical graphs given $(n, k)$; namely, dynamically-changing $(n, k)$ scenarios are out of scope.
>
> __[W4] Discussion of implementation or runtime cost of Algorithms 3 and 4.__
>
> First, we assume that our graph is constructed at the initialization step of decentralized learning. Since we do not consider dynamically changing networks, as you pointed out in [W3], we believe that computational cost would not pose a serious issue.
>
> Second, __we emphasize that the runtime cost of Algorithms 1-4 is negligible for many $(n, k)$-configurations.__ Algorithms 1-3 are straightforward to implement and incur negligible runtime overhead. Algorithm 4, which is used to select the roots $q$, expansions $m$, and communication orders $c$ for computing SNFs, may be complex to implement and may introduce noticeable runtime costs. This potential complexity stems from the fact that combinations of $(q, m, c)$ are not uniquely determined, requiring a greedy search to identify suitable candidates. While we have not exhaustively evaluated the runtime cost for all possible $(n, k)$-configurations, we confirm that for those configurations tested in our experiments (as reported in Section 5 and Appendix F), the runtime costs were negligible. To further address this concern, we will include runtime measurements for the $(n,k)$-configurations used in our experiments and will release the corresponding source code upon the publication of this paper.
>
> __[W5] Missing period in paragraph title.__
>
> As you suggested, we separate the paragraph title from the body using period in the revised version.

---

> ### Comment · Reviewer_Nrs7 · 2025-08-04
> **Thanks for your response.**
>
> I have read your rebuttal and have decided to keep my rating since my score is already 5 (accept). Good luck!

---

> > ### Author Response · Authors · 2025-08-05
> >
> > Thank you for taking the time to review our rebuttal. Please let us know if any additional information is needed to support your evaluation. Thank you again.

---

### Official Review · Reviewer_cN3U · 2025-07-02

**Clarity:** 2
**Significance:** 3
**Originality:** 3
**Rating:** 4
**Confidence:** 2

**Summary:**

This paper studies communication-efficient decentralized learning by improving the expected spectral gap to reduce deviations from global averaging. This paper addresses certain limitations of the 1-peer exponential graph and propose two new dynamic graphs: the k-peer exponential graph and the null-cascade graph. The null-cascade graph achieves finite-time convergence for any n while ensuring commutativity.

**Questions:**

1. Why do we need the 1-peer exponential graph? What is the advantage over the other graphs with constant spectral gap near 1, especially some well-known static graphs? What is the advantage of the 1-peer exponential graph?

2. The construction of Null-cascade graph is a double loop algorithm. Is constructing such networks time-consuming in practical applications?

3. Do we need to run Algorithm 1 at each iteration of the dentralized algorithm (e.g., DSGD), or every $\tau$ iterations?

4. In the experiments, this paper considers the number of agents ranging from 15 to 30. What occurs when scaling to 1000+ agents?

**Ethical Concerns:**

["NO or VERY MINOR ethics concerns only"]

**Final Justification:**

I agree to accept this paper.

**Limitations:**

The main paper is difficult to follow.

**Quality:**

3

**Strengths And Weaknesses:**

Strengths:

1. To address the limitation of the 1-peer exponential graph that the number of nodes n should be a power of two, this paper explores the extension to the k-peer exponential graph and observe that finite-time convergence can be interpreted by the DC component, suggesting that desiring finite-time convergence is only achieved when n is a power of k+1.

2. To achieve finite-time convergence for any n while preserving commutativity, this paper proposes the null-cascade graph. The effectiveness of the proposed null-cascade graph was confirmed through numerical experiments.

3. This study is significant in decentralized learning because communication serves as the bottleneck in distributed optimization, making the design of communication networks to reduce overhead a significant research problem.

4. This paper seems theoretically solid. It addresses the limitation of the 1-peer exponential graph in theory and thus it seems novel.

Weaknesses:

1. There are many static graphs with nearly constant spectral gap p such that $\frac{1}{p}$ in Theorem 1 is not large. Why do we need to design a new complex dynamic graph?

2. The proposed null-cascade graph seems complex to implement. The reviewer suggests to discuss its disadvantage in practical implementation.

3. The reviewer suggests to verify the efficiency of the proposed null-cascade graph on larger networks with 1000+ agents.

4. While theoritically solid, this paper's theoretical complexity creates barriers to comprehension. The main paper is difficult to follow.

5. Suggest introducing more examples for the practical applications of 1-peer exponential graph and the proposed null-cascade graph in the machine learning community.

In conclusion, this paper is theoretically solid. The problem studied in this paper is significant. However, it is difficult to follow.

---

> ### Author Rebuttal · Authors · 2025-07-31
>
> First of all, we sincerely appreciate your insightful comments. We will carefully incorporate your feedback to improve the quality of this paper.
>
> __[W1] Why do we need to design a new complex dynamic graph?__
>
> When the maximum degree $k$ (maximum number of each node connections) is restricted to be a small value $(k \ll n)$, static graphs cannot achieve a spectral gap $p$ such that $1/p$ in Theorem 1 is not large. In Table 3 of Appendix D, we present the convergence rates of DSGD using various graphs, including several static graphs such as ring, torus, and exponential graphs. These results are obtained by substituting the following spectral gap into the convergence rate for DGSD given in Theorem 1:
>
> Ring: $p = O(1 / n^2), k = 2, \tau=1$,
>
> Torus: $p = O(1 / n), k = 4, \tau=1$,
>
> Exponential: $p = O(1 / \log_{2} n), k = \lceil \log_{2}(n) \rceil, \tau=1$.
>
> Among these static graphs, the spectral gap can be improved using the exponential graph; however, it requires $k = \lceil \log_{2}(n) \rceil$ connections. Therefore, it is infeasible when $k$ is restricted to be a small value relative to $n$ (i.e., $k \ll n$).
>
> To address this limitation, dynamic graphs have been explored for communication-efficient decentralized learning. Notable examples include the 1-peer exponential graph, base-(k+1) graph, and EquiTopo graphs.
>
> __[W2] Discuss disadvantages of the proposed null-cascade in practical implementation.__
>
> We emphasize that Algorithms 1-3 are not difficult to implement. However, as you pointed out, Algorithm 4--which is used to select roots $q$, expansions $m$, and communication orders $c$ to compute SNFs may be complex. This complexity arises because combinations of $(q, m, c)$ are not uniquely determined, requiring a greedy search to identify suitable candidates. To mitigate your concern, we will include a discussion of this limitation in the null-cascade graph implementations, and we will release the organized source code upon publication of this paper.
>
> __[W3], [Q4] Verify the efficiency of the proposed null-cascade graph on larger networks.__
>
> To address your concern, __we conducted additional evaluations using a relatively large network size.__ While you suggest testing at an even larger scale (e.g., $n > 1000$), such an experiment exceeds the limits of our available computing resources. As a compromise, we selected $(n, k) = (75, 2)$. The evaluation setting follows the same CIFAR-10 classification task with heterogeneous data allocation ($\alpha = 0.1$) as in Section 5.2 (Test 2).
>
> __[CIFAR-10 classification ($\alpha=0.1$) with $(n, k) = (75, 2)$]__
>
> |Communication round |r=5|r=100|r=200|r=300|r=400|r=500|r=600|r=700|r=800|
> |---------------- | --- | --- | --- | --- | --- | --- | --- | --- | --- |
> |(C1a) Ring|__0.1953__|0.4882|0.5173|0.5496|0.5802|0.616|0.6442|0.6735|0.7038|0.7252|
> |(C2) Base-(k+1)|0.1840|0.6036|0.7144|0.7963|0.8274|0.8527|0.8717|0.8848|0.8941|0.9003|0.9019|
> |(C4) (k+1)-pertite rand.|0.1795|0.5788|0.7118|0.7800|0.8312|0.8550|0.8732|0.8885|0.8962|
> |(P1) k-peer exp.|0.1896|0.5837|0.7121|0.7874|0.8274|0.8545|0.8736|0.8860|0.8942|
> |(P2) Null-cascade|0.1927|__0.6301__|__0.7506__|__0.8095__|__0.8417__|__0.8649__|__0.8793__|__0.8935__|__0.8995__|
>
> Even with a relatively large network ($n = 75$), the highest test accuracy was achieved using our proposed null-cascade graph. This indicates that the experimental trend remains consistent with the results reported in the original paper.
>
> We will include additional experimental results using relatively large $n$ in the revised version, along with consensus rate investigations similar to those shown in Section 5.1 (Test 1).
>
> __[W4] This paper's theoretical complexity creates barriers to comprehension.__
>
> We apologize if parts of our paper were difficult to follow. Balancing accessibility with theoretical completeness within the space constraints has indeed been challenging. We would like to respectfully highlight that other reviewers found the writing to be clear.
>
> __[W5] Introducing examples for the practical applications.__
>
> A promising application involves __decentralized learning of machine learning models, such as deep learning models, across data center networks.__ As modern data centers are geographically distributed across multiple regions, aggregating all data to a single site is often impractical due to regulatory, privacy, or legal constraints. Therefore, effectively leveraging both the distributed computational resources and the locally stored data at these sites is essential.
>
> Given the operational overheads associated with, e.g., parameter tuning and algorithm implementation, the simplest decentralized learning algorithm, DSGD, is frequently adopted in real-world distributed computing environments. When deploying DSGD over data center networks, one can select a communication graph to perform partial averaging of local models. In this regard, the communication graphs we propose offer significant advantages in terms of both communication efficiency and the test accuracy of the resulting models.
>
> __[Q1] What is the advantage of the 1-peer exponential graph?__
>
> When the maximum degree $k$ is constrained (e.g., to reduce communication overhead), the spectral gap of a static graph can only be improved up to a certain point. In contrast, allowing dynamically changing connections between nodes opens up the possibility of increasing the spectral gap in expectation.
>
> Motivated by this aim, prior works have explored dynamic graphs, such as 1-peer exponential graph, base-(k+1) graph, and EquiTopo graph. In our work, we further investigate more advanced dynamic graphs, including $k$-peer exp. graph and null-cascade graph.
>
> __[Q2] Is constructing such networks time-consuming in practical applications?__
>
> __[Q3] Do we need to run Algorithm 1 at each iteration?__
>
> __The communication graph is constructed based on $(n,k)$ at the initialization step, prior to the start of model training, that is, Algorithm 1 does NOT need to run at each iteration.__ This is identified in line 2 of Algorithm 6 in Appendix F. While, as you rightly pointed out, constructing such networks can be time-consuming, this is required only once at initialization. We suspect this may not have been fully apparent, and we will clarify it in the revised version.

---

> > ### Comment · Reviewer_cN3U · 2025-08-04
> >
> > Thank you for your response. I have read through the rebuttal and have decided to keep my rating.

---

### Official Review · Reviewer_3NBf · 2025-07-02

**Clarity:** 4
**Significance:** 3
**Originality:** 3
**Rating:** 5
**Confidence:** 4

**Summary:**

This paper tackles the problem of communication efficiency in decentralized learning by revisiting and extending the concept of the 1-peer exponential graph. The authors introduce two dynamic graph constructions, i.e., the k-peer exponential graph and the null-cascade graph. The core contribution centers on the theoretical analysis of eigenvalues of the mixing matrix. Empirical evaluations on consensus error and decentralized DNN training support the effectiveness of these new graphs, particularly the null-cascade graph.

**Questions:**

Apart from the comments in the weakness section, I have the following questions:
1. In the simulation, could the authors explain why choosing $k = 2$? Is it purely an arbitrary choice of any benefits from it?
2. Regarding the applications, could the authors provide some examples where the network topology in the decentralized learning is by user's design?

**Ethical Concerns:**

["NO or VERY MINOR ethics concerns only"]

**Final Justification:**

The authors have addressed all my concerns about simulations and the motivation. The position of this work is clear to me now. I believe this work makes a meaningful contribution to the field of decentralized learning with efficient dynamic graph constructions. Therefore, I would raise my score to 5 and recommend the acceptance of this paper.

**Limitations:**

Yes.

**Quality:**

3

**Strengths And Weaknesses:**

Strengths:
1. A clear and rigorous theoretical advancement is offered in this paper. I learned the DSP techniques from this paper to analyze eigenvalues, which also provides design intuition and a principled approach to constructing efficient dynamic graphs. Proofs for spectral gap and commutativity are rigorous.
2. Both the k-peer exponential graph and the null-cascade graph represent novel additions to the literature on decentralized graph topologies.

Weaknesses:
1. The empirical evaluations are conducted on relatively small networks (up to $n=30$ nodes).
2. The motivation and benefits of extending from 1-peer exponential graph to k-peer graph are limited. I would expect more discussions about how this extension could bring benefits to both theoretical and numerical improvements of communication efficiency.
3. Following the previous point, k-peer exponential graph leads to more communication costs between agents. Thus, only showing the results based on communication rounds or iterations are inadequate or unconvincing (to claim efficiency).

---

> ### Author Rebuttal · Authors · 2025-07-31
>
> First of all, we sincerely appreciate your insightful comments. We will carefully incorporate your feedback to improve the quality of this paper.
>
> __[W1] Empirical evaluations on relatively small networks $(n \leq 30)$.__
>
> To address your concern, __we additionally conducted evaluations using a relatively large network size (e.g., $n = 75$).__ The evaluation setting follows the same CIFAR-10 classification task with heterogeneous data allocation ($\alpha = 0.1$) as in Section 5.2 (Test 2).
>
> __[CIFAR-10 classification ($\alpha=0.1$) with $(n, k) = (75, 2)$]__
>
> |Communication round |r=5|r=100|r=200|r=300|r=400|r=500|r=600|r=700|r=800|
> |---------------- | --- | --- | --- | --- | --- | --- | --- | --- | --- |
> |(C1a) Ring|__0.1953__|0.4882|0.5173|0.5496|0.5802|0.616|0.6442|0.6735|0.7038|0.7252|
> |(C2) Base-(k+1)|0.1840|0.6036|0.7144|0.7963|0.8274|0.8527|0.8717|0.8848|0.8941|0.9003|0.9019|
> |(C4) (k+1)-pertite rand.|0.1795|0.5788|0.7118|0.7800|0.8312|0.8550|0.8732|0.8885|0.8962|
> |(P1) k-peer exp.|0.1896|0.5837|0.7121|0.7874|0.8274|0.8545|0.8736|0.8860|0.8942|
> |(P2) Null-cascade|0.1927|__0.6301__|__0.7506__|__0.8095__|__0.8417__|__0.8649__|__0.8793__|__0.8935__|__0.8995__|
>
> Even with a relatively large network ($n = 75$), the highest test accuracy was achieved using our proposed null-cascade graph. This indicates that the experimental trend remains consistent with the results reported in the original paper.
>
> We will include additional experimental results using relatively large $n$ in the revised version, along with consensus rate investigations similar to those shown in Sec. 5.1 (Test 1).
>
> __[W2] Limited motivation and benefits of extending from 1-peer exponential graph to k-peer graph.__
>
> We may be misunderstanding your question, but if high-throughput distributed computing resources capable of supporting fast $k$-peer communication are available, then increasing the number of connections (with a reasonably large $k$) can enable communication-efficient decentralized training of models. Thanks to your question, we realized that our motivation for designing graphs extended to $k$-peer communication may not have been clearly conveyed. In the revised version of the Introduction, we would like to improve the explanation to make this point clearer.
>
> __[W3] Only showing the results based on communication rounds or iterations is inadequate.__
>
> Thank you for pointing this out. First, we would like to clarify the limitations of our computing resources: our computational setup consists of eight GPUs in a single server without NVLink connectivity. Due to these constraints, we are required to simulate multiple nodes' updates on a single GPU. For example, when $n=30$, updates from $3$ or $4$ nodes are processed on each GPU. As a result, __it is difficult to accurately and fairly evaluate the impact of communication overheads.__ Moreover, these overheads become particularly significant when the number of neighbors $k$ is large. We kindly ask you to take our computational resource limitations into consideration. We believe that our proposed graphs are well-suited for deployment on more advanced computing infrastructures that offer high throughput and/or native support for simultaneous multicasting.
>
> For this reason, we conducted comparisons with baseline graphs that have the same maximum degree $k$, including the base-$(k+1)$ graph, static random graph, and dynamic random graph (i.e., $(k+1)$-pertite random match graph). We will revise the future work section in Appendix H to reflect this discussion.
>
> __[Q1] In the simulation, why choosing k=2?__
>
> Although we present experimental results with $k=2$ in the main paper, __we also provided additional experimental results for $k \in \{ 3, 4 \}$ in Appendix F (see Table 5 and Figure 33).__ Our proposed graphs are not limited to the case of $k=2$. If an additional page is available, we will include these extended results in the revised version.
>
> __[Q2] Application examples.__
>
> A promising application involves __decentralized learning of machine learning models, such as deep learning models, across data center networks.__ As modern data centers are geographically distributed across multiple regions, aggregating all data to a single site is often impractical due to regulatory, privacy, or legal constraints. Therefore, effectively leveraging both the distributed computational resources and the locally stored data at these sites is essential.
>
> Given the operational overheads associated with, e.g., parameter tuning and algorithm implementation, the simplest decentralized learning algorithm, DSGD, is frequently adopted in real-world distributed computing environments. When deploying DSGD over data center networks, one can select a communication graph to perform partial averaging of local models. In this regard, the communication graphs we propose offer significant advantages in terms of both communication efficiency and the test accuracy of the resulting models.

---

### Comment · Area_Chair_tHv3 · 2025-08-04
**Reminder: Please Review Author Responses**

Dear Reviewers,

As the author-reviewer discussion deadline approaches, I would like to remind you to please carefully review the author responses. This rebuttal phase is a critical opportunity for authors to clarify misunderstandings, address concerns, and provide supplementary evidence for their work.

When evaluating the rebuttals, please consider the authors' arguments and assess whether your initial concerns have been satisfactorily addressed.

Your thorough engagement in this dialogue is vital for a fair and constructive review process and is essential for upholding the high standards of NeurIPS. Thank you for your dedication and expertise.

Best regards,

Area Chair

---

### Decision · Program_Chairs · 2025-09-17

**Decision:**

Accept (poster)

**Comment:**

This work develops new dynamic graphs for communication-efficient decentralized learning. Building on the 1-peer exponential graph, the authors propose the k-peer exponential and null-cascade graphs, with the latter guaranteeing finite-time convergence for any number of nodes and demonstrating superior empirical performance.

During the rebuttal, the authors provided detailed experiments and derivations that clarified the reviewers’ concerns. All reviewers  recommend accepting the paper.